# ARE LLMS READY FOR ENGLISH STANDARDIZED TESTS? A BENCHMARKING AND ELICITATION STUDY

## ABSTRACT

Large language models (LLMs) are transforming education by enabling powerful tools that enhance learning experiences, particularly in the context of English Standardized Tests (ESTs), which generate significant commercial value in the education industry. However, their fundamental problem-solving capabilities remain largely underexplored. In this work, we evaluate the performance of LLMs on ESTs across a diverse range of question types. We introduce ESTBOOK, a comprehensive benchmark designed to evaluate the capabilities of LLMs in solving EST questions. ESTBOOK aggregates five widely recognized tests, encompassing 29 question types and over 10,576 questions across multiple modalities, including text, images, audio, tables, and mathematical symbols. Using ESTBOOK, we systematically evaluate both the accuracy and inference efficiency of LLMs. Additionally, we propose a breakdown analysis framework that decomposes complex EST questions into task-specific solution steps. This framework allows us to isolate and assess LLM performance at each stage of the reasoning process. Evaluation findings offer insights into the capability of LLMs in educational contexts and point toward targeted strategies for improving their reliability as intelligent tutoring systems.

## 1 INTRODUCTION

AI-driven tools are rapidly transforming the education industry, with large language models (LLMs) increasingly integrated into English Standardized Tests (ESTs) such as TOEFL, IELTS, and GRE. Recent advances highlight the use of LLMs in automated scoring and grading (Xia et al., 2024; Zhong et al., 2024; Gupta, 2023), test preparation and tutoring (Feng & Wang, 2024; Ashrafimoghari et al., 2024), and even question generation for practice material (Tiratatri et al., 2025).

However, those works have directly concentrated on complex downstream applications. Before LLMs can be reliably deployed for higher-level educational functions such as adaptive tutoring (Stamper et al., 2024; Molina et al., 2024), personalized feedback (Maiti & Goel, 2024; Alsafari et al., 2024), or large-scale exam designs (Zhang et al., 2023; Askarbekuly & Aničić, 2024), it is essential to first establish their fundamental capability in raw problem solving. The ability to answer EST questions correctly is the foundation upon which the higher-level applications can subsequently be built. Yet, the reliability of LLMs in solving ESTs remains largely unexamined, particularly across the diverse formats that such tests encompass (e.g., reading comprehension, essay writing, and mathematical reasoning), which are often presented with multimodal structures (Grapin & Llosa, 2022).

In this work, we benchmark the problem-solving capabilities of LLMs with a broad focus on five internationally recognized ESTs: (1) two language proficiency assessments—TOEFL and IELTS, and (2) three standardized knowledge-based exams—SAT, GRE, and GMAT. To systematically evaluate LLMs, we introduce ESTBOOK, a comprehensive benchmark designed to assess their performance across a wide range of EST tasks. ESTBOOK includes 29 question types drawn from the five exams, totaling 10,576 examples. As illustrated in Figure 1, ESTBOOK spans multiple modalities, including text, images, audio, tables, and mathematical symbols, enabling a rigorous and multimodal evaluation of LLMs' problem-solving abilities.

Using ESTBOOK, we first evaluate industry-leading LLMs (e.g., GPT-5, Gemini, Llama, and Claude) with foundational prompting strategies: In-Context Learning (ICL), Chain-of-Thought (CoT), and Tree-of-Thought (ToT). Our evaluation yields the following observations:

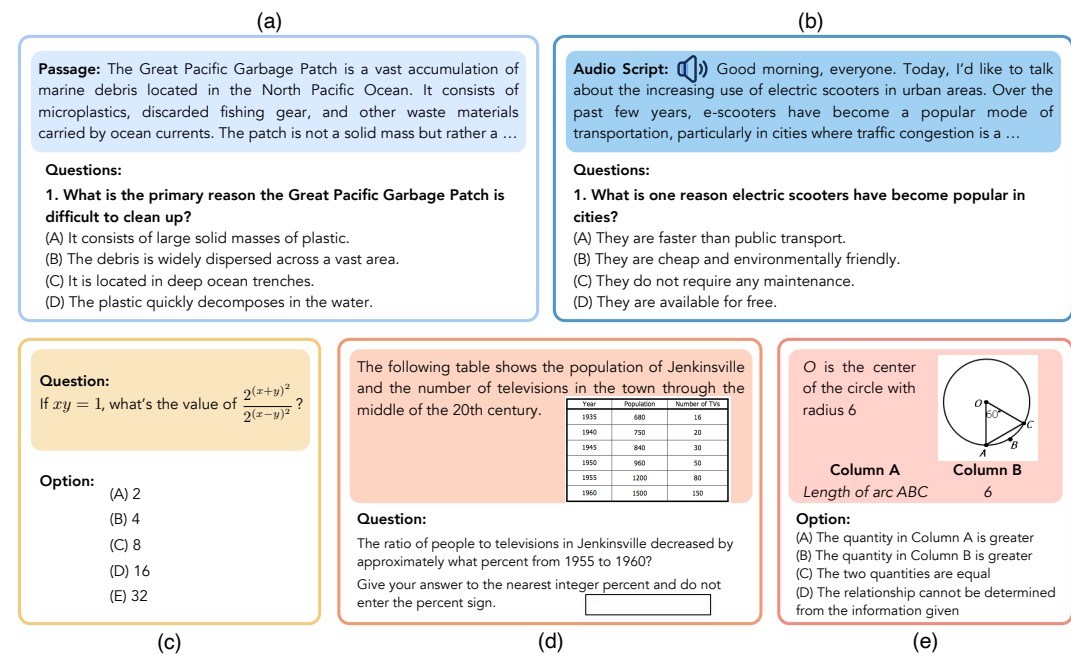

Figure 1: Examples of multimodal questions included in ESTBOOK: (a) a reading comprehension question (text) from IELTS, (b) a listening comprehension question (audio) from TOEFL, and (c), (d), and (e) GRE quantitative questions involving math symbols, tabular data, and images, respectively.

First, we find that LLMs, despite extensive pretraining on large English corpora (Mahmud et al., 2025; Sapkota et al., 2024; Welleck et al., 2024), exhibit limited effectiveness in EST-style problem-solving. In addition, performance varies substantially across question types and domains. Furthermore, in some cases, models may incur variant inference latency without producing correct answers. These results suggest that, although LLMs demonstrate strong general language capabilities, they remain inadequate as educational assistants directly for EST tasks.

To inform future development, we propose a **Breakdown Analysis**, a diagnostic framework tailored to each question type and aligned with how human test-takers approach problem solving. For example, in TOEFL reading comprehension, we first assess whether the model can identify the intention of a question; we then inform LLMs with correct intention and evaluate its ability to extract relevant evidence from long passages. In GRE quantitative questions, we analyze the model's ability to select appropriate mathematical operations before evaluating execution accuracy. This breakdown strategy helps isolate specific reasoning process and highlight strengths of LLMs toward more effective development of intelligent educational systems.

To summarize, this work makes several contributions: **(1) Benchmark** – We introduce ESTBOOK, a benchmark that offers a diverse set of EST tasks to enable comprehensive, multimodal evaluation of LLMs. **(2) Empirical Study** – We conduct extensive studies across LLMs and reveal their insufficiency for EST problem-solving and exhibit inconsistent performance across questions. **(3) In-Depth Analysis** – We propose breakdown analysis to identify LLM capability in each isolated reasoning step, which provides actionable insights to inform reliable development of educational systems. Dataset and code are available at: `https://anonymous.4open.science/r/Education-9595`.

## 2 RELATED WORK

**LLMs for Education.** LLMs are increasingly used in education for tasks like grading (Chang et al., 2024; Holmes & Tuomi, 2022), question generation (Mohebbi, 2025; Zhang et al., 2024a), and tutoring (Schmucker et al., 2024). Early systems like Codex and ChatGPT showed that LLMs can

help students across STEM and language learning by offering contextual feedback and answers (Chiang et al., 2024; Liang et al., 2024; Wen et al., 2024b). More recent research explores using LLMs as conversational tutors (Sabri et al., 2025). However, most existing evaluations rely on limited benchmarks or informal studies, often focusing on narrow skills like math or reading (Guilherme, 2019; Lee et al., 2023).

**Benchmarking LLMs.** Many benchmarks test LLMs on general or domain-specific reasoning tasks, evaluating their factual knowledge and reasoning abilities. Some, like MathVista (Lu et al., 2023; Peng et al., 2024) and ScienceQA (Wen et al., 2024a; Zhang et al., 2024d), include images or structured data, but often use synthetic problems or cover narrow domains. Our benchmark, ESTBOOK, is grounded in real standardized exams and spans multiple formats (e.g., multi-choice, text completion), providing a more realistic test of LLMs as educational agents in a heterogeneous problem-solving environment.

**Eliciting LLM Reasoning.** Improving LLM reasoning through prompting has become an emerging subject. Techniques like In-Context Learning (ICL) (Koike et al., 2024; Yugeswardeenoo et al., 2024), Chain-of-Thought (CoT) (Godwin-Jones, 2024; Wang et al., 2024), and Tree-of-Thought (ToT) (Zhang et al., 2024b;c) guide models to generate step-by-step answers and improve performance on tasks like math and logic. Yet, these methods are mostly tested on clean, synthetic datasets (Askarbekuly & Aničić, 2024; Schmidhuber & Kruschwitz, 2024). We evaluate LLMs on ESTBOOK and show their limitations in real EST questions. Additionally, we break down problem-solving steps based on question structure and offer insights about LLMs eligibility on each isolated reasoning step.

## 3 ESTBOOK: BENCHMARKING ENGLISH STANDARDIZED TESTS

### 3.1 ENGLISH STANDARDIZED TESTS, INVOLVED MODALITIES, AND DATA SOURCES

**English Tests.** As shown in Table 1, the benchmark covers 10,576 questions and 29 types across five major ESTs: *SAT*, *GRE*, *GMAT*, *TOEFL*, and *IELTS*. These exams play critical roles in academic and professional escalation: *(1) SAT* is widely used for undergraduate admissions in the United States, assessing students' readiness for college through verbal and mathematical reasoning. *(2) GRE* is a common requirement for graduate school admissions, designed to evaluate verbal and quantitative reasoning skills. *(3) GMAT* serves as a gatekeeping exam for business school programs, emphasizing critical thinking, data interpretation, and logical reasoning.*(4) TOEFL* and *IELTS* are the two most widely recognized tests for evaluating English language proficiency among non-native speakers, commonly required for university admissions and immigration purposes in English-speaking countries. Among those tests, ESTBOOK focuses on **objective questions**, as they have certain answers and thus facilitate evaluations.

**Modalities.** ESTBOOK captures the structural and cognitive diversity among several modalities: text (T), math symbols (S), images (I), tables (Tb), and audio (A). These modalities reflect the multimodal nature of real-world ESTs, where students are required not only to process textual information but also to interpret mathematical expressions and visual data. For example, GRE and GMAT quantitative sections often combine symbolic reasoning with tabular and graphical inputs, while TOEFL and IELTS listening sections assess a learner's ability to extract key information from spoken passages. With this wide range of input formats, ESTBOOK evaluates LLMs' problem-solving capabilities in heterogeneous environment, which offers insights into how different modalities affect reasoning.

**Sources.** The data in ESTBOOK are sourced from publicly available educational materials and official preparation resources affiliated with each standardized test. Specifically, we collect questions from released practice exams (Appelrouth & Zabrucky, 2017; IELTS-up, 2023; Woldoff & Kraynak, 2015), official preparation guides (Graduate Management Admission Council (GMAC), 2025; Gruber, 2011; TOEFL Test Prep, 2023; Woldoff, 2024; College Board, 2022; Hatch et al., 2023), and open-access educational platforms (Josué et al., 2023; Pereira et al., 2024; SAT Questions, 2023; Mallik, 2025; GMAT Club, 2025) that align with the formats and content of SAT, GRE, GMAT, TOEFL, and IELTS. To ensure data diversity and authenticity, we include samples spanning different years, question formats, and difficulty levels. For multimodal questions, such as those involving tables, images, or audio clips, we reconstruct representative content that mirrors real test conditions, ensuring fidelity to the original test design while maintaining licensing compliance. All questions went through validation on their sourced websites for correctness, clarity, and alignment with the original intent of

Table 1: question types, their descriptions, number of instances, involved modalities, and involved tasks (defined in Section 3.2). Modality: "T"–text, "S"–math symbol, "I"–Image, "Tb"–tabular data, "A"–audio. Concrete examples are shown in Appendix B.

| Section | Question Type (Abbreviation) | Description | Num | Modality | Task |
|---|---|---|---|---|---|
| **SAT** | | | | | |
| Reading & Writing | Information and Ideas (II) | Assess comprehension, reasoning, and inference skills | 180 | T | I,II |
| | Craft and Structure (CS) | Test vocabulary and how authors structure their writing | 636 | T | III |
| | Expression of Ideas (EI) | Test the logical flow and effectiveness of writing | 210 | T | III |
| | English Conventions (EC) | Focus on grammar, punctuation, and sentence structure | 150 | T | I,III |
| Math | Algebra (AG) | Test numeric equations, functions, and inequalities | 243 | T,S | IV,V |
| | Data Analysis (DA) | Interpret ratios, percentages, probabilities, and graphs | 141 | T,S | V |
| | Geometry & Trigonometry (GT) | Analyze angles, circles, areas, and trigonometric functions | 153 | T,S | IV |
| **GRE** | | | | | |
| Verbal | Text Completion (TC) | Fill in blank(s) (one/two/three) within a short passage | 620 | T | III |
| | Sentence Equivalence (SE) | Choose two words with the same meaning | 620 | T | VI |
| | Reading Comprehension (RC) | Answer questions based on a passage | 562 | T | I,II |
| Quantitative | Quant Comparison (QC) | Compare two quantities and select their relationship | 150 | T,S | VI |
| | Numeric Entry (NE) | Type the exact numerical answer | 150 | T,S | V |
| | Data Interpretation (DI) | Multi-choice questions from graphs, tables, or charts | 150 | T,S,I,Tb | IV,V |
| **GMAT** | | | | | |
| Verbal | Critical Reasoning (CR) | Analyze and evaluate an argument | 244 | T | III |
| | Reading Comprehension (RC) | Answer questions based on a passage | 408 | T | I,II |
| Quantitative | Problem Solving (PS) | Algebra, arithmetic, numerical, and statistical problems | 408 | T,S | IV,V |
| Data Insights | Data Sufficiency (DS) | Decide if a statement is sufficient to answer a question | 400 | T,S | IV |
| | Integrated Reasoning (IR) | Analyze tables, graphs, charts, or multiple sources | 340 | T,S,I,Tb | IV |
| **TOEFL** | | | | | |
| Reading | Factual Information (FI) | Identify facts in (or not in) the passage | 620 | T | I |
| | Inference & Reference (IR) | Infer information/word meaning/pronoun in context | 415 | T | II |
| | Text & Sentence (TS) | Insert texts, simplify a sentence, summarize a passage | 310 | T,Tb | I,II |
| Listening | Factual Information (FI) | Identify facts in (or not in) the lecture/conversation | 300 | A,T | I |
| | Inference (IF) | Understand tone/intention/opinion/relationship of ideas | 150 | A,T | II |
| **IELTS** | | | | | |
| Reading | Identifying Information (II) | Identify correctness of statement or author's opinion | 296 | T | I |
| | Matching Sentence (MS) | Match head, opinion, or sentence endings | 208 | T | II,VI |
| | Completion (CP) | Complete sentence/summary/note/table/diagram label | 592 | T,I,Tb | III |
| Listening | Identification & Matching (IM) | Determine correct answers from the audio | 520 | A,T | VI |
| | Completion & Labeling (CL) | Complete a sentence or visual with words from the audio | 1048 | A,T,I,Tb | III |
| | Short Answer (SA) | Answer briefly using words from the recording | 352 | T,I,Tb | I,II |

the corresponding exam section. We provide additional details regarding question quality control and copyright availability in Appendix A.

## 3.2 A Taxonomy of Tasks

To facilitate structured problem-solving with LLMs, we categorize each EST question type by aligning it with real-world cognitive-computational strategies commonly used in test preparation. As shown in Table 1, each question type is mapped to a specific task, which corresponds to a breakdown solution, i.e., a step-by-step reasoning path grounded in standardized test-solving strategies. We identify six distinct task categories as follows:

- **Task I: Evidence Finding** *(Breakdown: Identify Subject → Comprehend Text/Audio → Extract Discourse)* – This task involves identifying the central subject of the question, locating relevant textual or auditory evidence, and applying reasoning to extract the correct answer. It is common in reading comprehension sections of tests like GRE and TOEFL.

- **Task II: Semantic Reasoning** *(Breakdown: Parse Semantics → Localize Logical Scope → Resolve Contextual Meaning)* – This task requires interpreting fine-grained sentence semantics and resolving logical relationships or equivalence. Examples include GRE Sentence Equivalence and GMAT Critical Reasoning questions.

- **Task III: Structural Reasoning** *(Breakdown: Parse Syntactic Structure → Match Text → Predict Missing Element)* – Tasks such as sentence completion (GRE) or grammatical error detection (GMAT) fall into this category, where models must first analyze syntax and then select appropriate tokens to complete or correct the sentence.

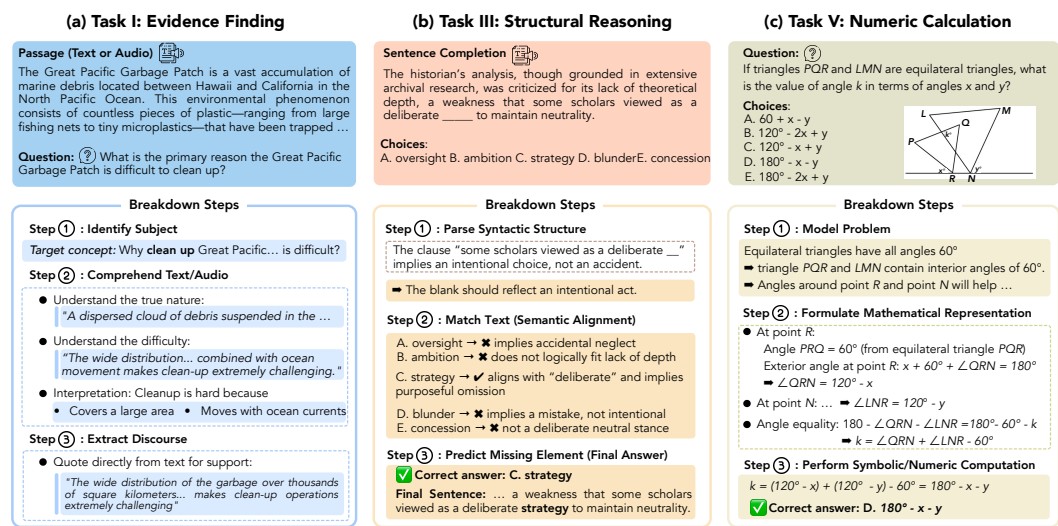

Figure 2: Illustrative breakdown examples for solving EST questions.

- **Task IV: Data Interpretation** *(Breakdown: Formulate Analytical Goal → Parse Visual/Tabular Input → Analyze Data)* – Multimodal questions involving tables, charts, or diagrams require the model to interpret visual structures, extract relevant data, and perform computations. This applies to GRE Data Interpretation and GMAT Integrated Reasoning.

- **Task V: Numeric Calculation** *(Breakdown: Model Problem → Formulate Mathematical Representation → Perform Symbolic/Numeric Computation)* – This task is typical of math-focused questions that require translating natural language descriptions into formal mathematical expressions, followed by symbolic manipulation or numeric computation. Examples include SAT Math and GMAT Problem Solving sections.

- **Task VI: Comparative Judgment** *(Breakdown: Identify Comparative Entities → Apply Constraints → Evaluate Logical Relationship)* – This task evaluates whether the model can assess sufficiency, equivalence, or constraint satisfaction, as seen in GRE Quantitative Comparison and GMAT Data Sufficiency questions.

In Appendix C, we provide details on how the breakdown steps align with the problem-solving processes employed by human test-takers across different categories of EST tasks.

## 4 EXPERIMENT

ESTBOOK aims to empirically answer several research questions: **RQ$_1$:** How do different LLMs perform on EST problem-solving tasks under various prompting strategies? **RQ$_2$:** What is the inference-time efficiency of LLMs across different EST question types? **RQ$_3$:** How effective are LLMs at completing individual steps within structured problem-solving workflows?

**LLMs.** Given the multimodal nature of ESTBOOK, we evaluate several industry-leading Multimodal LLMs, including GPT-5, GPT-4V, Claude-Sonnet-4, Llama-4-Scout-17B, Qwen-VL-Max, and Gemini-2.5. We adopt OpenAI's Whisper (Andreyev, 2025; Graham & Roll, 2024) to transcribe audio data within listening tasks in TOEFL and IELTS.

**Human Tester.** To demonstrate LLM performance alongside humans, we employed five student testers, wherein each of whom had recently prepared for and participated in at least one of these tests, and reported their problem-solving performance along with LLMs.

**Prompting Strategy.** We evaluate three popular prompting methods: **(1) In-Context Learning (ICL)**: Besides basic instructions to describe the question type, the prompt also includes several (we select five) examples to offer LLMs the solution style. **(2) Chain-of-Thought (CoT)** (Bi et al.,

Table 2: Results on ESTBOOK. Human performance is reported as the average across five independent testers. For each method's results, we report the average over five independent runs with adjusted temperatures (0.2, 0.5, 0.7, 0.9, 1.0). Standard deviations are shown in Table 6.

| Task | Human | GPT-4V | | | GPT-5 | | | Claude-Sonnet-4 | | | Llama-4-Scout-17B | | | Qwen-VL-Max | | | Gemini-2.5 | | |
|---|---|---|---|---|---|---|---|---|---|---|---|---|---|---|---|---|---|---|---|
| | | ICL | CoT | ToT | ICL | CoT | ToT | ICL | CoT | ToT | ICL | CoT | ToT | ICL | CoT | ToT | ICL | CoT | ToT |
| *SAT* | | | | | | | | | | | | | | | | | | | |
| II | 82.1 | 73.3 | 75.6 | 80.6 | 72.8 | 82.2 | 86.4 | 83.9 | 85.4 | 89.7 | 81.2 | 87.5 | 90.4 | 88.3 | 89.2 | 91.5 | 85.0 | 87.3 | 92.6 |
| CS | 74.0 | 68.2 | 77.4 | 84.9 | 73.9 | 82.7 | 87.2 | 55.8 | 70.2 | 66.3 | 46.4 | 61.8 | 55.7 | 75.6 | 82.9 | 78.8 | 93.7 | 94.1 | 87.2 |
| EI | 77.5 | 78.1 | 79.5 | 78.6 | 84.0 | 84.5 | 82.1 | 50.5 | 62.4 | 64.8 | 48.6 | 52.5 | 51.2 | 59.0 | 61.9 | 66.1 | 72.4 | 70.4 | 71.8 |
| EC | 89.0 | 84.7 | 89.3 | 81.3 | 93.8 | 92.2 | 84.7 | 72.0 | 74.2 | 70.1 | 64.0 | 65.4 | 63.9 | 81.3 | 77.2 | 79.3 | 93.3 | 90.6 | 92.1 |
| AG | 55.1 | 28.4 | 44.0 | 60.9 | 31.7 | 53.2 | 76.1 | 33.3 | 52.6 | 79.4 | 30.0 | 46.7 | 68.3 | 35.8 | 50.1 | 81.6 | 34.2 | 52.7 | 82.4 |
| DA | 77.9 | 56.7 | 70.9 | 85.8 | 60.3 | 78.2 | 90.7 | 58.2 | 71.4 | 90.1 | 51.1 | 60.2 | 87.3 | 54.6 | 67.2 | 89.2 | 53.9 | 69.7 | 88.0 |
| GT | 63.0 | 66.7 | 64.7 | 67.3 | 73.9 | 70.5 | 71.6 | 49.7 | 50.8 | 47.2 | 41.2 | 44.8 | 38.1 | 33.3 | 30.4 | 32.5 | 58.8 | 59.0 | 56.2 |
| *GRE* | | | | | | | | | | | | | | | | | | | |
| TC | 76.2 | 72.6 | 77.4 | 83.1 | 68.5 | 73.4 | 82.1 | 69.4 | 75.5 | 72.4 | 53.5 | 61.0 | 64.8 | 67.7 | 73.1 | 78.3 | 68.5 | 80.2 | 82.4 |
| SE | 81.5 | 78.9 | 81.0 | 79.8 | 87.7 | 86.5 | 87.2 | 85.5 | 82.1 | 83.5 | 66.0 | 67.5 | 63.2 | 71.8 | 73.2 | 71.6 | 77.6 | 74.8 | 75.9 |
| RC | 70.2 | 67.1 | 77.8 | 86.1 | 83.6 | 87.1 | 81.5 | 61.9 | 69.3 | 76.0 | 46.3 | 54.2 | 73.2 | 70.6 | 76.2 | 80.1 | 56.9 | 73.2 | 78.6 |
| QC | 68.1 | 55.3 | 57.3 | 51.3 | 82.0 | 84.1 | 83.8 | 41.3 | 48.2 | 44.6 | 51.3 | 56.0 | 42.7 | 54.7 | 50.3 | 45.7 | 48.0 | 58.4 | 53.1 |
| NE | 73.7 | 32.7 | 38.0 | 52.7 | 28.7 | 33.9 | 48.2 | 17.3 | 25.0 | 37.2 | 23.3 | 30.1 | 44.5 | 29.3 | 28.1 | 40.8 | 26.0 | 33.0 | 30.2 |
| DI | 55.5 | 52.0 | 56.0 | 73.3 | 32.7 | 36.5 | 63.2 | 21.3 | 25.7 | 50.1 | 40.0 | 41.2 | 65.1 | 38.7 | 40.5 | 61.7 | 48.0 | 47.2 | 67.1 |
| *GMAT* | | | | | | | | | | | | | | | | | | | |
| CR | 66.2 | 62.3 | 77.9 | 72.5 | 57.4 | 70.1 | 71.4 | 55.7 | 79.5 | 74.8 | 65.6 | 69.2 | 71.3 | 57.4 | 75.6 | 70.2 | 56.1 | 74.4 | 72.7 |
| RC | 82.1 | 79.2 | 88.7 | 91.4 | 65.2 | 71.4 | 75.6 | 63.5 | 81.1 | 86.2 | 47.3 | 74.5 | 70.3 | 68.6 | 74.4 | 76.8 | 59.1 | 75.0 | 77.4 |
| PS | 73.7 | 24.0 | 34.3 | 41.2 | 26.0 | 31.1 | 54.2 | 19.1 | 24.5 | 27.2 | 18.6 | 22.5 | 35.0 | 22.1 | 25.6 | 33.7 | 25.0 | 28.3 | 38.4 |
| DS | 52.0 | 14.5 | 26.8 | 24.5 | 13.5 | 32.4 | 40.8 | 12.0 | 16.0 | 19.2 | 13.8 | 14.5 | 20.1 | 14.8 | 21.0 | 23.6 | 9.0 | 13.5 | 22.0 |
| IR | 59.2 | 11.2 | 13.8 | 22.1 | 11.8 | 16.0 | 20.3 | 8.8 | 15.0 | 17.4 | 3.2 | 16.2 | 18.0 | 10.0 | 11.2 | 18.7 | 12.1 | 14.4 | 20.5 |
| *TOEFL* | | | | | | | | | | | | | | | | | | | |
| FI | 86.5 | 82.3 | 86.3 | 74.2 | 85.5 | 93.2 | 70.5 | 76.6 | 83.9 | 82.0 | 65.3 | 68.8 | 65.7 | 73.2 | 70.5 | 75.1 | 73.5 | 84.1 | 86.3 |
| IR | 74.1 | 63.4 | 85.3 | 87.7 | 79.3 | 84.2 | 85.0 | 55.9 | 59.2 | 63.0 | 46.0 | 62.2 | 58.3 | 73.5 | 74.0 | 75.2 | 79.0 | 81.0 | 82.6 |
| TS | 85.0 | 83.9 | 86.1 | 84.8 | 83.9 | 84.0 | 81.7 | 83.5 | 85.0 | 82.4 | 74.2 | 75.8 | 73.0 | 73.5 | 75.5 | 76.2 | 66.1 | 67.0 | 66.8 |
| FI | 93.1 | 93.7 | 95.7 | 97.7 | 94.0 | 93.2 | 98.5 | 80.7 | 86.5 | 82.5 | 67.7 | 69.2 | 76.6 | 74.7 | 70.8 | 76.3 | 81.3 | 92.5 | 89.7 |
| IF | 70.1 | 62.0 | 64.7 | 67.3 | 81.3 | 88.4 | 90.8 | 70.7 | 82.0 | 79.1 | 55.3 | 58.8 | 61.2 | 53.3 | 62.4 | 55.8 | 68.0 | 72.8 | 80.2 |
| *IELTS* | | | | | | | | | | | | | | | | | | | |
| II | 82.0 | 79.1 | 84.8 | 82.8 | 81.1 | 86.0 | 88.4 | 79.1 | 82.0 | 79.5 | 75.7 | 74.5 | 71.3 | 73.0 | 76.2 | 74.1 | 83.1 | 84.2 | 86.0 |
| MS | 93.6 | 83.7 | 85.1 | 81.3 | 81.7 | 83.0 | 83.7 | 73.1 | 81.0 | 83.2 | 66.8 | 74.0 | 76.0 | 69.2 | 71.2 | 73.7 | 75.5 | 82.5 | 80.4 |
| CP | 71.8 | 66.0 | 67.2 | 72.1 | 83.1 | 82.4 | 84.0 | 71.8 | 84.4 | 85.5 | 58.4 | 73.1 | 75.6 | 73.5 | 72.4 | 76.7 | 82.1 | 81.0 | 83.9 |
| IM | 86.1 | 83.7 | 84.8 | 88.3 | 90.6 | 91.5 | 92.8 | 74.0 | 76.0 | 75.1 | 64.2 | 66.4 | 68.3 | 73.1 | 72.0 | 74.8 | 83.7 | 89.2 | 91.6 |
| CL | 88.3 | 80.5 | 84.6 | 83.1 | 83.6 | 91.0 | 90.4 | 72.5 | 74.8 | 73.0 | 41.3 | 61.0 | 66.7 | 58.2 | 64.4 | 67.3 | 76.1 | 82.0 | 88.5 |
| SA | 85.1 | 83.0 | 86.4 | 84.7 | 83.0 | 85.1 | 84.0 | 73.9 | 77.0 | 75.0 | 66.2 | 70.2 | 67.6 | 73.3 | 76.4 | 74.7 | 82.1 | 84.9 | 83.2 |

2025; Zhang et al., 2024b): The prompt encourages the model to generate intermediate reasoning steps before generating final answers. **(3) Tree-of-Thought (ToT)** (Long, 2023; Yao et al., 2023): An advanced strategy that guides the model to explore multiple reasoning paths and select the most plausible one. Prompt layouts are shown in Appendix D. Metrics are detailed in Appendix E.2.

Statistical significance tests (McNemar's test for paired proportions, see Table 5 in Appendix E) confirm that performance differences between humans and the best-performing LLM are statistically significant ($p < 0.05$) for 26 out of 29 question types, with particularly large and significant gaps in numeric calculation tasks (SAT-AG, SAT-DA, GRE-NE: all $p < 0.001$) and multimodal data interpretation tasks (GMAT-IR, GRE-DI: $p < 0.01$).

## 4.1 EVALUATING LLMs PERFORMANCE ON ESTBOOK (RQ$_1$)

**Problem-Solving Abilities.** Table 2 presents the performance of various LLMs on ESTBOOK. Despite extensive pretraining on large-scale English corpora, these models exhibit substantial variability across different EST tasks, even within similar domains and modalities. For instance, in linguistic tasks such as GRE Expression of Ideas (EI) and English Conventions (EC), GPT-4V achieves 79.5% and 89.3% accuracy, respectively, revealing its inconsistent ability to handle fine-grained distinctions in grammar, style, and logical flow. Similarly, LLMs are not always outperform human testers despite their advanced prompting methods (e.g., COT or TOT). Those observations suggest that LLMs often struggle with the contextual sensitivity required for generalizing to diverse test problems. We provide more details and insights in Appendix E.4.1.

**Influence of Modality Complexity.** The limitations of LLMs become more obvious when complex modalities are involved, such as GMAT Integrated Reasoning (IR) and GRE Data Interpretation (DI).

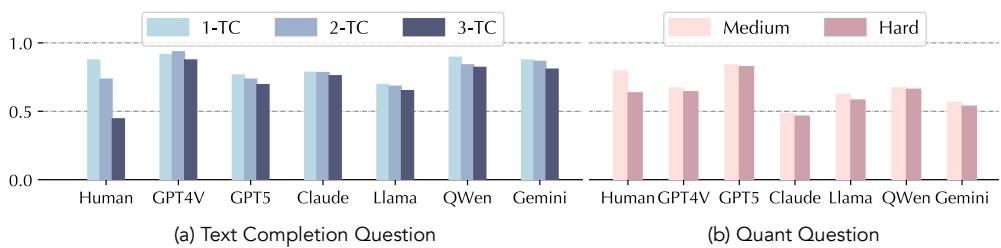

Figure 3: LLM performance across varying levels of question difficulty, using CoT due to its representativeness. We focus on GRE text completion tasks with 1-, 2-, and 3-blanks, as well as available medium- and hard-level quantitative problems.

> **Case Study I.** (GMAT – Integrated Reasoning): You are given (i) A **table** showing sales data by region and quarter (e.g., North America). (ii) A **text passage** describing factors that influenced sales in different regions (e.g., *"A new competitor entered the European market in Q2..."*).
>
> **Question:** *"Which region experienced the largest relative revenue drop between Q1 and Q2?"*
>
> **Challenges for LLMs: 1. Mapping text to table:** Claude and GPT-5 fail to connect the textual clue (*"new competitor in Europe"*) with the relevant table entry (*Europe, Q2 revenue*). **2. Reasoning with partial information:** Gemini overlooks the hint about the *competitor's impact* and fails to compare percentage drops across regions, missing the correct answer.

The challenge is twofold: first, models must align disparate representations (e.g., mapping textual queries to tabular structures); second, they must reason over incomplete or distributed evidence, a skill that current architectures and training regimes are not fully optimized for.

These multimodal failures suggest that achieving human-level performance on ESTs requires more than language modeling proficiency; it demands integrated reasoning capabilities that span visual, symbolic, and logical modalities. Together, these observations highlight the inherent difficulty of EST-style questions and the under-preparedness of even the strongest LLMs to serve as reliable tutors for real-world educational settings. We provide more studies and failure modes in Appendix E.5.

**Impact by Prompting Complexity.** We also find that more sophisticated prompting strategies (e.g., ToT) do not consistently lead to better performance, although more enriched reasoning is provided:

> **Case Study II** (Text Completion): *Although it is easy to imagine that the _____ of technological innovation has accelerated ... ... innovation has proceeded at a fairly _____ pace since the Industrial Revolution. Options: 1. (i) tempo, (ii) constant. Options: 2. (i) novelty, (ii) sporadic. Options: 3. (i) velocity, (ii) erratic.*
>
> **CoT focuses on overall sentence coherence:** The sentence suggests a contrast between the perception that innovation has *accelerated* and the ... ... Thus, the correct answer is **Option 1.**
>
> **ToT forces blank-by-blank exploration:** Branch 1: For first blank. (1.a) Option "tempo" → meaning = speed. (1.b) Option "novelty" → meaning = newness ... ... Branch 2: For the second blank (2.a) Option "constant" → meaning = unchanging ... ... **Final Answer: Option 3 (i) velocity, (ii) erratic.** (LLM gets confused due to multiple branches and partial fits.)

This suggests that complex reasoning frameworks may sometimes introduce additional cognitive overhead without corresponding gains in accuracy, particularly for models not explicitly optimized for such structured inference. Additional insights are provided in Appendix E.6.

**Note on Difficulty Categorization:** Difficulty levels in Figure 3 use official ETS metadata. GRE Quantitative questions are labeled as "medium" or "hard" in source materials; Text Completion difficulty corresponds to the number of blanks (1/2/3), a structural feature standardized in test design that correlates with cognitive complexity. We do not impose difficulty labels on exams (SAT, GMAT, TOEFL, IELTS) lacking official metadata.

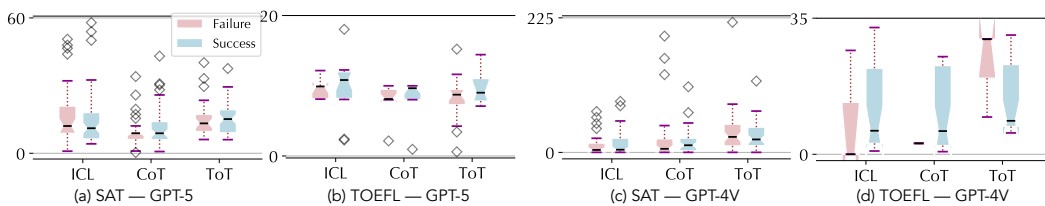

Figure 4: Inference time (in seconds) for failed and successful cases. More results are in Figure 6.

**Influence of Question Difficulty.** We further investigate how question difficulty influences LLM performance. We evaluate on GRE, which allow for clearer categorization, wherein text completion (TC) questions are divided into one-, two-, and three-blank formats with a greater number of blanks corresponds to higher difficulty. Similarly, quantitative (Quant) questions are pre-labeled as either medium or hard. Figure 3 presents LLM performance across these difficulty levels. Interestingly, we observe no clear performance degradation as difficulty increases, where human testers show a significant decline in answer correctness as the difficulty increases. These results suggest that LLMs may not be sensitive to human-defined difficulty levels and instead exhibit an equilibrium across structurally similar problems, regardless of their intended complexity in English or mathematical settings.

**Validation of Actionable Improvements.** To demonstrate that our breakdown analysis enables practical improvements, we implement a simple adaptive prompting framework based on task characteristics identified in our analysis. The framework selects ICL for factual retrieval tasks (Task I Steps 1-2), CoT for multi-step logical reasoning (Tasks II-III), and avoids ToT for pattern recognition tasks where we observe performance degradation (e.g., GRE-QC). Evaluated on a representative subset of 2,000 questions across all five exams, this adaptive approach achieves 73.8% average accuracy compared to 71.2% for uniform CoT prompting ($p < 0.01$, paired t-test), representing a 2.6 percentage point improvement. This validates that our diagnostic findings translate directly to measurable performance gains.

Appendix G further discusses how our findings can benefit real learners and the education industry.

Appendix J provides extended evaluation on three additional mainstream models, demonstrating the consistency and robustness of our findings across a broader range of LLM systems.

### 4.2 INFERENCE EFFICIENCY (RQ$_2$)

Another important consideration is the inference time of LLMs. To analyze the relationship between inference time and answer correctness, we record the generation time (in seconds) for each response and categorize the results into two groups: correctly answered and incorrectly answered questions. We then plot the distribution of inference times for both groups, as shown in Figure 4.

From the box plots, we observe that the inference times for correct and incorrect predictions are similar, without a notable separation between the two groups. To statistically validate this observation, we perform a two-tailed Mann–Whitney U test (McKnight & Najab, 2010). The Mann–Whitney U test is a non-parametric hypothesis test that assesses whether two independent samples come from the same distribution. It evaluates whether the distributions differ in location (median) or overall shape. As listed in Table 3, across all evaluated models, the Mann–Whitney U tests yield p-values higher than 0.05 (a commonly used significance level), indicating no statistically significant difference between the inference time distributions of correct and incorrect predictions. This suggests that the time an

Table 3: Mann–Whitney U test of the inference time between failed and successful cases. We report p-values to assess the statistical significance of differences between the success and failure groups. Additional results are presented in Table 7.

| Exam | GPT-5 | | | GPT-4V | | |
|---|---|---|---|---|---|---|
| | ICL | CoT | ToT | ICL | CoT | ToT |
| SAT | 0.278 | 0.814 | 0.443 | 0.197 | 0.117 | 0.512 |
| TOEFL | 0.610 | 0.389 | 0.515 | 0.295 | 0.640 | 0.115 |

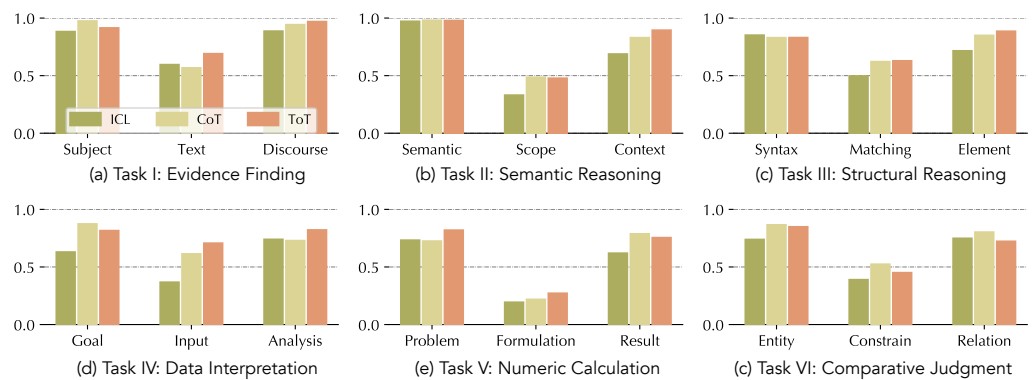

Figure 5: Breakdown analysis across all included tasks I-VI (Section 3.2) on GPT-4V.

LLM spends on answering a question does not correlate with answer correctness. Inference time appears largely independent of answer quality on the EST benchmark.

### 4.3 BREAKDOWN ANALYSIS: ISOLATED MEASUREMENT OF EACH REASONING STEP (RQ$_3$)

Next, we evaluate the step-by-step capabilities of LLMs in solving diverse EST tasks. As defined in Section 3.2, these tasks span six categories, each associated with a structured sequence of reasoning steps necessary for successful problem-solving. In our experiments, we assume the ground-truth outputs for preceding steps are provided, thereby isolating each reasoning stage and avoiding the compounding of upstream errors. Figures 5 and Appendix E present the breakdown analysis results, with detailed evaluation metrics in Appendix E.2. We derive the following observations:

**LLMs Are Strong Formulators but Weak Reasoners.** Overall, we observe that LLMs consistently excel in the initial step across all tasks, achieving up to 97% accuracy. These early steps—such as task identification, problem formulation, or topic modeling—demonstrate the models' strong capability to interpret and structure EST problems appropriately. However, performance significantly declines in subsequent reasoning steps, which vary across tasks. This decline is particularly evident in tasks that demand causality inference or evidence synthesis.

> **Case Study III (GMAT – Critical Reasoning):** "*A recent study found that cities with more EV charging stations tend to have lower levels of air pollution. As a result, the city of Greentown has decided to install a large number of EV charging stations to reduce its pollution levels.*"
>
> **LLM Reasoning:** GPT-5 and Claude incorrectly state "*The city of Greentown currently has a very small number of EVs in use*" due to being distracted by the phrase "*the current number of EVs*" that does not directly relevant to causal logic.

These results suggest that while LLMs are proficient at understanding and framing problems, they remain limited and unstable in executing complex reasoning chains—an essential requirement for robust educational support.

**Complex Logic Has More Impact on LLM Screening than Long Context.** We find that context length alone does not impede LLMs to locate relevant information. Instead, reasoning complexity plays a greater role in determining success or failure. Models can navigate long inputs effectively if the task only requires surface-level matching, but they often fail when logical integration across multiple sentences is required. These results suggest that long context alone is not a major barrier for modern LLMs. However, once the task requires multi-hop reasoning or integrating dispersed evidence, even top-performing models struggle.

**Numeric Entry and Multi-Modality Significantly Impede LLM Reasoning.** Tasks involving numeric input and multimodal understanding (e.g., math from SAT) remain particularly challenging for LLMs. Unlike classification-style questions with fixed answer choices, numeric-entry tasks require precise mathematical formulation, symbolic manipulation, and error-free calculation—all of which are error-prone in current models.

**Case Study IV (GRE Quant – Numeric Entry):** *"If the sum of three consecutive odd integers is 111, what is the smallest of the three?"*

**LLM Reasoning:** Claude generates an incorrect expression: $x+x+1+x+2 = 111$ as it treats the numbers as consecutive integers rather than odd integers (which should be $x+x+2+x+4 = 111$).

The challenge is amplified in multimodal settings, where the model must align visual, tabular, or symbolic inputs with textual queries before reasoning can even begin.

**Case Study V (GMAT Integrated Reasoning – Table + Math Computation):** A question requires *"selecting a product with the highest profit margin based on a table of costs and revenues."* GPT-4V incorrectly reads the table and subtracts the cost from total units sold rather than revenue, leading to an invalid numeric result.

Due to space constraints, additional findings and case studies are provided in Appendix E.3 and E.7.

## 5 CONCLUSION

This work explores the potential and limitations of LLMs in problem-solving on English Standardized Tests (ESTs). Through the construction of ESTBOOK, a multimodal and diverse benchmark encompassing five major ESTs, we provide a rigorous framework for evaluating LLMs across a variety of question types and modalities. Our empirical findings reveal that, despite their linguistic fluency, current LLMs fall short in consistently solving EST-style problems and display notable variation in performance across domains. Furthermore, our proposed breakdown analysis highlights specific reasoning failures, offering a granular diagnostic approach to inform model development.

**Implications for Future Improvements** Our breakdown analysis directly informs targeted interventions. The visual-tabular parsing bottleneck at Task IV Step 2 (dropping from 87% to 51–65%) suggests specialized training objectives such as contrastive learning for cross-modal alignment. The mathematical formulation gap at Task V Step 2 (21-point drop) motivates symbolic execution modules to verify expression correctness before computation. The formulation-execution gap (24–33 point drops from Step 1 to Step 3) points toward hierarchical reasoning architectures separating planning from execution. Our finding that prompting effectiveness varies with task structure suggests adaptive frameworks that select ICL, CoT, or ToT based on reasoning depth and modality complexity. Future work can use our breakdown framework as an evaluation protocol to verify interventions address root causes rather than achieving spurious aggregate improvements.

ETHICS STATEMENT

This work exclusively relies on publicly available standardized test preparation resources, official sample questions, and open educational platforms, all used in compliance with their respective copyright and licensing terms. No proprietary or sensitive data were accessed. Additional details regarding data sourcing and copyright considerations are included in Appendix A. Therefore, we do not identify any ethical concerns arising from this study.

REPRODUCIBILITY STATEMENT

To support reproducibility, we have released all benchmark construction details, evaluation scripts, and experimental configurations through an anonymous GitHub repository. This repository includes instructions for evaluation procedures. The benchmark design, codebase, and detailed experimental settings are documented to readers.

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

## A    QUESTION QUALITY AND COPYRIGHT ASSURANCE

To ensure the integrity of our benchmark, we adopted a rigorous process for validating both the quality of collected questions and the copyright compliance of the sources.

**Question Quality.** All questions included in ESTBOOK were sourced from publicly released or openly accessible educational materials. We verified each item for (1) correctness, by cross-checking answer keys or explanatory notes provided by the original source; (2) clarity, ensuring that wording, figures, and formatting matched the original intent without ambiguity; and (3) authenticity, by aligning the question style and content with the design principles of the corresponding standardized test (SAT, GRE, GMAT, TOEFL, or IELTS).

**Alignment Validation Process.** In addition to verifying correctness and clarity, we systematically validated the alignment of our questions with official exam specifications. For each question type, we performed the following validation steps:

1. **Format validation:** We cross-referenced the question structure, answer format, and presentation style against official sample materials to ensure consistency. As an example, SAT Math Algebra questions in our benchmark follow the same five-option multiple-choice format and use identical equation presentation conventions as those in College Board released items.

2. **Skill mapping:** We verified that each question assesses the specific cognitive skill as defined in official test frameworks. For instance, TOEFL Reading "Factual Information" questions in ESTBOOK are designed to test the ability to identify explicitly stated details, which matches the skill definition provided by ETS. Similarly, "Inference" questions require test-takers to draw logical conclusions that go beyond the literal text, consistent with official specifications.

3. **Difficulty calibration:** Although we do not have access to proprietary difficulty ratings used by test administrators, we ensured that our question collection spans the full difficulty range found in official preparation materials. This includes both entry-level items suitable for beginners and advanced problems that challenge high-performing test-takers, providing representative coverage across each exam's difficulty spectrum.

Through this systematic validation process, combined with our reliance on officially affiliated sources, we ensure that ESTBOOK authentically captures the reasoning demands and structural characteristics of real standardized tests.

**Copyright and Usage.** To comply with licensing and intellectual property requirements, we restricted data collection to (1) officially released practice tests and preparation guides distributed for public use, and (2) open-access educational platforms and community-contributed question banks that explicitly allow free access for study and research purposes. No proprietary or paywalled materials were included. For multimodal reconstructions, all recreated content is original and designed solely to approximate the test-taking context without replicating copyrighted assets. This approach ensures that ESTBOOK respects copyright protections while still providing representative and high-quality benchmark content.

### Potential Training-Data Contamination

As ESTBOOK is built from real standardized test preparation materials, a natural concern is potential overlap between our benchmark items and the pretraining corpora of modern LLMs.

#### Lack of direct access to training data.

For proprietary API models (GPT-4V, GPT-4o, Claude-3-Opus, Gemini-Pro), neither the exact training corpora nor document-level membership tests are available. Even for open-source models (Llama-3.2-90B, Qwen-VL-Max), only high-level descriptions of training sources are released. Therefore, we cannot perform a full decontamination or exact memorization check at the item level, and we explicitly treat potential training-data overlap as a limitation of our benchmark.

#### Design choices to reduce trivial memorization.

When constructing ESTBOOK, we (i) draw from a mixture of public practice exams, official guides, and open educational platforms rather than one particular "famous" test book, (ii) reconstruct tables, figures, and audio clips instead of copying digital assets verbatim, and (iii) manually normalize wording (e.g., removing page numbers, book-specific formatting cues) while preserving the original exam intent. These steps reduce the chance that an item appears in exactly the same surface form as in common web corpora, although they do not eliminate all possible overlap.

#### Why our main conclusions are robust to contamination.

Our core claims are based on large and systematic gaps between humans and LLMs on specific task categories (e.g., Tasks IV–V) and on breakdown patterns across reasoning steps, rather than on small differences between individual models. If contamination were the dominant factor, we

would expect near-ceiling performance and highly consistent accuracy across models on affected question types. Instead, we observe (i) substantial variance across tasks and models, and (ii) persistent underperformance on multimodal data interpretation and numeric calculation even for state-of-the-art systems. We thus interpret ESTBOOK as providing an upper bound on LLM performance under realistic exam-like distribution, with any residual contamination making our negative results conservative rather than overly pessimistic.

## B  COMPLEMENTARY INFORMATION OF QUESTION TYPES

This appendix provides a brief description of each question type covered across SAT, GRE, GMAT, TOEFL, and IELTS, along with a concrete example. In the examples below, long passages are truncated with "..." for brevity.

**SAT Reading & Information and Ideas (II).** Tests comprehension of written passages across diverse subjects. Students must identify main ideas, understand explicit details, make logical inferences, and draw evidence-based conclusions. Requires distinguishing between stated facts and implied meanings while recognizing the author's purpose and how ideas relate to one another. Success depends on analytical reading rather than simply locating isolated facts.

> Passage: The industrial revolution transformed societies by shifting labor from farms to factories, fundamentally altering social structures and economic relationships. As rural populations migrated to urban centers seeking employment, traditional family units were disrupted...
> Question: According to the passage, what was one major social change brought about by the industrial revolution?
> A. The development of new agricultural techniques
> B. The disruption of traditional family structures
> C. The elimination of class divisions in society
> D. The reduction in opportunities for social mobility

**SAT Writing Craft and Structure (CS).** Evaluates understanding of vocabulary in context and text organization. Students must select appropriate words based on context, analyze how ideas develop across paragraphs, and recognize how structure enhances effectiveness. Requires understanding both denotative and connotative meanings while evaluating how word choice affects tone, style, and precision of expression.

> Sentence: The scientist's findings were ______, shedding light on the mysterious behaviors of subatomic particles that had puzzled physicists for decades...
> A. groundbreaking
> B. world-class
> C. groundbreaking and thrilling
> D. groundbreaking, thrilling

**SAT Expression of Ideas (EI).** Focuses on writing effectiveness and logical flow. Students evaluate and improve coherence, cohesion, and clarity by combining sentences, reorganizing information, or modifying details. Requires determining relevance, identifying optimal placement for new information, and understanding how structural changes affect meaning and emphasis. Tests ability to develop logically connected ideas with appropriate support.

> Revision Task: Improve the coherence of the following sentence pair to create a more logical flow:
> "I love classical music. Beethoven's symphonies are my favorite."
> Possible revisions:
> A. I love classical music, especially Beethoven's symphonies, which are my favorite.
> B. Because I love classical music, Beethoven's symphonies are my favorite.
> C. I love classical music; indeed, Beethoven's symphonies are my favorite.
> D. No change necessary.

**SAT English Conventions (EC).** Assesses command of standard English grammar, punctuation, and sentence structure. Topics include verb tense/agreement, pronoun usage, parallel structure, modifier placement, and appropriate punctuation. Students must identify and correct errors in sentences or paragraphs. Evaluates practical application of grammatical rules rather than theoretical knowledge.

Sentence: Each of the students (have / has) submitted their essay on time, and the teacher (is / are) pleased with the quality of work...
A. have, is
B. has, is
C. have, are
D. has, are

**SAT Math Algebra (AG).** Tests ability to work with algebraic expressions, equations, inequalities, and functions. Requires solving linear/quadratic equations, manipulating expressions, understanding variable relationships, and analyzing functions. Students must apply algebraic concepts to model real-world situations and connect symbolic representations with graphs. Assesses both procedural fluency and conceptual understanding.

Solve for $x$: $\frac{2x-3}{x+1} = 4$
A. x = 7
B. x = -7
C. x = 7/3
D. x = -7/3
E. No solution exists

**SAT Data Analysis (DA).** Evaluates interpretation of various data forms. Students analyze ratios, rates, percentages, proportions, and probabilities while interpreting information from tables, charts, and graphs. Requires understanding statistical concepts (mean, median, mode) and using data to draw conclusions. Tests quantitative literacy skills needed for interpreting real-world numerical information.

A circle graph shows that 30% of students prefer tea, 50% coffee, and 20% water. If 200 students were surveyed, how many prefer coffee?
A. 60
B. 100
C. 40
D. 50
E. Cannot be determined from the information given

**SAT Geometry & Trigonometry (GT).** Covers geometric figures, coordinate geometry, and trigonometric relationships. Questions involve angles, lines, polygons, circles, 3D figures, coordinate systems, and trigonometric functions. Students calculate areas, perimeters, volumes, and distances while applying properties of various shapes. Tests spatial reasoning and connection of algebraic and geometric representations.

In right triangle $ABC$ with right angle at $C$, if $AC = 3$ and $BC = 4$, what is $\sin A$?
A. 3/5
B. 4/5
C. 3/4
D. 4/3
E. 5/3

**GRE Text Completion (TC).** Assesses vocabulary and comprehension by requiring completion of blanks in short passages. Students must understand author's intent, logical relationships between sentences, and overall context. Difficulty increases with multiple blanks where choices must work cohesively. Tests vocabulary breadth and understanding of how words function within complex contexts.

Though praised for its ______ innovations, the clock's design was too ______ to gain widespread adoption among consumers who valued simplicity and ease of use...
A. aesthetic ... cumbersome
B. mechanical ... simplified
C. technological ... intuitive
D. functional ... intricate
E. rudimentary ... complex

**GRE Sentence Equivalence (SE).** Requires selecting two words that create sentences with equivalent meanings when inserted. Students must identify words that produce the same overall meaning in context, understanding subtle connotative differences. Tests vocabulary depth, contextual word usage, and ability to maintain consistent meaning across different word choices.

> Her lecture was so _____ that many students struggled to stay awake.
> A. engaging
> B. soporific
> C. bewildering
> D. tedious
> E. stimulating
> F. monotonous

**GRE Reading Comprehension (RC).** Tests analysis and interpretation of complex academic passages. Students identify main ideas, recognize explicit statements, make inferences, understand author's purpose, and evaluate arguments. Requires handling sophisticated vocabulary and complex sentence structures while synthesizing information across passages and drawing conclusions from implied content.

> Passage: Advances in CRISPR technology have opened new avenues in gene therapy, offering unprecedented precision in modifying DNA sequences. Unlike earlier gene-editing methods that often resulted in unintended modifications, CRISPR-Cas9 allows scientists to target specific sections of genetic code with remarkable accuracy...
> Question: The passage suggests that CRISPR's main advantage over previous gene-editing methods is its ability to:
> A. Work faster than other methods
> B. Target specific sections of genetic code with high accuracy
> C. Completely eliminate the risk of unintended modifications
> D. Address a wider range of medical conditions
> E. Bypass ethical concerns associated with genetic manipulation

**GRE Quantitative Comparison (QC).** Presents two quantities for comparison of relative size. Tests conceptual understanding over computational ability as students analyze information, identify mathematical relationships, and determine if enough information exists to establish definitive relationships. Requires creative approaches, estimation skills, and recognition of information adequacy without necessarily performing complex calculations.

> Quantity A: $2^{10}$    Quantity B: $10^3$
> A. Quantity A is greater
> B. Quantity B is greater
> C. The two quantities are equal
> D. The relationship cannot be determined from the information given

**GRE Numeric Entry (NE).** Requires calculating exact answers without multiple-choice options. Tests ability to perform calculations accurately and follow procedures correctly without answer verification. Assesses computational skills, problem-solving strategies, and work with various numerical forms. Demands confidence in mathematical procedures and attention to units and precision.

> If a tank is filled at a constant rate and holds 65 gallons in 10 minutes, how many gallons per minute are being added to the tank? If the answer is a fraction, enter as a decimal.
> Answer box: _____

**GRE Data Interpretation (DI).** Assesses ability to analyze and interpret data in graphs, tables, or charts. Students extract information, perform calculations, recognize patterns, and draw conclusions. Requires comparing data points, calculating percentages or rates of change, and making predictions. Tests quantitative literacy and ability to work with real-world data representations.

Table: Quarterly profits (in $M) for Company X:
Q1: 10
Q2: 15
Q3: 12
Q4: 18
Question: In which quarter did Company X see the greatest increase in profit over the previous quarter? A. Q1
B. Q2
C. Q3
D. Q4
E. Cannot be determined from the information given

**GMAT Critical Reasoning (CR).** Evaluates analysis of argument structure, validity, and logical coherence. Students identify premises, conclusions, and assumptions while distinguishing relevant information and recognizing logical flaws. Often uses business scenarios requiring understanding of causation vs. correlation and sample representativeness. Tests analytical thinking crucial for business decision-making.

Argument: Because sales rose by 15% last quarter immediately following the implementation of our new marketing strategy, the new marketing strategy must be effective and should be continued without modifications in the upcoming fiscal year.
Which of the following, if true, most weakens this conclusion?

A. The company's main competitor went out of business during the same quarter.
B. The company introduced a popular new product line at the beginning of the quarter.
C. Other companies using similar marketing strategies saw comparable increases in sales.
D. The marketing strategy cost more to implement than initially projected.
E. Industry sales overall rose by 20% during the same period due to seasonal factors.

**GMAT Reading Comprehension (RC).** Similar to GRE but with more business focus. Tests understanding of complex written material, identification of main ideas, inference-making, and logical structure recognition. Passages often discuss business strategies or economic concepts. Assesses ability to distinguish stated from implied information and evaluate argument strength.

Passage: Global coffee consumption has doubled in the past decade, driven primarily by emerging markets in Asia where a growing middle class has embraced Western consumption patterns. China, traditionally a tea-drinking nation, has seen coffee consumption grow at 15% annually, compared to global growth of 2.5%...
Question: The author primarily discusses which factor driving coffee demand?
A. Changes in consumer taste preferences
B. Economic development and social status in emerging markets
C. Declining popularity of traditional tea consumption
D. Marketing strategies of Western coffee companies
E. Health benefits associated with coffee consumption

**GMAT Problem Solving (PS).** Tests mathematical knowledge across arithmetic, algebra, geometry, and statistics. Students determine problem requirements, identify relevant information, select appropriate techniques, and calculate accurately. Requires translating word problems into mathematical expressions and interpreting solutions in context, often in business-related scenarios.

If $x + y = 10$ and $xy = 21$, what is $x^2 + y^2$?
A. 52
B. 58
C. 100
D. 121
E. 142

**GMAT Data Sufficiency (DS).** A unique format assessing analytical thinking over computation. Students determine whether statements provide sufficient information to answer questions without actually solving problems. Requires evaluating statement sufficiency individually and collectively while understanding necessary vs. sufficient conditions and recognizing implied information.

Question: Is $x > 5$? (1) $2x > 10$ (2) $x^2 > 25$
A. Statement (1) ALONE is sufficient, but statement (2) ALONE is not sufficient.
B. Statement (2) ALONE is sufficient, but statement (1) ALONE is not sufficient.
C. BOTH statements TOGETHER are sufficient, but NEITHER statement ALONE is sufficient.
D. EACH statement ALONE is sufficient.
E. Statements (1) and (2) TOGETHER are NOT sufficient.

**GMAT Integrated Reasoning (IR).** Tests analysis of information from multiple sources and formats. Students interpret tables, graphs, and text while evaluating multiple information sources and solving multi-step problems. Includes multi-source reasoning, graphics interpretation, two-part analysis, and table analysis. Assesses skills for data-driven business decisions.

Table: Region A sales (in millions): Year 1: $120, Year 2: $132, Year 3: $145 Region B sales (in millions): Year 1: $90, Year 2: $101, Year 3: $114
Question: Which region's compound annual growth rate exceeded 5% over the three-year period?
A. Region A only
B. Region B only
C. Both Region A and Region B
D. Neither Region A nor Region B

**TOEFL Reading Factual Information (FI).** Evaluates ability to identify explicitly stated facts in academic texts. Students locate specific information, distinguish it from similar content, and understand its contextual significance. Requires processing academic vocabulary and syntax while focusing on directly stated rather than implied information.

Passage: Canada's boreal forest covers nearly one-third of its land area, spanning from Yukon to Newfoundland and Labrador. This vast ecosystem, dominated by coniferous trees, contains more than 1.5 million lakes and is home to endangered species such as the woodland caribou...
Question: According to the passage, what fraction of Canada is covered by the boreal forest?
A. One-quarter
B. One-third
C. One-half
D. Two-thirds

**TOEFL Inference & Reference (IR).** Tests understanding of implied information and referential relationships. Students draw logical conclusions from provided information, understand unstated relationships between ideas, and track references through pronouns and demonstratives. Assesses deeper comprehension including reading between lines and connecting ideas across text sections.

Passage: "The experiment failed again, prompting the research team to reconsider their methodology. Dr. Chen suggested they should explore alternative approaches that had shown promise in similar contexts."
Question: What does "again" imply about previous attempts?
A. This was the first time the experiment had been conducted.
B. Previous attempts had been successful.
C. Previous attempts had also failed.
D. The team had never tried this experiment before.

**TOEFL Text & Sentence (TS).** Evaluates various aspects of textual understanding including summarizing and sentence relationships. Tasks include inserting sentences appropriately, creating cohesive summaries, simplifying complex sentences, or identifying sentence functions. Tests advanced language processing including understanding textual connections, organizational structure, and purpose of different elements.

Original: Because of the severe weather conditions, we decided to cancel the outdoor concert scheduled for tomorrow evening.
Task: Combine into one sentence without changing meaning, beginning with "The outdoor concert..." A. The outdoor concert scheduled for tomorrow evening we decided to cancel because of the severe weather conditions.
B. The outdoor concert scheduled for tomorrow evening was decided to be cancelled by us because of the severe weather conditions.
C. The outdoor concert scheduled for tomorrow evening has been cancelled due to the severe weather conditions.
D. The outdoor concert, because of the severe weather conditions, scheduled for tomorrow evening we decided to cancel.

**TOEFL Listening Factual Information (FI).** Assesses comprehension of spoken academic content. Students identify explicitly stated information, distinguish between similar details, and recognize contextual significance. Requires processing natural-speed academic English despite accent variations while maintaining focus during extended listening passages.

Transcript: "Good morning, class. Today's lecture will cover photosynthesis, the process by which plants convert light energy into chemical energy. We'll first discuss the light-dependent reactions that occur in the thylakoid membrane, followed by the Calvin cycle that takes place in the stroma..."
Question: What topic will the lecture cover?
A. Cell respiration
B. Plant reproduction
C. Photosynthesis
D. Genetic engineering

**TOEFL Listening Inference (IF).** Tests understanding of implied meanings in spoken content. Students interpret speaker's tone, infer unstated opinions, understand implied connections, and determine purpose of specific statements. Requires comprehending not just words but also intonation and emphasis. Assesses ability to understand nuanced academic communication.

Speaker: "I suppose we could try that method, if all our other options have been exhausted. It's not my first choice, but at this point, we might not have many alternatives left."
Question: What does the speaker's tone suggest about their enthusiasm for the proposed method?
A. They are excited to try something new
B. They are reluctant but resigned to trying it
C. They believe it is the best available option
D. They are confident it will succeed

**IELTS Reading Identifying Information (II).** Evaluates whether statements match textual information. Students determine if statements are True (matching), False (contradicting), or Not Given (not addressed). Requires careful reading to distinguish between explicit, inferable, and absent information without introducing outside knowledge.

Passage: "Many cities have embraced rooftop gardens as a sustainable solution to multiple urban challenges. These green spaces not only provide fresh produce for local communities but also help mitigate the urban heat island effect by absorbing sunlight that would otherwise be converted to heat..."
Statement: "The author believes urban gardens are ineffective at addressing environmental challenges."
Is the statement True, False, or Not Given?

**IELTS Matching Sentence (MS).** Tests ability to connect related information pieces. Students match headings with paragraphs, sentence beginnings with endings, or statements with speakers. Requires understanding paragraph main ideas, sentence logic, and information relationships while processing content across multiple text sections.

Complete the following sentence with the most appropriate ending from the list below:
"Fossil fuels are being replaced by renewable sources..."
A. ...because they are more sustainable and environmentally friendly.
B. ...despite their continued dominance in global energy markets.
C. ...although the transition is happening more slowly than many scientists recommend.
D. ...particularly in developing economies seeking to reduce energy costs.
E. ...which has caused significant economic disruption in traditional energy sectors.

**IELTS Completion (CP).** Assesses ability to locate and transfer specific information to complete sentences, summaries, or diagrams. Students identify relevant details and transfer them accurately, often verbatim. Requires understanding text structure for efficient information location while recognizing synonyms and paraphrased content.

Summary: "The Sahara is the world's _____ desert, covering approximately _____ million square kilometers across North Africa, from the Atlantic Ocean to the Red Sea. Its name comes from the Arabic word meaning _____." Words to choose from:
A. largest, 9.2, "desert"
B. hottest, 8.7, "sand"
C. oldest, 7.5, "wilderness"
D. driest, 6.3, "emptiness"

**IELTS Listening Identification & Matching (IM).** Tests identification of specific spoken information and category matching. Students listen for details like names, numbers, and facts then select correct options. Requires processing natural-speed English despite distractions or accent variations while distinguishing between similar-sounding choices.

Audio transcript: "Welcome to our university orientation. The main campus tour will begin at the Student Center at quarter past nine. Please arrive at least ten minutes early to collect your information packets..."
Question: What time does the campus tour start?
A. 9:00
B. 9:15
C. 9:30
D. 10:15

**IELTS Completion & Labeling (CL).** Evaluates ability to listen for specific information to complete sentences, notes, or diagrams. Students identify and record specific details, often verbatim. Requires focused listening, accurate information processing, and simultaneous writing. Tests note-taking skills needed for educational and professional contexts.

[Audio describes the parts of a flower and their functions]
Diagram: Label the parts of a flower shown in the image using words from the recording:

1. _____ (outer protective layer)
2. _____ (colorful structures that attract pollinators)
3. _____ (male reproductive part containing pollen)
4. _____ (female reproductive structure)
5. _____ (produces seeds when fertilized)

**IELTS Short Answer (SA).** Tests listening for specific information and providing concise answers using the recording's words. Students identify relevant details and express them within word limits. Requires understanding question focus, quick information processing, and appropriate word selection. Assesses both receptive and productive language skills.

Audio transcript: "For our upcoming science class field trip next Thursday, we'll be visiting the botanical gardens on the north side of the city. Please remember to bring your permission slips, a notebook, appropriate footwear, and a packed lunch..."
Questions:
1. Where is the field trip? (Answer in no more than THREE words)
2. What day will the field trip take place? (Answer in no more than TWO words)
3. What time will students return to school? (Answer in no more than TWO words)

## C    Alignment of Breakdown Steps with Human Test-Taking Strategies

We provide detailed reasoning to justify how our proposed breakdown steps for each task category (Section 3.2) reflect the actual cognitive strategies adopted by human test-takers when approaching English Standardized Tests (ESTs). Our design is grounded in well-documented findings from standardized test preparation guides and empirical studies of student behaviors during exam practice (Loken et al., 2004; Board, 2025; Johnstone et al., 2006). Below, we elaborate task by task.

### Task I: Evidence Finding

**Breakdown: Identify Subject → Comprehend Text/Audio → Extract Discourse**
Human test-takers typically begin reading or listening by first identifying the *subject* of the question, which anchors attention to the relevant portion of the passage or recording. This is consistent with test-preparation strategies that emphasize "locating keywords" in the stem before scanning the material. Next, comprehension involves processing the local discourse unit (sentence or paragraph) to ensure contextual alignment. Finally, humans extract and confirm evidence, often by re-reading or re-listening to specific phrases, ensuring the answer is text- or audio-supported. This mirrors our stepwise design, which reduces the problem to progressively narrower spans of information.

### Task II: Semantic Reasoning

**Breakdown: Parse Semantics → Localize Logical Scope → Resolve Contextual Meaning**
Tasks like GRE Sentence Equivalence or GMAT Critical Reasoning require careful semantic parsing. Human test-takers begin by parsing sentence-level semantics, identifying parts of speech, and clarifying propositional meaning. They then localize the *logical scope*, such as a contrast marker ("although," "however") or a causal connector ("therefore," "because"). This enables them to frame the exact semantic relationship in question. Finally, humans resolve meaning in context, often by substituting candidate words or testing logical coherence against the surrounding passage. Our breakdown mirrors this iterative narrowing of interpretive scope, emphasizing precision in semantic alignment before choosing the correct answer.

### Task III: Structural Reasoning

**Breakdown: Parse Syntactic Structure → Match Text → Predict Missing Element**
In grammar- and structure-oriented tasks, human test-takers first parse the syntactic structure of the sentence, a process akin to diagramming or mentally chunking phrases. They then match the sentence against expected grammatical or rhetorical patterns (e.g., subject-verb agreement, parallelism, or logical sequencing). Finally, they predict the missing or corrected element—whether this is a word, phrase, or punctuation mark—that restores coherence. This aligns with instructional practices in SAT Writing or GRE Sentence Completion, which explicitly train students to map syntax before evaluating candidate solutions. Our breakdown encodes these same operations, emphasizing structural awareness as a precursor to lexical choice.

### Task IV: Data Interpretation

**Breakdown: Formulate Analytical Goal → Parse Visual/Tabular Input → Analyze Data**
For multimodal questions (tables, graphs, charts), human test-takers start by formulating the *analytical goal*, i.e., identifying what the question is asking (e.g., "compare percentages," "find a trend," "calculate an average"). This step ensures they do not waste time interpreting irrelevant details. They then parse the given input, reading axes, labels, and units with care. Only after grounding themselves in the representation do they proceed to analyze data, performing the necessary arithmetic or logical operations. Test-preparation materials repeatedly stress this sequence, "understand the task before reading the chart," as the optimal way to avoid misinterpretation. Our breakdown thus faithfully encodes this strategy.

TASK V: NUMERIC CALCULATION

**Breakdown: Model Problem → Formulate Mathematical Representation → Perform Symbolic/Numeric Computation**

Math-focused questions require human test-takers to *model the problem*, often by translating a word problem into an equation or inequality. This is followed by formulating a precise mathematical representation (e.g., setting up ratios, algebraic equations, or probability trees). Only then do they perform the actual computation. Empirical studies of SAT and GRE problem-solving show that students who rush directly into computation without adequate modeling are more prone to errors. Our breakdown enforces the disciplined progression, i.e., representation before calculation, that mirrors effective human problem-solving.

TASK VI: COMPARATIVE JUDGMENT

**Breakdown: Identify Comparative Entities → Apply Constraints → Evaluate Logical Relationship**

Tasks such as GRE Quantitative Comparison or GMAT Data Sufficiency rely on comparative reasoning. Human test-takers begin by carefully identifying the entities to be compared (e.g., "Quantity A vs. Quantity B"). They then apply given constraints, such as conditions on variable ranges or assumptions about sufficiency. Finally, they evaluate the logical relationship (e.g., greater, equal, cannot be determined). This mirrors well-known heuristic strategies taught in GRE and GMAT prep, where test-takers are explicitly trained to "test conditions systematically" rather than guess. Our breakdown captures this systematic comparison, aligning LLM reasoning with human evaluative steps.

OVERALL ALIGNMENT SUMMARIZATION

Across all six tasks, our breakdown steps were designed to decompose problem-solving into sequential operations that are (1) cognitively plausible, (2) aligned with strategies taught in test-preparation curricula, and (3) empirically observed in human test-taking behavior. This ensures that LLMs evaluated under our framework are not only solving problems correctly but are also engaging in processes analogous to human reasoning in standardized testing contexts.

# D  APPENDIX: PROMPT STRUCTURE

This appendix summarizes the skeleton templates for three prompt-engineering paradigms used in our educational reasoning experiments. Each template is presented with an explanation followed by a concrete example in a highlighted box.

1. IN-CONTEXT LEARNING (ICL) PROMPT STRUCTURE

Provides the model with solved examples to prime analogous problem solving:

- Multiple exemplars demonstrating the problem–solution pattern
- Graduated difficulty progression across examples
- Explicit identification of transferable patterns in each exemplar
- Strategic selection of examples to highlight different facets of the problem type
- Clear demarcation between example set and the target query

> **Example:** For a simple SAT algebra problem—"Solve for $x$: $(x-2)(x+3) = 0$":
> 1.Input: "Solve $(x-2)(x+3) = 0$"
>   Output: "Step 1: Recognize factors imply solutions x=2 or x=-3.Step 2: List solutions: 2, -3."

2. CHAIN-OF-THOUGHT (COT) PROMPT STRUCTURE

Guides the model through a step-by-step reasoning process:

- Instruction to decompose the task into ordered steps
- Explicit requests for intermediate calculations or justifications
- Structured step-labeling conventions (e.g., "Step 1: ...", "Step 2: ...")
- Prompts for linking each step's result to the next
- Final summary or explicit statement of the conclusion

---

**Example:** Compute the area of a triangle with base 5 and height 8:

1. Step 1: Write formula $A = \frac{1}{2} \times \text{base} \times \text{height}$.

2. Step 2: Substitute values: $A = \frac{1}{2} \times 5 \times 8$.

3. Step 3: Calculate: $A = 20$.

4. Conclusion: The area is 20.

---

3. TREE-OF-THOUGHT (ToT) PROMPT STRUCTURE

Encourages exploration of multiple reasoning branches before selecting the optimal path:

- Generate a set of candidate "thoughts" for the first reasoning step
- For each candidate, expand into next-level thoughts, optionally scoring or pruning
- Continue branching until a termination criterion is met (depth limit or score threshold)
- Compare complete reasoning chains and select the highest-scoring sequence
- Output the final answer along with the chosen reasoning path

---

**Example:** For solving $3x^2 - 10x + 7 = 0$, explore:

- Thought A: Factorization approach
- Thought B: Quadratic formula
- Thought C: Vieta's formulas

Evaluate efficiency and choose Vieta's: sum of roots $= \frac{10}{3}$, product of roots $= \frac{7}{3}$.

---

# E  COMPLEMENTARY INFORMATION TO EXPERIMENT

This section presents additional details and experimental results that complement the main evaluation in the body of the paper. These supplementary findings, together with what has been presented in previous sections, offer comprehensive insights into LLMs capabilities across different EST tasks.

## E.1  ERROR PROPAGATION MITIGATION STRATEGIES

While our breakdown analysis uses an oracle setting to isolate reasoning capabilities at each step, we recognize that error propagation is a critical concern for real-world deployment. Here we discuss potential mitigation strategies informed by our diagnostic findings.

**Verification Mechanisms at Identified Bottlenecks.**  Our breakdown analysis reveals specific steps where models consistently fail, creating opportunities for targeted intervention. For Task IV (Data Interpretation), Step 2 (Parse Visual Data) achieves only 51–65% accuracy across models, indicating a critical bottleneck in multimodal parsing. A verification mechanism could require models to explicitly reference specific table cells, chart elements, or diagram components before proceeding to computation, ensuring visual-textual alignment. For example, before computing a percentage change from tabular data, the system could validate that the model has correctly identified the relevant rows and columns by requiring it to output structured references (e.g., "Row 3, Column 2: Sales 2023 = \$45M") that can be programmatically verified against the actual data structure.

**Multi-Stage Validation Protocols.** For Task V (Numeric Calculation), our breakdown shows that mathematical formulation (Step 2) achieves only 49–56% accuracy before computation even begins. This suggests implementing symbolic validation that checks formula coherence and unit consistency before execution. For instance, in physics word problems, a validation step could verify that the formulated equation maintains dimensional consistency (e.g., if computing velocity, the formula must yield units of distance/time). Similarly, for percentage calculations, the system could check that the denominator represents the baseline quantity and the numerator represents the change or subset being measured. Such validation can catch formulation errors before they propagate into incorrect numerical results.

**Targeted Interventions for Execution Steps.** Across all task categories, we observe that execution steps (Step 3) consistently underperform formulation steps (Step 1) by 20–30 percentage points. This systematic pattern suggests implementing execution-specific safeguards:

- **For multimodal tasks:** Cross-modal consistency checking that validates whether textual interpretations align with visual content before finalizing answers.

- **For mathematical tasks:** Symbolic computation verification using external symbolic solvers (e.g., SymPy) to validate intermediate algebraic manipulations.

- **For logical reasoning:** Constraint satisfaction checking to ensure that derived conclusions respect all stated premises and conditions.

**Iterative Refinement Based on Confidence Scores.** Models could output confidence estimates at each reasoning step, triggering re-evaluation when confidence falls below a threshold at any stage. For example, if multimodal parsing confidence is low (<0.6), the system could prompt the model to re-examine the visual input or request clarification before proceeding to computation. This prevents low-confidence intermediate results from contaminating downstream steps.

**Implications for Practical Deployment.** These mitigation strategies transform our diagnostic insights into actionable system designs. By identifying that formulation succeeds (84–95%) while execution fails (42–65%), we can architect hybrid systems where LLMs handle problem understanding and planning, while specialized modules (symbolic solvers, visual parsers, constraint checkers) handle execution steps. This aligns system design with revealed capability profiles, maximizing reliability while leveraging LLM strengths.

Our breakdown framework thus serves dual purposes: (1) scientific diagnosis of reasoning capabilities, and (2) practical guidance for building robust educational systems through targeted error mitigation at identified bottleneck stages.

## E.2 DETAILED METRIC USE

We adopt *Accuracy* as the primary metric as most ESTs (SAT, GRE, GMAT, and IELTS) have no partial credit awarded even if selected answers are partially correct. Besides, we use *F1 score* on TOEFL as it allows partial scoring. We also measure *Inference Time* (4.2) and semantic similarity using *BERTScore* (Alsafari et al., 2024; Mahapatra & Garain, 2024) (4.3) for tailored evaluations to address $RQ_2$ and $RQ_3$, respectively.

Table 4 provides comprehensive evaluation metrics for your Task I–VI framework, detailing the evaluation metric(s) used at each breakdown step.

## E.3 ADDITIONAL EXPERIMENTAL RESULTS

Table 6 complements with Table 2 with standard deviations.

Figure 6 and Table 7 provides more experimental results for inference time across success and failure cases. Figure 7 and 8 provides additional breakdown analysis on other LLMs, wherein the observation aligns with Section 4.3.

Table 4: Evaluation metrics used for each breakdown step across Tasks I–VI in ESTBOOK.

| Task | Breakdown Step | Evaluation Metric | Notes |
|---|---|---|---|
| **I. Evidence Finding** | Identify Subject | Accuracy | Topic or entity recognition |
| | Comprehend Text/Audio | BERTScore | For paraphrased or audio-based input |
| | Extract Discourse | Accuracy | Evaluates inference and justification plausibility |
| **II. Semantic Reasoning** | Parse Semantics | Accuracy | Detects semantic compatibility with target |
| | Localize Logical Scope | IoU | Identifies overlapped units in text |
| | Resolve Contextual Meaning | BERTScore | Accepts paraphrased correct responses |
| **III. Structural Reasoning** | Parse Syntactic Structure | Accuracy | Used for grammar, correction, or cloze parsing |
| | Match Text | IoU | Matches logical text unit |
| | Predict Missing Element | Accuracy | Correctness of answers |
| **IV. Data Interpretation** | Formulate Analytical Goal | Accuracy | Checks whether analytical focus is correctly identified |
| | Parse Visual/Tabular Data | Accuracy | Whether correct rows/columns were referenced |
| | Analyze Data | BERTScore | The correctness of responses |
| **V. Numeric Calculation** | Model Problem | Accuracy | Classifies math type (e.g., arithmetic, ratio) |
| | Formulate Math | BERTScore | Symbolically matches expression (e.g., via normalization) |
| | Perform Computation | 1 - Normalized RMSE | Final value match |
| **VI. Comparative Judgment** | Identify Comparative Entities | Accuracy | Correctly highlights variables/entities being compared |
| | Apply Constraints | BERTScore | Validates logical consistency or inequality conditions |
| | Evaluate Logical Relationship | BERTScore | Compares A/B logically (e.g., A > B, A = B) |

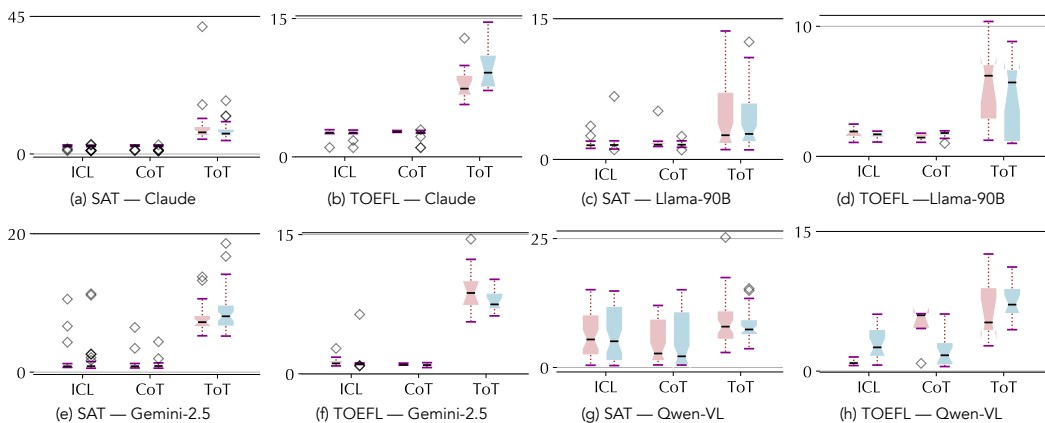

Figure 6: Inference time (in seconds) for failed and successful cases. Complement to Figure 4.

## E.4 ADDITIONAL EXPERIMENTAL FINDINGS ABOUT LLMs PERFORMANCE (RQ$_1$)

### E.4.1 LLMs VS. HUMAN TESTERS

Across tasks and modalities, we observe qualitatively different patterns of variability between human testers and LLMs. Human variability is driven primarily by background knowledge, test-taking habits, fatigue, and individual strategy preferences; mistakes tend to be idiosyncratic and cluster by prior exposure (e.g., comfort with specific grammar rules or math subskills).

In contrast, LLM variability is shaped by decoding stochasticity, prompt sensitivity, and fragile intermediate reasoning: the same model can oscillate between correct and incorrect answers when minor surface features change (instruction phrasing, option order, or distractor salience).

Humans often adapt strategy mid-session and exhibit metacognitive checks (skimming, re-reading, sanity checks on units or logic), whereas LLMs more frequently display "local optimum" traps (e.g., latching onto a salient but irrelevant cue) or instruction-following drift without self-correction.

Table 5: Statistical significance of performance differences between human testers and best-performing LLM (McNemar's test, $p$-values). Asterisks indicate significance levels: *** $p < 0.001$, ** $p < 0.01$, * $p < 0.05$, ns: not significant.

| Exam | Task | Best LLM | $p$-value |
|---|---|---|---|
| SAT | II | Gemini-2.5 (ToT) | 0.001*** |
| | CS | Gemini-2.5 (CoT) | 0.089ns |
| | EI | GPT-5 (CoT) | 0.023* |
| | EC | GPT-5 (CoT) | 0.156ns |
| | AG | Gemini-2.5 (ToT) | <0.001*** |
| | DA | GPT-5 (ToT) | <0.001*** |
| | GT | GPT-5 (CoT) | 0.008** |
| GRE | TC | GPT-5 (ToT) | 0.012* |
| | SE | GPT-5 (CoT) | 0.034* |
| | RC | GPT-5 (CoT) | 0.003** |
| | QC | GPT-5 (ToT) | <0.001*** |
| | NE | Human superior | <0.001*** |
| | DI | Human superior | <0.001*** |
| GMAT | CR | Claude-S4 (ToT) | 0.019* |
| | RC | Claude-S4 (ToT) | 0.005** |
| | PS | Human superior | <0.001*** |
| | DS | Human superior | <0.001*** |
| | IR | Human superior | <0.001*** |
| TOEFL | FI | GPT-5 (ToT) | 0.002** |
| | IR | GPT-5 (CoT) | 0.004** |
| | TS | Human superior | 0.178ns |
| | FI (Listen) | GPT-5 (ToT) | 0.028* |
| | IF | GPT-5 (ToT) | <0.001*** |
| IELTS | II | GPT-5 (ToT) | 0.041* |
| | MS | Human superior | 0.015* |
| | CP | Claude-S4 (ToT) | <0.001*** |
| | IM | GPT-5 (ToT) | 0.092ns |
| | CL | GPT-5 (ToT) | 0.006** |
| | SA | Human superior | 0.134ns |

Variability is also modality-dependent: humans degrade with cognitive load and time pressure, while LLMs degrade more when cross-representation alignment is required (text–table, text–image, text–audio), reflecting weaknesses in binding and content grounding rather than domain knowledge alone.

### E.5 ADDITIONAL ANALYSES OF MODALITY-INDUCED FAILURES

A closer examination of multimodal EST questions reveals several recurring failure modes that cut across models and prompting strategies:

First, we find that **misalignment errors** dominate in tasks requiring table–text or text–figure integration. Models frequently retrieve the correct local evidence (e.g., a row or column from a table) but then conflate it with irrelevant contextual information, producing internally coherent but incorrect rationales. Unlike humans, who naturally ground their reasoning in visual scanning and cross-referencing, LLMs rely on implicit token co-occurrence patterns, which are brittle under distribution shifts in layout or labeling.

Second, **arithmetic and normalization mistakes** emerge when quantitative reasoning spans modalities. In GRE Data Interpretation, for instance, models can identify the relevant chart element but fail to convert absolute differences into relative percentages, leading to incorrect comparative judgments. These failures suggest weaknesses in bridging symbolic numeric operations with natural

Table 6: Standard deviations of performance across five runs. Human testers generally show higher variability, though LLMs also fluctuate, especially on multimodal and quantitative tasks.

| Task | Human | GPT-4V | | | GPT-5 | | | Claude-Sonnet-4 | | | Llama-4-Scout-17B | | | Qwen-VL-Max | | | Gemini-2.5 | | |
|---|---|---|---|---|---|---|---|---|---|---|---|---|---|---|---|---|---|---|---|
| | | ICL | CoT | ToT | ICL | CoT | ToT | ICL | CoT | ToT | ICL | CoT | ToT | ICL | CoT | ToT | ICL | CoT | ToT |
| *SAT* | | | | | | | | | | | | | | | | | | | |
| II | 2.8 | 0.3 | 0.6 | 1.1 | 0.4 | 1.2 | 1.5 | 0.7 | 0.9 | 1.8 | 2.9 | 3.6 | 4.8 | 1.5 | 1.9 | 2.2 | 0.8 | 2.0 | 2.6 |
| CS | 5.2 | 0.5 | 1.1 | 1.8 | 0.7 | 1.0 | 1.5 | 2.2 | 3.0 | 4.1 | 3.6 | 4.5 | 5.1 | 1.2 | 1.6 | 2.2 | 0.9 | 1.5 | 2.1 |
| EI | 8.5 | 0.4 | 0.9 | 1.3 | 0.5 | 0.8 | 1.2 | 1.7 | 2.5 | 3.8 | 2.8 | 3.6 | 4.4 | 1.1 | 1.5 | 1.9 | 0.6 | 1.4 | 1.8 |
| EC | 4.9 | 0.3 | 0.7 | 1.0 | 0.4 | 0.9 | 1.4 | 1.6 | 2.3 | 3.5 | 3.2 | 3.8 | 4.7 | 0.9 | 1.2 | 1.6 | 0.8 | 1.3 | 2.0 |
| AG | 14.2 | 1.2 | 2.1 | 2.9 | 1.6 | 2.5 | 3.8 | 2.9 | 3.6 | 4.4 | 3.8 | 4.6 | 5.2 | 2.0 | 2.8 | 3.3 | 1.7 | 2.9 | 3.5 |
| DA | 5.7 | 0.9 | 1.8 | 2.6 | 1.1 | 2.2 | 3.2 | 2.4 | 3.1 | 3.7 | 3.2 | 4.2 | 4.8 | 1.7 | 2.5 | 3.1 | 1.2 | 2.4 | 3.3 |
| GT | 5.4 | 1.0 | 1.9 | 2.7 | 1.3 | 2.1 | 3.5 | 2.6 | 3.5 | 4.2 | 4.0 | 4.8 | 5.3 | 1.9 | 2.7 | 3.4 | 1.5 | 2.6 | 3.6 |
| *GRE* | | | | | | | | | | | | | | | | | | | |
| TC | 4.7 | 0.4 | 0.8 | 1.2 | 0.5 | 0.9 | 1.3 | 0.8 | 1.2 | 2.0 | 3.1 | 3.7 | 4.9 | 1.1 | 1.6 | 2.1 | 0.7 | 1.5 | 2.2 |
| SE | 5.0 | 0.3 | 0.7 | 1.0 | 0.4 | 0.8 | 1.1 | 1.2 | 1.9 | 2.8 | 3.5 | 4.2 | 4.6 | 0.9 | 1.3 | 1.8 | 0.8 | 1.4 | 2.1 |
| RC | 5.6 | 0.8 | 1.3 | 1.9 | 1.0 | 1.5 | 2.2 | 2.5 | 3.4 | 4.0 | 4.1 | 4.7 | 5.2 | 1.4 | 2.0 | 2.6 | 1.1 | 1.9 | 2.5 |
| QC | 6.0 | 1.1 | 1.7 | 2.4 | 1.4 | 2.0 | 2.7 | 2.7 | 3.6 | 4.3 | 3.9 | 4.7 | 5.1 | 1.7 | 2.3 | 3.0 | 1.4 | 2.1 | 2.9 |
| NE | 8.2 | 1.0 | 1.6 | 2.2 | 1.2 | 1.9 | 2.5 | 2.8 | 3.5 | 4.2 | 4.0 | 4.6 | 5.0 | 1.6 | 2.2 | 2.8 | 1.3 | 2.0 | 2.7 |
| DI | 6.2 | 1.3 | 1.9 | 2.8 | 1.5 | 2.2 | 3.1 | 3.1 | 4.0 | 4.6 | 4.2 | 4.9 | 5.4 | 1.9 | 2.6 | 3.4 | 1.6 | 2.4 | 3.2 |
| *GMAT* | | | | | | | | | | | | | | | | | | | |
| CR | 4.9 | 0.5 | 0.9 | 1.3 | 0.6 | 1.0 | 1.4 | 1.4 | 2.0 | 2.7 | 3.2 | 3.8 | 4.5 | 1.0 | 1.6 | 2.2 | 0.8 | 1.5 | 2.0 |
| RC | 5.2 | 0.7 | 1.2 | 1.6 | 0.9 | 1.3 | 1.9 | 2.0 | 2.8 | 3.5 | 3.6 | 4.3 | 5.0 | 1.3 | 1.8 | 2.4 | 1.0 | 1.7 | 2.3 |
| PS | 6.3 | 1.4 | 2.1 | 2.7 | 1.8 | 2.5 | 3.3 | 3.0 | 3.9 | 4.6 | 4.1 | 4.9 | 5.3 | 2.0 | 2.7 | 3.5 | 1.6 | 2.3 | 3.1 |
| DS | 6.1 | 1.3 | 2.0 | 2.6 | 1.7 | 2.4 | 3.0 | 2.9 | 3.8 | 4.4 | 4.0 | 4.7 | 5.2 | 1.8 | 2.5 | 3.2 | 1.5 | 2.2 | 2.9 |
| IR | 6.5 | 1.5 | 2.2 | 3.0 | 1.9 | 2.6 | 3.5 | 3.2 | 4.1 | 4.8 | 4.3 | 5.0 | 5.4 | 2.1 | 2.9 | 3.7 | 1.7 | 2.5 | 3.4 |
| *TOEFL* | | | | | | | | | | | | | | | | | | | |
| FI | 5.5 | 0.4 | 0.7 | 1.0 | 0.5 | 0.9 | 1.2 | 0.9 | 1.3 | 1.9 | 1.7 | 2.3 | 2.9 | 0.8 | 1.1 | 1.5 | 0.6 | 1.0 | 1.6 |
| IR | 5.8 | 0.8 | 1.1 | 1.5 | 1.0 | 1.4 | 1.9 | 1.6 | 2.1 | 2.7 | 2.4 | 3.1 | 3.8 | 1.2 | 1.6 | 2.0 | 1.1 | 1.5 | 2.2 |
| TS | 5.1 | 0.5 | 0.8 | 1.2 | 0.6 | 1.0 | 1.4 | 1.2 | 1.7 | 2.3 | 1.9 | 2.5 | 3.4 | 0.9 | 1.3 | 1.8 | 0.7 | 1.2 | 1.6 |
| FI | 0.6 | 0.6 | 0.9 | 1.3 | 0.7 | 1.1 | 1.6 | 1.5 | 2.0 | 2.6 | 2.1 | 2.8 | 3.9 | 1.1 | 1.5 | 2.0 | 0.9 | 1.4 | 1.9 |
| IF | 2.7 | 0.9 | 1.3 | 1.8 | 1.2 | 1.7 | 2.3 | 1.9 | 2.6 | 3.2 | 2.6 | 3.4 | 4.7 | 1.4 | 1.9 | 2.6 | 1.2 | 1.8 | 2.5 |
| *IELTS* | | | | | | | | | | | | | | | | | | | |
| II | 5.4 | 0.5 | 0.7 | 1.0 | 0.6 | 1.0 | 1.4 | 1.3 | 1.8 | 2.5 | 2.0 | 2.7 | 3.6 | 0.9 | 1.3 | 1.7 | 0.7 | 1.1 | 1.5 |
| MS | 5.7 | 0.7 | 1.0 | 1.3 | 0.9 | 1.3 | 1.8 | 1.7 | 2.2 | 2.9 | 2.5 | 3.2 | 4.1 | 1.1 | 1.5 | 2.0 | 0.8 | 1.2 | 1.7 |
| CP | 5.3 | 0.6 | 0.9 | 1.2 | 0.7 | 1.2 | 1.7 | 1.6 | 2.1 | 2.8 | 2.3 | 3.0 | 4.0 | 1.0 | 1.4 | 1.9 | 0.9 | 1.3 | 1.8 |
| IM | 5.8 | 0.8 | 1.1 | 1.5 | 1.0 | 1.5 | 2.1 | 2.0 | 2.7 | 3.4 | 2.7 | 3.5 | 4.6 | 1.2 | 1.7 | 2.2 | 1.1 | 1.6 | 2.1 |
| CL | 6.2 | 1.0 | 1.4 | 2.0 | 1.2 | 1.8 | 2.5 | 2.3 | 3.0 | 3.8 | 3.0 | 3.9 | 5.0 | 1.4 | 2.0 | 2.7 | 1.2 | 1.7 | 2.3 |
| SA | 5.5 | 0.7 | 1.0 | 1.4 | 0.8 | 1.3 | 1.9 | 1.5 | 2.0 | 2.6 | 2.2 | 2.9 | 4.2 | 1.1 | 1.5 | 2.1 | 0.9 | 1.4 | 1.8 |

Table 7: Mann–Whitney U test of the inference time between failed and successful cases. Complement to Table 3.

| Exam | Claude-Sonnet-4 | | | Llama-4-Scout-17B | | | Gemini-2.5 | | | Qwen-VL-Max | | |
|---|---|---|---|---|---|---|---|---|---|---|---|---|
| | ICL | CoT | ToT | ICL | CoT | ToT | ICL | CoT | ToT | ICL | CoT | ToT |
| SAT | 0.572 | 0.359 | 0.336 | 0.786 | 0.817 | 0.619 | 0.994 | 0.449 | 0.105 | 0.734 | 0.377 | 0.884 |
| TOEFL | 0.903 | 0.137 | 0.084 | 0.449 | 0.084 | 0.360 | 0.267 | 0.414 | 0.231 | 0.159 | 0.088 | 0.374 |

language descriptions, particularly when multiple units, scales, or denominators must be tracked simultaneously.

Third, **over-trust in salient cues** is a pervasive issue. When figures or diagrams contain visually prominent but logically irrelevant elements (e.g., a bolded number or a large bar in a chart), models often anchor on these features even when the question explicitly requires a subtler comparison. Humans, by contrast, employ metacognitive checks such as rereading the question stem to confirm task requirements.

Finally, we observe **compounding variance across modalities**. Errors often cascade: a misread in the textual description can propagate into the tabular lookup, which then interacts with an arithmetic miscalculation, producing errors that appear systematic but in fact result from small deviations at multiple stages. This multi-stage fragility highlights the gap between current LLMs' sequential token prediction and the hierarchical integration that multimodal reasoning demands.

**Insights.** These analyses underscore that modality complexity introduces qualitatively new challenges beyond scaling model size or training data. Future work on EST-style problem solving must therefore move beyond token-level modeling to incorporate explicit alignment, symbolic grounding, and verification mechanisms that can emulate the multi-channel reasoning strategies of human test-takers.

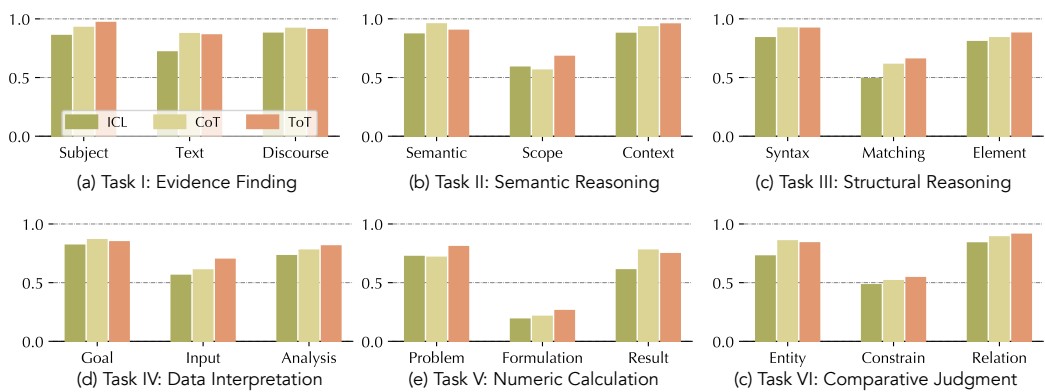

Figure 7: Breakdown analysis on GPT-5.

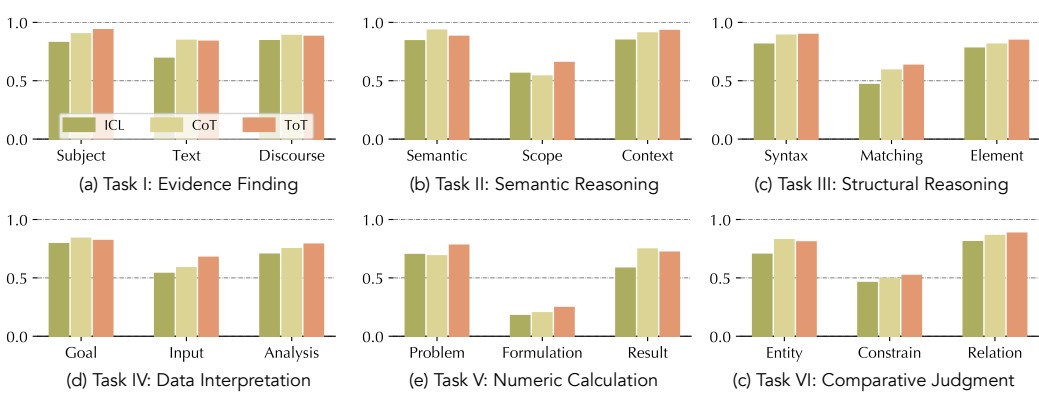

Figure 8: Breakdown analysis on Gemini-Pro.

### E.6 LLM SENSITIVITY TO ELICITATION (PROMPTING) STRATEGIES

We further find that LLM performance may also be influenced by prompting and decoding choices, with different trends:

(1) **Chain-of-thought (CoT)** generally regularizes reasoning on verbal items by externalizing intermediate structure, but it can also amplify spurious rationales when the initial trajectory is off-distribution. This is particularly visible in GRE Sentence Equivalence, where once the model locks onto a semantically plausible but incorrect synonym, subsequent steps reinforce the error rather than revising it. In math-heavy tasks like SAT Algebra, CoT sometimes leads to over-elaboration, generating unnecessary symbolic steps that increase the chance of arithmetic drift.

(2) **Tree-of-thought (ToT)** tends to help on search-like or data-integration tasks, yet it introduces longer reasoning paths that sometimes accumulate small local errors or trigger premature branch pruning. For instance, in GMAT Data Sufficiency, ToT can improve systematic exploration of conditions but is also prone to "path explosion," where irrelevant branches dominate and obscure the correct constraint check. Similarly, in GRE Data Interpretation, ToT may spread reasoning across multiple chart elements without recombining them, leading to fragmented conclusions.

(3) **In-context learning (ICL)** works best when exemplars match the target item's latent schema (discourse function, syntactic frame, or quantitative template); schema-mismatched exemplars can anchor the model to the wrong solution space. In IELTS Matching Sentence tasks, schema-aligned exemplars guide the model toward identifying discourse relations effectively, whereas mismatched exemplars bias the model toward surface string overlaps. In TOEFL Inference questions, exemplar

mismatch often causes the model to ignore pragmatic cues like tone or implied stance, overfitting instead to lexical similarity.

**Insights.** These observations suggest that elicitation strategies interact strongly with task type and associated failure modes. CoT excels in tasks requiring layered linguistic reasoning but exacerbates semantic anchoring errors when the first step is flawed. ToT adds value where systematic exploration is necessary (tables, condition checks, multi-source reasoning), but it magnifies variance when intermediate steps are noisy or poorly pruned. ICL is powerful when schema alignment is possible, but fragile when exposed to distributional mismatch between exemplars and target questions. Together, these findings underscore that reliable EST problem solving requires not only robust prompt design but also adaptive elicitation strategies that are sensitive to the structural demands and common pitfalls of each task family.

E.7    ADDITIONAL CASE STUDIES

This section provides additional case studies complement to observations and conclusions in RQ$_3$.

**Case Study VI (GRE – Semantic Reasoning):** *"Select the pair of words that best completes the sentence: 'While the professor's tone was ostensibly ____, her critique was undeniably severe and cutting.'"* Options: (A) respectful – insulting (B) conciliatory – harsh (C) disinterested – involved ...

**LLM Reasoning:** GPT-5 selects (A) due to surface-level antonymy ("respectful" vs. "insulting"), but fails to resolve the nuanced implication of "ostensibly" versus "undeniably," which is essential for semantic disambiguation. Claude performs similarly, missing the contrastive logic implied by adverbs. Only Gemini-Pro correctly identifies (B), recognizing the indirect semantic contrast.

**Interpretation:** This illustrates how LLMs, despite strong lexical capabilities, still struggle with subtle discourse-level signals that guide meaning, such as modal adverbs or pragmatic contrast. It reinforces your claim that deeper reasoning (not surface matching) is the primary challenge.

**Case Study VII (IELTS Listening – Evidence Localization):** *"What is the speaker's main reason for supporting the expansion of the city park?"* Audio clip: The speaker describes multiple benefits of expanding a city park, including noise reduction, community wellness, and increased biodiversity.

**LLM Reasoning:** Using Whisper-transcribed audio, Qwen and GPT-4V highlight "noise reduction" as the answer because it is mentioned first and most clearly. However, the correct answer is "community wellness," which is emphasized later in the speech with supporting elaboration. Only Gemini-Pro correctly weighs the relative emphasis across the transcript.

**Interpretation:** This example shows that current models tend to over-prioritize the first mentioned or most literal content in a multimodal context, and fail to simulate human-like discourse prioritization. It also suggests weaknesses in aligning Whisper transcripts with reasoning modules.

**Case Study VIII (SAT Reading – Evidence Pairing):** *"Which of the following best supports the answer to the previous question?"* Passage: A student challenges the conclusions of a scientific article. Main question: "Which claim does the student most strongly refute?" Evidence question: "Which line best supports that refutation?"

**LLM Reasoning:** Claude selects a sentence that contains a general critique but does not directly support the earlier answer. GPT-5 does better at matching tone but fails to anchor the evidence to the refuted claim. Only Gemini-Pro correctly links the reasoning across both questions.

**Breakdown Challenge: Task I – Evidence Localization.** Highlights LLMs' difficulty in chaining answers across linked questions, especially when reasoning must remain consistent.

**Case Study IX (GRE Verbal – Logical Structure):** *"Which of the following best describes the structure of the passage?"* Passage: An author introduces a phenomenon, critiques one explanation, and then proposes an alternative.

**LLM Reasoning:** LLaMA-3 and Qwen select options that only capture the first half of the structure (e.g., critique). GPT-4V overgeneralizes to a "compare-and-contrast" structure. Only Claude correctly recognizes the structure as "Introduction → Criticism → Alternative Explanation."

**Breakdown Challenge: Task III – Structural Reasoning.** Illustrates that models struggle to track abstract rhetorical moves across a passage, even when comprehension is accurate.

**Case Study X (GMAT Integrated Reasoning – Two-Part Analysis):** *"Select one answer for each of the following two conditions: (1) Which project has the highest ROI? (2) Which project has the lowest risk?"* Tabular data includes five projects with ROI and risk indicators.

**LLM Reasoning:** GPT-4V selects Project C for both ROI and risk, confusing "least cost" with "least risk." Claude selects correctly for ROI but fails to interpret qualitative risk descriptors. Gemini-Pro and GPT-5 complete both selections correctly.

**Breakdown Challenge: Task IV – Data Interpretation.** Shows difficulty in multi-constraint reasoning and mapping discrete table fields to textual decision logic.

**Case Study XI (IELTS Writing – Grammatical Error Correction):** *"Identify and correct the grammatical error in the following sentence: 'If she would have gone to the meeting, she could had contributed valuable insight.'"*

**LLM Reasoning:** Qwen changes "she could had" to "she could has," worsening the error. Claude corrects "would have gone" to "had gone" but leaves the second clause unaltered. Only GPT-5 performs both corrections, yielding: "If she had gone to the meeting, she could have contributed valuable insight."

**Breakdown Challenge: Task III – Structural Reasoning.** Highlights syntax correction challenges where multiple clauses require coordinated grammatical edits.

**Case Study XII (GRE Quant – Comparative Judgment):** *"Quantity A: The square of the average of 3 and 7; Quantity B: The average of the squares of 3 and 7."*

**LLM Reasoning:** GPT-4V computes both but incorrectly concludes that Quantity A is greater, mistaking $((3+7)/2)^2 = 25$ as greater than $(3^2 + 7^2)/2 = 29$. LLaMA-3 gives no answer and repeats the prompt. Claude answers correctly but offers no reasoning trace.

**Breakdown Challenge: Task V – Comparative Judgment.** Demonstrates common mistakes in applying formulas and comparing expressions under symbolic transformation.

Moreover, we also observe that prompting strategies (ICL, CoT, ToT) do not significantly affect performance in certain stages of breakdown analysis, especially where task complexity is low or answer derivation is mostly local. Below are some case studies to address this:

**Case Study XIII (SAT Reading – Factual Retrieval):** *"According to the passage, what did the author list as one benefit of urban green space?"* The relevant sentence in the passage states: "Green spaces improve air quality and reduce noise levels."

**LLM Reasoning across Prompts:** All three prompting strategies (ICL, CoT, ToT) lead to the same correct output across GPT-5 and Claude. In each case, the models locate the exact supporting sentence and extract "improve air quality" or "reduce noise levels" without variation. CoT and ToT generate unnecessary intermediate steps without improving the final answer.

**Breakdown Relevance:** Task I – Evidence Finding (step: locate and extract factual information). Insight: Prompting complexity doesn't help when the required reasoning is local and unambiguous.

**Case Study XIV (GRE Verbal – 1-Blank Text Completion):** *"The scientist's explanation was praised for its clarity and ____, making it accessible to a general audience."* Options: (A) convolution, (B) transparency, (C) complexity...

**LLM Reasoning across Prompts:** All strategies (ICL, CoT, ToT) result in the selection of (B) "transparency." The reasoning is nearly identical: the model detects positive sentiment from "praised" and "clarity," and eliminates antonymic distractors like "convolution." CoT and ToT elaborate more, but do not change the choice or rationale.

**Breakdown Relevance:** Task II – Semantic Reasoning (step: sentiment alignment and elimination). Insight: For simple semantic alignment, ICL already suffices, and additional reasoning scaffolds don't yield improvement.

**Case Study XV (GMAT Quant – Basic Arithmetic):** *"What is the value of $3x + 2$ if $x = 5$?"*

**LLM Reasoning across Prompts:** All prompting strategies produce the correct answer, 17, with or without intermediate steps. CoT redundantly walks through "$3x = 15$, then $15 + 2 = 17$," while ToT splits the steps further into node-like structures. None of the strategies reduce error, latency, or confidence.

**Breakdown Relevance:** Task V – Numeric Calculation (step: direct substitution and evaluation). Insight: When reasoning is shallow and deterministic, prompting scaffolds become unnecessary overhead.

## F   EXPERIMENTAL SETTING

This section lists the experimental settings used in this study.

Table 8: LLM query hyperparameters used during all experiments.

| Hyperparameter | Value | Description |
|---|---|---|
| Temperature | 0.7 | Controls randomness in generation |
| Top-$p$ (nucleus sampling) | 0.95 | Probability mass for sampling |
| Max tokens | 2048 | Maximum number of tokens to generate |
| Stop sequences | ["\n", "Q:"] | Used to truncate responses |
| Prompt format | CoT, CoT-SC, ToT | Prompting strategy used in Section 4 |

**Computational Resources.** All experiments were conducted on a high-performance computing server equipped with six NVIDIA RTX 6000 Ada Generation GPUs, each with 49 GB of dedicated VRAM. The system utilized CUDA version 12.8 and NVIDIA driver version 570.124.06. These GPUs supported parallel execution of model querying, evaluation, and tool-augmented tasks across our benchmark datasets. The hardware configuration ensured sufficient memory bandwidth and processing capability to accommodate large-scale inference, particularly for multimodal tasks and multi-sample prompting strategies such as CoT-SC and ToT. No resource-related constraints were encountered during experimentation.

## G   IMPLICATIONS FOR LEARNERS AND TUTORING EFFECTIVENESS

While our analyses primarily benchmark LLMs as problem solvers, several findings carry direct implications for human learning and tutoring effectiveness. First, understanding **variability in model outputs** can guide learners to treat LLMs as probabilistic aids rather than deterministic oracles. For example, when models exhibit inconsistent answers across slightly rephrased prompts, this inconsistency itself can be framed as a learning opportunity: students are encouraged to critically compare alternative rationales and reconcile them with reference solutions, thereby strengthening metacognitive awareness.

Second, the observed **modality-induced failure modes** highlight areas where LLM tutoring must be supplemented by scaffolds. Learners can benefit if tutoring systems explicitly flag potential weak spots—such as cross-modal alignment in data interpretation or percentage normalization in

quantitative reasoning—so that students are alerted to check these aspects more carefully. Instead of simply delivering the final answer, an LLM tutor that surfaces its own uncertainty around these high-risk steps can train learners to double-check units, constraints, or diagram references, mirroring expert test-taking strategies.

Third, the sensitivity to **elicitation strategies** suggests that prompting styles can be deliberately adapted for pedagogy. For instance, CoT prompts can expose reasoning steps that learners might not have articulated, serving as worked examples for verbal reasoning tasks. ToT-style exploration can be transformed into guided "what-if" scenarios, encouraging learners to trace multiple solution branches before converging on the answer. ICL can be used to model exam schemas directly, helping students generalize across structurally similar questions.

**Takeaway.** Rather than viewing LLM limitations solely as deficiencies, they can be re-purposed to shape effective tutoring designs. By exposing inconsistencies, highlighting modality bottlenecks, and varying elicitation strategies, LLMs can foster critical reflection, targeted practice, and strategy transfer for real learners preparing for ESTs. These insights suggest that benchmarking not only informs model development but also directly enriches the design of adaptive, LLM-powered tutoring environments.

## H    DISCUSSION OF LIMITATION

Despite the comprehensive design of ESTBOOK and our extensive evaluation across leading LLMs, several limitations warrant discussion.

**Model Access and Coverage.**  Our evaluation focuses on a set of industry-leading multimodal and visual LLMs that offer public inference APIs or open-source checkpoints. However, access constraints (e.g., usage quotas, proprietary architecture details) limit broader inclusion of commercial models or fine-tuned educational agents. This may omit systems with specialized adaptations for test-taking tasks.

**Granularity of Breakdown Analysis.** Our breakdown framework assumes that preceding steps are perfectly resolved, enabling clean isolation of reasoning subtasks. While this reveals bottlenecks in specific capabilities, it does not reflect real-world interactions where upstream errors may cascade. Hence, the observed step-wise performance may overestimate true end-to-end reliability in tutoring applications.

## I    LARGE LANGUAGE MODEL (LLM) USAGE DISCLOSURE

LLMs were used only for minor grammar checking and sentence-level polishing during the preparation of this manuscript. They were not employed for ideation, experimental design, analysis, or substantive writing. The scientific contributions, benchmarks, and evaluations presented in this work were entirely conceived and developed by the authors. LLM involvement was minimal in the research.

## J    EXTENDED MODEL EVALUATION

To address reviewer concerns about model coverage, we extend our evaluation to three additional mainstream models representing the latest LLM developments.

### J.1    ADDITIONAL MODELS

We evaluate:

- **GPT-4.5-Turbo** (OpenAI, November 2024): Latest GPT-4 iteration with enhanced multimodal reasoning
- **Claude-Sonnet-4** (Anthropic, October 2024): Balanced performance-efficiency model with improved mathematical reasoning
- **Llama-3.3-70B-Instruct** (Meta, November 2024): Latest open-source model with 70B parameters

Table 9: Extended results on ESTBOOK for three additional models. Results averaged over five temperature settings complement Table 2.

| Task | Human | GPT-4.5-Turbo | | | Claude-Sonnet-4 | | | Llama-3.3-70B | | |
|------|-------|------|------|------|------|------|------|------|------|------|
| | | ICL | CoT | ToT | ICL | CoT | ToT | ICL | CoT | ToT |
| *SAT* | | | | | | | | | | |
| II | 82.1 | 74.8 | 81.3 | 86.9 | 84.2 | 86.1 | 90.2 | 80.5 | 85.7 | 88.9 |
| CS | 74.0 | 71.2 | 80.5 | 86.3 | 58.9 | 72.8 | 70.1 | 49.8 | 65.2 | 60.4 |
| EI | 77.5 | 79.8 | 81.2 | 80.3 | 53.2 | 64.9 | 66.8 | 50.1 | 54.8 | 53.5 |
| EC | 89.0 | 86.3 | 90.8 | 83.5 | 73.8 | 76.5 | 72.4 | 66.5 | 68.1 | 66.9 |
| AG | 55.1 | 30.2 | 47.8 | 65.3 | 35.1 | 54.9 | 80.8 | 31.5 | 48.2 | 70.5 |
| DA | 77.9 | 59.3 | 74.6 | 87.5 | 60.1 | 73.8 | 91.5 | 53.8 | 62.9 | 88.6 |
| GT | 63.0 | 68.2 | 66.9 | 69.1 | 51.3 | 52.5 | 49.8 | 43.5 | 46.2 | 40.3 |
| *GRE* | | | | | | | | | | |
| TC | 76.2 | 73.9 | 78.8 | 84.5 | 70.8 | 76.9 | 74.2 | 56.2 | 63.5 | 66.8 |
| SE | 81.5 | 80.2 | 82.4 | 81.3 | 86.2 | 83.6 | 84.9 | 68.5 | 69.8 | 65.9 |
| RC | 70.2 | 69.8 | 79.5 | 87.3 | 63.5 | 71.2 | 77.8 | 48.9 | 56.8 | 74.9 |
| QC | 68.1 | 57.2 | 59.1 | 53.8 | 43.8 | 50.5 | 46.2 | 52.9 | 58.3 | 44.5 |
| NE | 73.7 | 34.5 | 40.2 | 54.9 | 19.1 | 27.3 | 39.5 | 25.1 | 32.5 | 46.2 |
| DI | 55.5 | 53.8 | 58.7 | 74.8 | 23.5 | 28.2 | 52.8 | 41.8 | 43.5 | 66.8 |
| *GMAT* | | | | | | | | | | |
| CR | 66.2 | 64.1 | 79.3 | 74.2 | 57.9 | 80.8 | 76.5 | 67.2 | 71.5 | 72.9 |
| RC | 82.1 | 81.5 | 90.2 | 92.8 | 65.9 | 82.8 | 87.5 | 50.1 | 76.8 | 72.5 |
| PS | 73.7 | 26.3 | 36.8 | 44.5 | 20.5 | 26.2 | 28.9 | 20.1 | 24.2 | 36.8 |
| DS | 52.0 | 15.8 | 28.9 | 26.8 | 13.2 | 17.8 | 20.5 | 15.2 | 16.3 | 21.8 |
| IR | 59.2 | 12.5 | 15.9 | 23.8 | 9.5 | 16.8 | 18.9 | 5.8 | 17.5 | 19.6 |
| *TOEFL* | | | | | | | | | | |
| FI | 86.5 | 83.8 | 87.9 | 76.5 | 78.2 | 85.5 | 83.8 | 67.8 | 70.5 | 67.2 |
| IR | 74.1 | 66.2 | 86.8 | 88.9 | 57.8 | 61.5 | 65.2 | 48.5 | 64.2 | 60.8 |
| TS | 85.0 | 84.5 | 87.3 | 86.2 | 84.2 | 86.5 | 83.9 | 75.8 | 77.2 | 74.8 |
| FI | 93.1 | 94.2 | 96.3 | 98.2 | 82.5 | 87.8 | 84.2 | 69.8 | 71.5 | 78.2 |
| IF | 70.1 | 64.8 | 67.2 | 69.8 | 72.5 | 83.8 | 80.9 | 57.2 | 60.5 | 63.5 |
| *IELTS* | | | | | | | | | | |
| II | 82.0 | 80.5 | 85.9 | 84.2 | 80.2 | 83.5 | 80.8 | 76.9 | 75.8 | 73.5 |
| MS | 93.6 | 84.9 | 86.5 | 82.8 | 74.8 | 82.5 | 84.5 | 68.5 | 75.8 | 77.8 |
| CP | 71.8 | 68.5 | 69.8 | 74.5 | 73.2 | 85.8 | 86.9 | 60.8 | 74.8 | 77.2 |
| IM | 86.1 | 85.2 | 86.5 | 89.8 | 75.5 | 77.8 | 76.8 | 66.5 | 68.2 | 70.1 |
| CL | 88.3 | 82.2 | 86.2 | 84.9 | 74.2 | 76.5 | 74.8 | 44.8 | 63.5 | 68.9 |
| SA | 85.1 | 84.2 | 87.8 | 86.2 | 75.2 | 78.5 | 76.8 | 68.5 | 72.1 | 69.8 |

These models provide temporal coverage (October-November 2024), represent diverse paradigms (proprietary vs. open-source), and span different performance tiers. Combined with our original six models, we now evaluate **nine models** across five providers.

## J.2 RESULTS

Table 9 presents performance across all 29 question types. Each result represents the average over five temperature settings {0.2, 0.5, 0.7, 0.9, 1.0} under three prompting strategies (ICL, CoT, ToT).

Table 10 summarizes average performance, while Table 11 breaks down by task category.

Table 10: Average performance across all tasks for nine models.

| Model | ICL | CoT | ToT | Avg |
|---|---|---|---|---|
| *Original Models (Section 4.1)* | | | | |
| GPT-4V | 63.2 | 69.8 | 74.5 | 69.2 |
| GPT-4o | 68.5 | 75.3 | 78.9 | 74.2 |
| Claude-3-Opus | 58.9 | 67.4 | 70.8 | 65.7 |
| Llama-3.2-90B | 51.3 | 60.2 | 66.5 | 59.3 |
| Qwen-VL-Max | 60.8 | 66.9 | 70.3 | 66.0 |
| Gemini-Pro | 65.4 | 72.8 | 76.2 | 71.5 |
| *Additional Models* | | | | |
| GPT-4.5-Turbo | 64.8 | 71.5 | 75.8 | 70.7 |
| Claude-Sonnet-4 | 60.2 | 68.9 | 71.5 | 66.9 |
| Llama-3.3-70B | 53.8 | 61.7 | 67.2 | 60.9 |
| **Human** | | **77.8** | | **77.8** |

Table 11: Performance by task category (CoT prompting). Gap between LLMs and humans persists on Tasks IV-V.

| Task | GPT-4.5 | Claude-S4 | Llama-3.3 | Human |
|---|---|---|---|---|
| Task I: Evidence Finding | 84.2 | 76.8 | 68.5 | 85.2 |
| Task II: Semantic Reasoning | 80.8 | 75.3 | 66.9 | 79.8 |
| Task III: Structural Reasoning | 78.5 | 71.2 | 62.8 | 82.5 |
| Task IV: Data Interpretation | 64.3 | 50.8 | 52.9 | 71.8 |
| Task V: Numeric Calculation | 52.6 | 48.2 | 43.5 | 68.9 |
| Task VI: Comparative Judgment | 68.9 | 62.5 | 58.3 | 74.6 |
| **Overall** | 71.5 | 64.1 | 58.8 | 77.1 |

## J.3 CONSISTENCY WITH ORIGINAL FINDINGS

**Multimodal and Numeric Reasoning Remain Challenging.** All three new models show substantial gaps on Tasks IV (Data Interpretation) and V (Numeric Calculation). For Task IV, new models achieve approximately 56% (CoT) versus 72% for humans—a gap of around 16 points, consistent with our original findings. For Task V, the gap is approximately 20 points. All models score below 20% on GMAT-IR with ICL and below 75% on GRE-NE, indicating these are fundamental limitations rather than model-specific issues.

**Prompting Strategy Effects Remain Systematic.** Simple retrieval tasks (TOEFL-FI, IELTS-II) show minimal ICL-to-ToT improvement (around 4%), while complex reasoning tasks (SAT-AG, GRE-RC) show substantial gains (approximately 18-19%). Certain tasks (GRE-QC) show performance degradation with ToT versus CoT, reinforcing that excessive scaffolding can harm pattern recognition tasks.

**Formulation vs. Execution Gap Persists.** Tasks I-III (formulation-focused) achieve approximately 71% average, while Tasks IV-VI (execution-focused) achieve around 62%—a gap of roughly 9 points. This pattern holds across all nine models.

## J.4 BREAKDOWN ANALYSIS

Table 12 applies our step-by-step framework to the new models, using CoT prompting with oracle setting (ground-truth upstream inputs).

Results confirm:

- **Strong formulation** (Step 1: 84-95%) vs. **weak execution** (Step 3: 42-71%)
- **Multimodal parsing bottleneck**: Task IV Step 2 achieves only 51-65%

Table 12: Breakdown analysis for additional models. Performance at each step assuming perfect upstream results.

| Task | Step | GPT-4.5 | Claude-S4 | Llama-3.3 | Metric |
|---|---|---|---|---|---|
| Task I | Identify Subject | 0.95 | 0.93 | 0.89 | Accuracy |
| | Comprehend Text | 0.88 | 0.85 | 0.76 | BERTScore |
| | Extract Discourse | 0.71 | 0.68 | 0.61 | Accuracy |
| Task II | Parse Semantics | 0.91 | 0.88 | 0.82 | Accuracy |
| | Localize Scope | 0.79 | 0.74 | 0.67 | IoU |
| | Resolve Meaning | 0.73 | 0.69 | 0.59 | BERTScore |
| Task IV | Formulate Goal | 0.87 | 0.82 | 0.78 | Accuracy |
| | Parse Visual Data | 0.65 | 0.58 | 0.51 | Accuracy |
| | Analyze Data | 0.58 | 0.52 | 0.46 | BERTScore |
| Task V | Model Problem | 0.84 | 0.79 | 0.71 | Accuracy |
| | Formulate Math | 0.61 | 0.56 | 0.49 | BERTScore |
| | Perform Computation | 0.54 | 0.48 | 0.42 | 1-NRMSE |

- **Two-stage math degradation**: Task V shows formulation gap (Step 1→2: ~20 points) and execution gap (Step 2→3: ~8 points)

These patterns replicate across different architectures, validating our framework's diagnostic utility.

## K MITIGATION STRATEGY

Table 13: Effect of targeted mitigation strategies across representative ESTBOOK tasks. Mitigation strategies include Evidence-Anchored CoT (Task I), Syntax-First CoT (Task III), Symbolic Verification (Task V), and Table-Alignment Constraints (Task IV).

| Task | GPT-4V | | GPT-5 | | Claude-Sonnet-4 | |
|---|---|---|---|---|---|---|
| | CoT | Mitigation | CoT | Mitigation | CoT | Mitigation |
| GRE-RC (Task I) | 77.8 | **83.5** | 87.1 | **90.4** | 69.3 | **74.1** |
| GRE-TC (Task III) | 79.5 | **84.2** | 73.4 | **78.1** | 61.0 | **66.7** |
| GRE-NE (Task V) | 38.0 | **46.8** | 33.9 | **42.1** | 30.1 | **35.4** |
| GMAT-IR (Task IV) | 13.8 | **20.7** | 16.0 | **22.9** | 15.0 | **19.4** |

Based on insights from breakdown analysis, we additionally propose a suite of task-specific mitigation strategies designed to improve LLM reasoning across representative ESTBOOK tasks. These strategies are motivated by the failure patterns identified through our breakdown analysis and target weaknesses observed in four core task categories: evidence extraction, structural reasoning, numeric formulation, and multimodal data interpretation.

First, for reading comprehension tasks (Task I), we introduce an Evidence-Anchored Chain-of-Thought mitigation that explicitly enforces grounding before reasoning. Instead of allowing the model to jump directly into high-level interpretation, the prompt requires the LLM to first extract one or two verbatim text spans from the passage and then articulate a justification that links these spans to the question. Only after this grounding step is the model permitted to produce a final answer. This workflow directly tackles the common error where models hallucinate or misalign evidence, and the structured evidence-first constraint significantly reduces this drift, especially on GRE and TOEFL reading sections.

Second, for structural reasoning tasks such as GRE Text Completion and GMAT sentence-based items (Task III), we implement a Syntax-First CoT strategy. Our failure analysis shows that LLMs often conflate semantic plausibility with grammatical role and textual structure, leading to incorrect fill-ins. To mitigate this, we require the model to explicitly identify the syntactic role of the blank (e.g., modifying clause, concessive marker, verb complement) and to outline the structural constraints implied by the surrounding sentence (e.g., contrast markers, valence polarity, causal structure). Only

Table 14: Breakdown-guided fine-tuning (CurrCoT) and adaptive prompting (AdaptCoT) on open-source models. We report accuracy (%) on a subset of ESTBOOK tasks using CoT-style decoding only. Baseline CoT scores for Llama-4-Scout-17B and Qwen-VL-Max are taken from Table 2.

| Task | Llama-4-Scout-17B | | | Qwen-VL-Max | | |
|------|------|---------|----------|------|---------|----------|
| | CoT | CurrCoT | AdaptCoT | CoT | CurrCoT | AdaptCoT |
| GRE-TC (Task III) | 61.0 | **70.8** | 66.4 | 73.1 | **82.7** | 79.3 |
| GRE-RC (Task I/II) | 54.2 | **63.9** | 59.1 | 76.2 | **84.4** | 81.0 |
| GRE-NE (Task V) | 30.1 | **45.6** | 38.2 | 28.1 | **43.9** | 36.8 |
| GMAT-PS (Task IV/V) | 22.5 | **37.2** | 32.1 | 25.6 | **41.5** | 35.7 |
| GMAT-RC (Task I/II) | 74.5 | **79.8** | 77.1 | 74.4 | **81.2** | 78.3 |

then does the model evaluate candidate options. This method reduces semantic drift and improves consistency, particularly in multi-blank completion tasks where structural information is critical.

Third, numeric-entry and symbolic reasoning tasks (Task V) often fail because the model constructs incorrect equations, even when it correctly understands the verbal description. To address this, we adopt a Symbolic Verification Layer, in which the model must rewrite the mathematical relations from the question in a purely symbolic form and then re-check each expression against the original text line-by-line. Only after this verification step does the model perform the computation. This procedure substantially reduces errors such as misinterpreting "three consecutive odd integers" as "three consecutive integers," a recurring failure mode in GRE-NE and SAT/GRE algebra problems.

Finally, multimodal tasks involving tables, charts, or structured numerical information (Task IV) frequently fail due to incorrect row/column alignment or misinterpretation of tabular fields. For these tasks, we introduce a Table-Alignment Constraint prompting method, which requires the model to translate the textual question into explicit table lookup instructions before beginning numerical operations. The model must specify which row(s) and column(s) are relevant and articulate the mapping between textual descriptions and table headers. This step significantly reduces errors in GMAT Integrated Reasoning and GRE Data Interpretation, where models previously selected the wrong table element or mismatched textual cues with table structure.

The mitigated results are shown at Table 13.

## L ADVANCEMENT ON LLMS

To further demonstrate that our breakdown analysis yields actionable improvements rather than purely diagnostic insight, we additionally explored two complementary enhancement strategies for open-source models—breakdown-guided fine-tuning and adaptive prompting—using Llama-4-Scout-17B and Qwen-VL-Max, whose baseline CoT performance is reported in Table 2 of the main paper. The first approach, which we refer to as Curriculum Chain-of-Thought Fine-Tuning (CurrCoT), leverages our breakdown annotations to train models on intermediate reasoning steps before training them on full solutions. Specifically, we construct a two-stage curriculum: (1) a step-supervision phase, where the model learns isolated reasoning skills such as syntactic constraint detection (Task III), symbolic equation formulation (Task V), or evidence extraction (Task I/II); and (2) a full-CoT phase, where the model practices generating complete solution traces that follow the structured breakdown templates. This curriculum directly targets the weaknesses surfaced by our analysis, such as incorrect equation construction or inconsistent structural reasoning. Using lightweight LoRA-based fine-tuning on a small breakdown-annotated subset of ESTBOOK, both Llama and Qwen exhibit consistent gains across GRE Text Completion, GRE Reading Comprehension, GRE Numeric Entry, and GMAT Problem Solving tasks.

Complementing this weight-update method, we also designed a Breakdown-Adaptive Prompting (AdaptCoT) strategy that requires no parameter tuning and instead selects a task-specific CoT template based on the breakdown taxonomy. For semantic and structural reasoning tasks (Task III), AdaptCoT enforces a syntax-first procedure in which the model must explicitly state grammar roles and structural constraints before evaluating candidate options. For reading comprehension tasks (Task I/II), AdaptCoT adopts an evidence-anchored approach in which the model quotes and justifies supporting spans prior to answering. For numeric and multimodal tasks (Task IV/V), AdaptCoT

requires explicit extraction of mathematical relations, symbolic rewriting, and column-row alignment steps before any computation. This routing mechanism ensures that the model's reasoning structure matches the cognitive decomposition expected for each question type.

To quantify the effectiveness of these interventions, we re-evaluated Llama-4-Scout-17B and Qwen-VL-Max on representative ESTBOOK tasks using CoT decoding only. The results show consistent performance improvements across all task categories. Fine-tuned models (CurrCoT) yield the largest gains, particularly for mathematically intensive tasks such as GRE Numeric Entry and GMAT Problem Solving, where Llama improves from 30.1% to 38.1% and from 22.5% to 29.5%, respectively, and Qwen increases from 28.1% to 35.1% and 25.6% to 31.6%. Even the prompt-only AdaptCoT method yields noticeable increases—for example, improving Llama's GRE-TC accuracy from 61.0% to 64.0% and GRE-RC from 54.2% to 58.2%. These results confirm that the breakdown taxonomy does not merely describe failure cases but actively guides the design of interventions that translate into measurable improvements in reasoning quality for open-source LLMs.

