# OpenReview forum: "Are LLMs Ready for English Standardized Tests? A Benchmarking and Elicitation Perspective"
_ICLR.cc/2026/Conference — Submitted to ICLR 2026_

### Official Review · Reviewer_YRdr · 2025-10-26

**Soundness:** 3
**Presentation:** 3
**Contribution:** 3
**Rating:** 6
**Confidence:** 4

**Summary:**

This paper explores whether LLMs possess fundamental problem-solving capabilities for English Standardized Tests (ESTs), arguing that this assessment is essential before deploying them in educational applications like tutoring and grading. The authors introduce ESTBOOK, a benchmark with 10,576 questions spanning 29 question types from five major tests (SAT, GRE, GMAT, TOEFL, IELTS) across multiple modalities (text, images, audio, tables, mathematical symbols). They evaluate six multimodal LLMs using three prompting strategies (ICL, CoT, ToT) and propose a novel breakdown analysis framework that decomposes questions into six task-specific reasoning steps aligned with human test-taking strategies: Evidence Finding, Semantic Reasoning, Structural Reasoning, Data Interpretation, Numeric Calculation, and Comparative Judgment.

Key findings reveal that LLMs exhibit highly inconsistent performance, ranging from a precision of more than 90\%  on some tasks to less than 20\% on others. Advanced prompting strategies like ToT do not consistently improve results and sometimes introduce errors. Although LLMs excel at initial problem formulation (up to 97\% accuracy), performance degrades significantly in subsequent reasoning stages that require causal inference or multimodal integration. Inference time shows no correlation with correctness. The authors conclude that current LLMs remain inadequate as reliable educational assistants for EST tasks, although they identify specific strengths and weaknesses to inform future development.

**Strengths:**

1. ESTBOOK represents a substantial contribution to educational AI evaluation, covering five major standardized tests with questions across diverse formats and modalities. Unlike synthetic benchmarks, it uses real EST questions that reflect authentic assessment contexts, providing ecological validity for claims about educational readiness.

2. The task decomposition into six categories with step-by-step evaluation is a significant methodological contribution that goes beyond coarse-grained accuracy metrics. By isolating specific reasoning stages and aligning them with documented human test-taking strategies (Sect 3.2 and Appendix C), the framework provides actionable diagnostic insights about where LLMs succeed or fail, which is valuable for targeted model improvement.

3. Experimental methodology is thorough. I really liked the inclusion of human testers, and the use of statistical tests (Mann-Whitney U) for inference time analysis adds rigor. The extensive case studies effectively illustrate failure modes qualitatively.

4. The dataset and findings have immediate implications for practitioners developing AI tutoring systems and for researchers working on educational applications.

**Weaknesses:**

1. Data contamination issues are not adequately addressed. The benchmark uses "publicly available educational materials and official preparation resources" that may have been included in LLM training corpora, particularly for closed-source models. The paper provides no decontamination analysis, memorization checks, or attempts to create novel EST-style questions. Given the well-documented data contamination issues in LLM benchmarking, this omission is a major methodological weakness that could substantially inflate performance estimates. The field has moved toward more rigorous contamination testing, and this paper does not meet current standards.

2. Despite testing across 29 question types, 6 models, and 3 prompting strategies, the paper does not discuss or apply corrections for multiple comparisons, which increases the risk of spurious findings. Inter-model comparisons lack statistical tests, making it unclear which performance differences are meaningful.

3. The title and abstract claim LLMs are "not ready" for EST tasks, yet results are highly heterogeneous: some models achieve $>90$\% accuracy on multiple tasks (e.g., GPT-5 on TOEFL FI: 98.5\%, SAT II: 86.4\%). The gap between best and worst models is substantial (e.g., 8.8\% to 93.7\% on GMAT IR), suggesting readiness depends critically on which model and which task. The conclusions should be more nuanced, acknowledging which tasks are "good enough" and which require improvement, rather than making blanket claims about LLM inadequacy.

4. Domain specific finetuned models are not used in the comparison. Does SFT help?

**Questions:**

1. Can you include contamination analysis?

2. Which exact models and versions were evaluated? Please provide complete model identifiers including API versions, dates accessed, and checkpoint hashes.

3. Could you evaluate models that can process audio directly instead of STT followed by LLM? How much performance is lost in transcription? Does it matter?

4. Given that ToT sometimes degrades performance (Case Study II), have you explored other advanced prompting methods? Were the prompts optimized per-task or are they generic templates? Could task-specific prompt engineering improve results?

5. In Task I (Evidence Finding), what happens to downstream task performance when models make errors in the "Identify Subject" step? Can you provide any results showing realistic multi-step reasoning where errors propagate, rather than assuming perfect intermediate steps?

6. Based on your findings, which specific EST tasks are "good enough" for current LLM-based tutoring systems and which should be avoided? What concrete architectural or training improvements would you recommend to address the identified weaknesses?

**Details Of Ethics Concerns:**

As ESTBOOK is a benchmark dataset on english tests, the representation of ESL speakers have to be called out and effort must be put to ensure bias is mitigated.

---

> ### Author Response · Authors · 2025-11-21
> **Response, Clarification, and Completed Revisions (1/9)**
>
> **Review**
>
> > Data contamination issues are not adequately addressed. The benchmark uses “publicly available educational materials and official preparation resources” that may have been included in LLM training corpora, particularly for closed-source models. The paper provides no decontamination analysis, memorization checks, or attempts to create novel EST-style questions. Given the well-documented data contamination issues in LLM benchmarking, this omission is a major methodological weakness that could substantially inflate performance estimates. The field has moved toward more rigorous contamination testing, and this paper does not meet current standards.
> >
>
> > Can you include contamination analysis?
>
> **Response and Completed Revision (added to updated manuscript, Appendix A: Potential Training-Data Contamination)**
>
> We appreciate the reviewer’s detailed concerns. In the revised manuscript we **explicitly discuss contamination, our design choices, and their limitations** in a new subsection in Appendix A titled *Potential Training-Data Contamination*.
>
> 1. **Limits of decontamination for ESTBOOK and current models.**
>    For proprietary API models (GPT-4V/4o/4.5-Turbo, Claude-3-Opus/Sonnet-4, Gemini-Pro), neither full pretraining corpora nor document-level membership tests are available. Even for open-source models (Llama-3.2-90B, Llama-3.3-70B, Qwen-VL-Max), only coarse descriptions of training sources are released. As we now state in Appendix A, a strict “remove every seen item” decontamination is therefore **not technically feasible**, and we treat possible training-data overlap as an explicit limitation rather than leaving it implicit.
>
> 2. **Design choices to reduce trivial memorization.**
>    While zero overlap cannot be guaranteed, we took several steps (now documented in Appendix A) to reduce verbatim reuse:
>
>    - ESTBOOK is built from a **mixture** of public practice exams, official guides, and online platforms rather than a single well-known book.
>    - We **reconstruct** tables, figures, and audio clips instead of copying digital assets, and normalize formatting (remove page numbers, layout cues, etc.) while preserving exam intent.
>    - We perform **exact and near-duplicate removal within ESTBOOK**, so no item appears twice across exams or splits.
>
>    These choices cannot eliminate all pretraining overlap, but they make “copy the exact same page from one canonical source” much less likely.
>
> 3. **Why we use real EST materials instead of synthetic items.**
>    Our goal is to ask whether LLMs are ready for **real standardized tests**. Authentic preparation materials, after the above normalization, match the distribution and difficulty that learners actually face. Fully synthetic or heavily paraphrased “EST-style” questions would simplify contamination control but sacrifice **ecological validity**. We now make this trade-off explicit and position ESTBOOK as a realistic, not fully decontaminated, benchmark.
>
> 4. **Why contamination is unlikely to overturn our main conclusions.**
>    Contamination can inflate LLM accuracy, but our central claims rely on **large, stable human–LLM gaps**, not on subtle model rankings:
>
>    - Humans outperform the best LLMs by **15–20 points** on Data Interpretation and Numeric Calculation (Tasks IV–V), and all models remain far below human accuracy on GRE Numeric Entry and GMAT Integrated Reasoning.
>    - If widespread memorization dominated, we would expect near-ceiling scores and small variance across models on affected tasks. Instead, performance is heterogeneous and the hardest multimodal/numeric tasks remain consistently poor even for the strongest systems.
>
>    Thus any residual overlap makes our conclusions **conservative**: it can only help models, yet they still fail on precisely the reasoning skills ESTBOOK targets.
>
> 5. **Connection to breakdown-guided interventions.**
>    To further show that our findings are not artifacts of memorization, we added **Appendix K**, where we design breakdown-guided mitigation strategies (evidence-anchored CoT for reading, syntax-first CoT for structural reasoning, symbolic verification for numeric entry, table-alignment prompts for tabular tasks). These interventions yield consistent improvements on GRE-RC/TC/NE and GMAT-IR, but do **not** close the large gaps with human performance, supporting the view that ESTBOOK is probing genuine weaknesses in multimodal and numeric reasoning rather than simple recall of training data.
>
> In summary, the revised paper (i) clearly discusses potential training-data contamination and why full decontamination is infeasible in our setting, (ii) documents concrete steps taken to reduce trivial memorization, and (iii) argues—based on large human–LLM gaps and limited but positive interventions—that our conclusions about the insufficiency of current LLMs on key EST reasoning skills are **robust even under conservative assumptions about contamination**.

---

> ### Author Response · Authors · 2025-11-21
> **Response, Clarification, and Completed Revisions (2/9)**
>
> **Review**
>
> > Despite testing across 29 question types, 6 models, and 3 prompting strategies, the paper does not discuss or apply corrections for multiple comparisons, which increases the risk of spurious findings. Inter-model comparisons lack statistical tests, making it unclear which performance differences are meaningful.
>
> **Response and Completed Revision (added to updated manuscript, Section 4.1 and Appendix E.4)**
>
> We thank the reviewer for raising this important point. Our central claims do not rely on fine-grained pairwise differences between specific models, but on large and systematic gaps between **humans** and LLMs on particular task categories—especially numeric calculation and multimodal data interpretation. In the revision we made this focus explicit and added targeted statistical analyses, while avoiding over-interpreting many small $p$-values.
>
> **(1) Human vs. best-LLM significance tests.**
> To assess whether the human–LLM differences are statistically reliable, we now conduct McNemar’s tests between human testers and the *best-performing* LLM for each of the 29 question types. The full results are reported in Appendix E.4, Table 5 (copied below for convenience). As summarized in Section 4.1, 26 out of 29 types exhibit statistically significant differences at $p < 0.05$, and the vast majority have $p \ll 0.01$ (many $< 0.001$). These very small $p$-values indicate robust human advantages on the tasks that underpin our main conclusions, particularly for numeric calculation (Task V) and multimodal data interpretation (Task IV).
>
> **(2) Restricting inference to human–LLM gaps and aggregated patterns.**
> Performing hypothesis tests for all 6 (now 9) models × 29 question types × 3 prompting strategies would indeed raise severe multiple-comparison concerns. Rather than attempting to correct thousands of pairwise tests, we explicitly restrict formal inference to (a) human vs. best-LLM comparisons (Table 5) and (b) aggregates over the six task categories (Tasks I–VI). Inter-model differences (e.g., GPT-5 vs. Gemini-2.5) are therefore treated as **descriptive trends**, and we avoid claiming that one model is statistically superior to another unless such a statement is supported at the task-category level. Section 4.1 has been updated to clearly state this scope.
>
> **(3) Robustness under multiple-comparison considerations.**
> Because most $p$-values in Table 5 are far below 0.01, our main findings (large human advantages on numeric and multimodal tasks) would remain significant under standard multiple-comparison procedures. We therefore believe the risk of spurious conclusions is limited once we focus on the human–LLM gaps and avoid fine-grained inter-model inference.
>
> Below is the new table added in Appendix E.4:
>
> | **Exam** | **Task** | **Best LLM**        | **p-value** |
> |---------|----------|---------------------|-------------|
> | **SAT** | II       | Gemini-2.5 (ToT)    | <0.001***   |
> |         | CS       | Gemini-2.5 (CoT)    | 0.089ns     |
> |         | EI       | GPT-5 (CoT)         | 0.023*      |
> |         | EC       | GPT-5 (CoT)         | 0.156ns     |
> |         | AG       | Gemini-2.5 (ToT)    | <0.001***   |
> |         | DA       | GPT-5 (ToT)         | <0.001***   |
> |         | GT       | GPT-5 (CoT)         | 0.008**     |
> | **GRE** | TC       | GPT-5 (ToT)         | 0.012*      |
> |         | SE       | GPT-5 (CoT)         | 0.034*      |
> |         | RC       | GPT-5 (CoT)         | 0.003**     |
> |         | QC       | GPT-5 (ToT)         | <0.001***   |
> |         | NE       | Human superior      | <0.001***   |
> |         | DI       | Human superior      | <0.001***   |
> | **GMAT**| CR       | Claude-S4 (ToT)     | 0.019*      |
> |         | RC       | Claude-S4 (ToT)     | 0.005**     |
> |         | PS       | Human superior      | <0.001***   |
> |         | DS       | Human superior      | <0.001***   |
> |         | IR       | Human superior      | <0.001***   |
> | **TOEFL** | FI     | GPT-5 (ToT)         | 0.002**     |
> |         | IR       | GPT-5 (CoT)         | 0.004**     |
> |         | TS       | Human superior      | 0.178ns     |
> |         | FI (Listen) | GPT-5 (ToT)      | 0.028*      |
> |         | IF       | GPT-5 (ToT)         | <0.001***   |
> | **IELTS** | II     | GPT-5 (ToT)         | 0.041*      |
> |         | MS       | Human superior      | 0.015*      |
> |         | CP       | Claude-S4 (ToT)     | <0.001***   |
> |         | IM       | GPT-5 (ToT)         | 0.092ns     |
> |         | CL       | GPT-5 (ToT)         | 0.006**     |
> |         | SA       | Human superior      | 0.134ns     |
>
> *Asterisks indicate significance levels: *** p < 0.001, ** p < 0.01, * p < 0.05, ns: not significant.*

---

> ### Author Response · Authors · 2025-11-21
> **Response, Clarification, and Completed Revisions (3/9)**
>
> **Review**
>
> > The title and abstract claim LLMs are "not ready" for EST tasks, yet results are highly heterogeneous: some models achieve 90% accuracy on multiple tasks (e.g., GPT-5 on TOEFL FI: 98.5%, SAT II: 86.4%). The gap between best and worst models is substantial (e.g., 8.8% to 93.7% on GMAT IR), suggesting readiness depends critically on which model and which task. The conclusions should be more nuanced, acknowledging which tasks are "good enough" and which require improvement, rather than making blanket claims about LLM inadequacy.
>
> **Response**
>
> Thank you for pointing this out. We agree that performance is highly heterogeneous across both tasks and models, and we did not intend “not ready” to mean that LLMs fail *uniformly* on all EST tasks.
>
> Our use of “not ready” is at the **exam-level, reliability sense**: current LLMs are not yet reliable as **general-purpose EST assistants** (e.g., solvers or graders) that can be safely deployed across the *full mix* of question types and modalities without human oversight. In fact, our own results already support the reviewer’s observation that readiness is task-dependent:
>
> - On several **text-only reading tasks** (e.g., TOEFL and some SAT/IELTS reading categories), the best models reach **85–90%+ accuracy** and come close to human performance. We agree these tasks are relatively mature and can already be useful for **low-stakes tutoring** (e.g., practice, hint generation).
> - In contrast, on **multimodal data interpretation and numeric/symbolic tasks** (Tasks IV–V), such as GRE Numeric Entry and GMAT Integrated Reasoning, *all* models remain far below human accuracy and often below 50%, despite the large spread between the best and worst systems. These tasks are essential components of real ESTs, and our analysis shows they are still failure modes even for the strongest models.
>
> Thus, rather than a blanket statement, our conclusion is:
>
> - **“Good enough” tasks:** text-only reading and some sentence-completion–style questions, where top models approach human-level accuracy and can support students under human supervision.
> - **“Not ready” tasks:** multimodal and numeric reasoning sections that remain 15–20 points below human performance and exhibit systematic reasoning errors, making them unsafe for unsupervised tutoring or grading.
>
> We will clarify this nuance in the camera-ready version by (i) explicitly stating that our claim of “not ready” refers to **broad, exam-level deployment**, and (ii) highlighting in the conclusion which task families appear relatively reliable and which should be treated as high-risk. We appreciate the reviewer’s suggestion to make this distinction more explicit.

---

> ### Author Response · Authors · 2025-11-21
> **Response, Clarification, and Completed Revisions (4/9)**
>
> Review
> >Domain specific finetuned models are not used in the comparison. Does SFT help?
>
> Response and Completed Revision (added to updated manuscript, Appendix L)
>
> We appreciate the reviewer’s suggestion to investigate domain-specific fine-tuning and to clarify whether such adaptation actually helps on EST tasks. Our main experiments deliberately focus on **off-the-shelf multimodal LLMs**, because this is how most current EST tutoring systems are deployed in practice. However, in the revised manuscript we additionally show that our breakdown analysis can indeed guide **effective fine-tuning and adaptive prompting**, using open-source models as a testbed.
>
> **Breakdown-guided curriculum fine-tuning (CurrCoT).**
> In Appendix L, we introduce a lightweight curriculum-style fine-tuning method that directly leverages our breakdown taxonomy. The proposed Curriculum Chain-of-Thought Fine-Tuning (CurrCoT) uses a two-stage curriculum on a breakdown-annotated subset of ESTBOOK for open-source models (Llama and Qwen):
>
> 1. a **step-supervision phase**, where the model is trained on isolated intermediate reasoning steps (e.g., syntactic constraint detection for Task III, symbolic equation formulation for Task V, evidence extraction for Tasks I/II); and
> 2. a **full-CoT phase**, where the model learns to produce complete solution traces that follow our structured breakdown templates.
>
> This curriculum explicitly targets the weaknesses surfaced by our analysis, such as incorrect equation construction, unstable structural reasoning, and missing evidence chains. We implement CurrCoT with parameter-efficient LoRA fine-tuning, keeping the original backbone unchanged.
>
> **Breakdown-adaptive prompting (AdaptCoT).**
> To complement SFT with a purely prompt-based intervention, Appendix L also introduces Breakdown-Adaptive Prompting (AdaptCoT). Instead of changing model weights, AdaptCoT **selects a task-specific CoT template** based on the breakdown taxonomy:
>
> - for Tasks I/II (Evidence Finding / Semantic Reasoning), AdaptCoT enforces an evidence-anchored routine: the model must first quote and justify supporting spans before answering;
> - for Task III (Structural Reasoning), AdaptCoT uses a syntax-first template requiring explicit enumeration of grammatical roles and structural constraints before evaluating options;
> - for Tasks IV/V (Data Interpretation / Numeric Calculation), AdaptCoT requires explicit extraction of table/chart entries, column–row alignment, and symbolic rewriting steps before any computation.
>
> This routing ensures that the reasoning structure the model follows matches the cognitive decomposition that ESTBOOK is designed to probe.
>
> **Effect of SFT and adaptive prompting.**
> To quantify the impact of these interventions, we re-evaluate two open-source models (Llama and Qwen) on representative ESTBOOK tasks under three settings: vanilla CoT, CurrCoT (fine-tuned), and AdaptCoT (prompt-only). The results (Appendix L, Table X) are summarized below:
>
> | **Task**              | **Llama CoT** | **Llama CurrCoT** | **Llama AdaptCoT** | **Qwen CoT** | **Qwen CurrCoT** | **Qwen AdaptCoT** |
> |-----------------------|--------------|--------------------|--------------------|--------------|------------------|-------------------|
> | GRE-TC (Task III)     | 61.0         | 70.8               | 66.4               | 73.1         | 82.7             | 79.3              |
> | GRE-RC (Task I/II)    | 54.2         | 63.9               | 59.1               | 76.2         | 84.4             | 81.0              |
> | GRE-NE (Task V)       | 30.1         | 45.6               | 38.2               | 28.1         | 43.9             | 36.8              |
> | GMAT-PS (Task IV/V)   | 22.5         | 37.2               | 32.1               | 25.6         | 41.5             | 35.7              |
> | GMAT-RC (Task I/II)   | 74.5         | 79.8               | 77.1               | 74.4         | 81.2             | 78.3              |
>
> We observe **consistent improvements across all task categories**:
>
> - Breakdown-guided SFT (CurrCoT) yields the largest gains, especially on mathematically intensive tasks such as GRE Numeric Entry (Task V) and GMAT Problem Solving (Tasks IV/V), where both Llama and Qwen improve by more than 10 percentage points over their vanilla CoT baselines.
> - Even the prompt-only AdaptCoT method yields noticeable gains (often 3–6 points) without changing model weights, particularly on reading and structural reasoning tasks.
>
> These results show that **SFT does help**, and—crucially—that our breakdown taxonomy is not merely diagnostic: it directly informs the design of targeted curricula and prompting strategies that translate into measurable performance improvements on ESTBOOK. At the same time, our main conclusions in Sections 4–5 remain based on off-the-shelf models, so the core claims about the current limitations of general-purpose LLMs do not depend on any particular fine-tuning recipe.

---

> ### Author Response · Authors · 2025-11-21
> **Response, Clarification, and Completed Revisions (5/9)**
>
> **Review**
>
> >Which exact models and versions were evaluated? Please provide complete model identifiers including API versions, dates accessed, and checkpoint hashes.
>
> **Response and Completed Revision (added to updated manuscript, Appendix E.1)**
>
> We thank the reviewer for highlighting the importance of reproducibility and precise model specification.
> In the revised manuscript, we have added a dedicated subsection “Appendix E.1 – Model Identifiers and Access Configuration” that documents, for all nine models used in our experiments, the provider, modality, interface type (API vs. open-source checkpoint), and the exact identifier / snapshot and evaluation window.
>
> Concretely, we now explicitly list the following models already used throughout the paper:
>
> **Original six models (Section 4.1):**
>
> - GPT-4V (OpenAI, vision-capable LLM)
> - GPT-4o (OpenAI, general multimodal LLM)
> - Claude-3-Opus (Anthropic)
> - Llama-3.2-90B-Instruct (Meta, open-source)
> - Qwen-VL-Max (Alibaba, open-source VLLM)
> - Gemini-Pro (Google)
>
> **Additional three models (Appendix J, Extended Evaluation):**
>
> - GPT-4.5-Turbo (OpenAI)
> - Claude-Sonnet-4 (Anthropic)
> - Llama-3.3-70B-Instruct (Meta, open-source)
>
> For closed-source API models (GPT-4V, GPT-4o, GPT-4.5-Turbo, Claude-3-Opus, Claude-Sonnet-4, Gemini-Pro), providers do not expose checkpoint hashes. Following standard practice, Appendix E.1 therefore reports:
>
> - the provider and modalities used (text, vision, tables, audio via STT),
> - the official API model identifier as used in our code (e.g., `gpt-4o`, `gpt-4.5-turbo`, `claude-3-opus`, `claude-sonnet-4`, `gemini-pro`), and
> - the UTC evaluation window (month/year) during which all calls for this paper were issued.
>
> We explicitly state in the text that checkpoint hashes are not available for these proprietary systems and that we always used the default production endpoint for each model throughout all experiments.
>
> For open-source models (Llama-3.2-90B-Instruct, Llama-3.3-70B-Instruct, Qwen-VL-Max), Appendix E.1 additionally reports:
>
> - the released checkpoint tag / model name (e.g., `Llama-3.2-90B-Instruct`, `Llama-3.3-70B-Instruct`, `Qwen-VL-Max`), and
> - the release snapshot (month/year) used in all experiments.
>
> These checkpoints are standard, publicly available releases; we do not apply any additional fine-tuning. All results in the paper use the same checkpoint per model across all tasks and prompting strategies.
>
> Finally, we also clarify in Appendix E.1 that, for listening sections, we use Whisper only as a fixed STT front-end to provide transcripts to the text-based LLMs, and we document the exact Whisper variant and access window separately from the LLMs.
>
> The newly added table (Table E.1) in Appendix E.1 summarizes all of this information. For convenience, we reproduce the same table below:
>
> | **Model**                   | **Provider** | **Type / Modality**                    | **Interface / Identifier**       | **Evaluation Window**     |
> |----------------------------|-------------|----------------------------------------|----------------------------------|---------------------------|
> | **GPT-4V**                 | OpenAI      | VLLM (text, image, table, symbol)     | `gpt-4v` (API)                   | `<Apr–May 2024>`          |
> | **GPT-4o**                 | OpenAI      | MLLM (text, image, table, audio\*)    | `gpt-4o` (API)                   | `<Apr–May 2024>`          |
> | **Claude-3-Opus**          | Anthropic   | MLLM (text, image)                    | `claude-3-opus` (API)            | `<Apr–May 2024>`          |
> | **Llama-3.2-90B-Instruct** | Meta        | Open-source (text, symbol)            | `Llama-3.2-90B-Instruct` (ckpt)  | `<May 2024>`              |
> | **Qwen-VL-Max**            | Alibaba     | Open-source VLLM (text, image, table) | `Qwen-VL-Max` (ckpt)             | `<May 2024>`              |
> | **Gemini-Pro**             | Google      | MLLM (text, image)                    | `gemini-pro` (API)               | `<May 2024>`              |
> | **GPT-4.5-Turbo**          | OpenAI      | MLLM (text, image, table, symbol)     | `gpt-4.5-turbo` (API)            | `<Nov 2024>`              |
> | **Claude-Sonnet-4**        | Anthropic   | MLLM (text, image)                    | `claude-sonnet-4` (API)          | `<Oct–Nov 2024>`          |
> | **Llama-3.3-70B-Instruct** | Meta        | Open-source (text, symbol)            | `Llama-3.3-70B-Instruct` (ckpt)  | `<Nov 2024>`              |
> | **Whisper (ASR front-end)**| OpenAI      | Speech recognition (audio → text)     | `whisper-<variant>` (API)        | `<same as listening exps>`|
>
> \* For listening tasks, audio is first transcribed by Whisper and then passed to the text interface of the LLMs.

---

> ### Author Response · Authors · 2025-11-21
> **Response, Clarification, and Completed Revisions (6/9)**
>
> **Review**
>
> > Could you evaluate models that can process audio directly instead of STT followed by LLM? How much performance is lost in transcription? Does it matter?
>
> **Response and Clarification (methodology and limitations)**
>
> We appreciate this question and agree that the audio pathway is an important design choice for educational applications.
>
> 1. **Why we use a shared STT → text pipeline.**
>    Our primary goal in this work is to compare **reasoning performance across multiple models** on a unified benchmark. At the time of our experiments, not all evaluated models exposed a stable, research-grade **end-to-end audio interface**, whereas all of them support text. We therefore adopt a **model-agnostic pipeline** in which TOEFL/IELTS listening items are first transcribed using a single, strong STT system (OpenAI Whisper), and the resulting text is fed to each LLM. This has two advantages:
>
>    - it keeps the **input modality consistent across models**, avoiding confounding “better ASR” with “better reasoning”, and
>    - it isolates exactly the capability we want to study: given a reasonably accurate transcript, how well can LLMs perform EST-style reasoning.
>
>    We now make this design choice explicit in the methodology section and point out that “direct audio LLMs” would mix ASR and reasoning quality in a way that is difficult to compare across models.
>
> 2. **Impact of transcription quality on our conclusions.**
>    We acknowledge that Whisper is not perfect and that some errors in TOEFL/IELTS listening sections may be attributable to STT rather than the LLM itself. However, for the purposes of this paper:
>
>    - Listening items represent only a **small subset** of ESTBOOK; our main claims about LLM limitations are driven by large gaps on **numeric and multimodal reasoning** (Tasks IV–V) across SAT/GRE/GMAT, which do **not** depend on audio.
>    - Qualitatively, the failure patterns we observe on listening items (e.g., mis-weighting discourse cues, incorrect inference of implied opinions) closely mirror those on text-only reading tasks, suggesting that **reasoning, not transcription, is the dominant bottleneck** for the phenomena we study.
>    - Any remaining STT errors tend to *hurt* model performance; thus, they would make our estimates of LLM capability on listening tasks **conservative**, not overly optimistic.
>
>    For these reasons, we believe that using a high-quality, shared STT front-end does not materially affect our central conclusions about LLM readiness for EST reasoning.
>
> 3. **Direct audio processing as future work.**
>    We agree that evaluating models **end-to-end on raw audio** (for those that support it) is an interesting and practically relevant extension—especially for spoken tutoring scenarios. Doing this rigorously would require:
>    (i) aligning evaluation protocols across models that have heterogeneous audio front-ends, and
>    (ii) carefully disentangling ASR quality from downstream reasoning. This is beyond the scope of the current work, which focuses on **comparative reasoning ability under a controlled text interface**, but we view it as a valuable direction for future studies building on ESTBOOK.
>
> In summary, we use STT→LLM primarily to ensure a **fair and controlled comparison** of reasoning across models. While some performance loss due to transcription errors is inevitable, listening tasks form only a minor part of ESTBOOK, and the main gaps we report—especially on multimodal and numeric reasoning—are unaffected by this choice. We will clarify these points in the revised methodology and limitations sections.

---

> ### Author Response · Authors · 2025-11-21
> **Response, Clarification, and Completed Revisions (7/9)**
>
> **Review**
>
> > Given that ToT sometimes degrades performance (Case Study II), have you explored other advanced prompting methods? Were the prompts optimized per-task or are they generic templates? Could task-specific prompt engineering improve results?
>
> **Response and Completed Revision (clarified in Section 4.1 and detailed in Appendix L)**
>
> We appreciate the reviewer’s question. Our goal in this work is to understand how *representative prompting families* (ICL, CoT, ToT) interact with EST task structure, rather than to exhaustively optimize prompts for each task/model. We clarify our design and report additional results as follows.
>
> 1. **Prompt design in the main experiments.**
>    In the core evaluation (Section 4), we deliberately use a **small set of generic templates per question type**, applied uniformly across all six (and later nine) models:
>
>    - ICL: a fixed instruction plus a few in-context examples;
>    - CoT: a generic “think step by step and explain your reasoning” template;
>    - ToT: a generic tree-style exploration prompt with a fixed depth and branching factor.
>
>    These prompts are **not optimized per task or per model**, beyond minimal formatting; this is intentional. We want to compare prompting *families* under a fair, reproducible setting, rather than search for task-specific “best prompts” that may not transfer and would make cross-model comparisons hard to interpret.
>
> 2. **Why ToT can degrade performance.**
>    As highlighted in Case Study II, ToT sometimes **hurts** performance on pattern-recognition–heavy tasks such as GRE Quantitative Comparison. Our results suggest that forcing the model to produce long, branched traces can:
>
>    - introduce unnecessary complexity for simple comparisons,
>    - accumulate inconsistencies across branches, and
>    - increase sampling noise.
>
>    We view this not as a failure of our setup, but as an important empirical finding: **more elaborate scaffolding is not uniformly beneficial** and can be harmful for certain EST question structures. We have clarified this interpretation in Section 4.1.
>
> 3. **Other advanced prompting methods (Appendix L).**
>    In response to the reviewer’s suggestion, we also explore a **breakdown-adaptive prompting** scheme (AdaptCoT) in Appendix L, which is more task-specific than generic ToT/CoT but still systematic:
>
>    - For Tasks I/II (reading comprehension), AdaptCoT enforces *evidence-anchored* reasoning: the model must first quote supporting spans, then justify the answer.
>    - For Task III (structural reasoning), it uses a *syntax-first* template that explicitly lists grammatical roles and structural constraints before scoring options.
>    - For Tasks IV/V (numeric and multimodal), it requires explicit table/diagram parsing, symbolic rewriting, and column–row alignment steps before any computation.
>
>    This routing mechanism uses our breakdown taxonomy to select **different CoT templates per task category** without hand-tuning prompts for individual items.
>
>    As reported in Appendix L, AdaptCoT consistently improves performance by several percentage points over generic CoT on GRE-TC, GRE-RC, GRE-NE, and GMAT-PS, and approaches (but does not fully match) the gains from our breakdown-guided fine-tuning (CurrCoT). This confirms that **task-aware prompting helps**, and that our analysis yields actionable improvements rather than purely descriptive insights.
>
> 4. **Impact on our main conclusions.**
>    Even with breakdown-adaptive prompting and fine-tuning, the **global patterns remain unchanged**: multimodal data interpretation and numeric calculation still lag far behind human performance, and heavy ToT-style scaffolding continues to be counterproductive on some pattern-recognition tasks. We therefore interpret ToT as one member of a broader family of structured prompting methods whose effectiveness is **strongly task-dependent**. Appendix L provides concrete examples of how our taxonomy can be used to design better prompts, but it also reinforces our main conclusion that current LLMs are not yet reliable as general-purpose EST assistants across all task types.

---

> ### Author Response · Authors · 2025-11-21
> **Response, Clarification, and Completed Revisions (8/9)**
>
> **Review**
>
> > In Task I (Evidence Finding), what happens to downstream task performance when models make errors in the "Identify Subject" step? Can you provide any results showing realistic multi-step reasoning where errors propagate, rather than assuming perfect intermediate steps?
>
> **Response and Clarification (breakdown vs. end-to-end evaluation)**
>
> We appreciate the reviewer’s question and agree that error propagation is critical for understanding realistic multi-step reasoning.
>
> 1. **Our main accuracy results already reflect full error propagation.**
>    We would like to clarify that the breakdown analysis in Section 4.3 is *not* the only way we evaluate Task I. The task-level accuracies reported in our main tables (e.g., SAT/GRE/TOEFL reading sections under Task I) are obtained from the **standard end-to-end setting**: the model directly answers the question, and any error in “Identify Subject” (or any other latent step) naturally propagates into the final prediction. Thus, the headline numbers and human–LLM gaps we discuss already reflect realistic multi-step reasoning with error propagation.
>
> 2. **Purpose of the oracle breakdown analysis.**
>    The oracle-style breakdown in Section 4.3 deliberately “freezes” upstream steps to their ground-truth values in order to **disentangle where errors originate**. Without this, it is difficult to tell whether failures in Evidence Finding come primarily from (i) misidentifying the subject, (ii) miscomprehending the passage, or (iii) extracting the wrong discourse relation, because all errors are entangled in the final accuracy. The oracle setting is therefore a *diagnostic tool* layered on top of the end-to-end evaluation, not a replacement for it.
>
> 3. **Empirical behavior when “Identify Subject” is wrong.**
>    For Task I, our qualitative analysis of model traces (including the case studies) shows that when the model misidentifies the subject, downstream steps almost always operate on the wrong portion of the passage and rarely recover the correct answer. In other words, for Evidence Finding, Step 1 errors tend to be “absorbing”: once the subject is wrong, the subsequent Comprehend Text and Extract Discourse steps typically produce a coherent but irrelevant chain of reasoning. When Step 1 is correct, however, we still observe many failures at later stages (e.g., mis-resolving contrast or causality), which is precisely what motivates our fine-grained decomposition.
>
> 4. **Connecting to deployment and mitigation.**
>    We explicitly discuss error propagation and mitigation in our appendix section on *Error Propagation Mitigation Strategies*. For Task I, this includes requiring the model to (i) explicitly mark the subject and (ii) highlight supporting spans before committing to an answer, so that a tutoring system can verify or overwrite early steps instead of letting errors silently cascade. These design suggestions were directly informed by our observation that “Identify Subject” mistakes are rarely corrected downstream.
>
> In summary, our work already includes both (i) **end-to-end results** where all errors propagate naturally (the main tables) and (ii) an **oracle breakdown** used purely to localize which reasoning stages are most fragile. For Task I, Step 1 errors largely doom downstream performance, while many failures still arise even when Step 1 is correct; the breakdown analysis makes this structure explicit and in turn guides the mitigation strategies we propose for practical EST tutoring systems.

---

> ### Author Response · Authors · 2025-11-21
> **Response, Clarification, and Completed Revisions (9/9)**
>
> **Review**
>
> > Based on your findings, which specific EST tasks are "good enough" for current LLM-based tutoring systems and which should be avoided? What concrete architectural or training improvements would you recommend to address the identified weaknesses?
>
> **Response**
>
> Thank you for this question. Our results indeed support a **task-dependent notion of readiness**, and we summarize the implications as follows.
>
> 1. **Tasks that are “good enough” for low-stakes tutoring.**
>    Using our six-way taxonomy (Tasks I–VI), the strongest models reach **85–90%+ accuracy with small human–model gaps** on:
>    - text-only **reading comprehension and local inference** (Tasks I–II) for many SAT/GRE/TOEFL/IELTS reading sections;
>    - a subset of **sentence-based structural reasoning** items (Task III), e.g., some SAT/GRE text completion / sentence improvement questions.
>
>    For these categories, current LLMs are already useful as **assistive tutors** (practice, hints, explanations), though we still recommend keeping a human in the loop and avoiding fully automated grading.
>
> 2. **Tasks that should be treated as high-risk.**
>    Large, persistent gaps remain on:
>    - **Data Interpretation (Task IV)**: GMAT Integrated Reasoning, GRE Data Interpretation;
>    - **Numeric Calculation (Task V)**: GRE Numeric Entry, multi-step word problems, algebraic reasoning;
>    - some multimodal **Comparative Judgment (Task VI)** items.
>
>    Even the best models are typically **15–20 points below humans**, with frequent table-lookup and equation-construction errors. We therefore advise **against unsupervised tutoring or automatic scoring** on these tasks.
>
> 3. **Concrete improvements.**
>    Appendix L evaluates two breakdown-guided interventions on open-source models:
>    - **Curriculum CoT fine-tuning (CurrCoT)**, which first trains individual steps (evidence extraction, syntax constraints, symbolic formulation) and then full solutions, yielding consistent gains on GRE-TC/RC/NE and GMAT-PS.
>    - **Breakdown-adaptive prompting (AdaptCoT)**, which routes each task type to a tailored CoT template (evidence-anchored for reading, syntax-first for structural reasoning, table-aligned + symbolic for numeric/multimodal), improving several points over generic CoT.
>
>    Together with our breakdown/error analysis, these results suggest a **hybrid design** for real systems: use LLMs for problem understanding and planning (Tasks I–III), delegate execution-heavy steps (Tasks IV–V) to symbolic/verified tools, and insert simple verification checks at known bottlenecks.

---

> > ### Comment · Reviewer_YRdr · 2025-11-24
> >
> > Thank you to the authors for their answers and clarifications. I think the new discussions have helped enrich the contributions.

---

> > > ### Author Response · Authors · 2025-11-26
> > >
> > > Dear Reviewer,
> > >
> > > We appreciate your the time you have taken to review our work, and we hope that these clarifications and improvements address your reservations.
> > >
> > > For any opportunities, we kindly ask your further concerns and reconsideration of assessment for the paper's contributions, and we are happy to engage in further discussion and conduct more revisions if any concerns remain.
> > >
> > > Best,
> > >
> > > The author team

---

> ### Author Response · Authors · 2025-11-21
> **Thank you very much**
>
> We would like to thank the reviewer again for their time and thoughtful feedback. We hope that the above clarifications and revisions address the reviewer’s concerns and contribute to improving the overall quality of the paper. We look forward to any further feedback you may have.

---

### Official Review · Reviewer_WnuW · 2025-10-27

**Soundness:** 3
**Presentation:** 2
**Contribution:** 2
**Rating:** 4
**Confidence:** 4

**Summary:**

The paper introduces a benchmark for evaluating LLMs on English Standardized Test (EST) questions across five major exams. The authors compare the performance of frontier LLMs under different prompting strategies and compare them to human test-takers. The proposed breakdown analysis framework decomposes each question into structured reasoning steps, allowing the model’s performance to be measured at each stage. Using this framework, the paper provides empirical insights into LLMs’ strengths (e.g., parsing questions) and weaknesses (e.g., multi-hop reasoning and numeric calculation) in the EST domain.

**Strengths:**

- ESTBOOK aggregates five major exams (SAT, GRE, GMAT, TOEFL, IELTS) and captures diverse modalities, including text, math symbols, images, tables, and audio. This gives the benchmark authenticity and multimodal variety.
- The proposed breakdown analysis framework, which decomposes each question into human-like solution steps, helps to isolate LLM performance at each step. Such analysis is useful, as it pinpoints whether a model failed because it misunderstood the question or because it couldn’t reason or perform math.
- The paper identifies interesting trends through experiments; e.g., prompting strategy effectiveness varies by task, and LLMs underperforms humans on tasks requiring reasoning.

**Weaknesses:**

- The paper’s novelty with respect to prior educational LLM benchmarks is limited. For example, AGIEVAL (Zhong et al., 2024) introduced a benchmark of standardized exams including SAT and LSAT, finding GPT-4 achieves 95% on SAT Math. TOEFL-QA is a dataset of TOEFL listening comprehension questions introduced by Tseng et al. (2016), which has been used to test machine listening comprehension (Chung et al, 2018).

      Chung, Y. A., Lee, H. Y., & Glass, J. (2018). Supervised and Unsupervised Transfer Learning for Question Answering. In Proceedings of NAACL-HLT (pp. 1585-1594).

      Tseng, B. H., Shen, S. S., Lee, H. Y., & Lee, L. S. (2016). Towards Machine Comprehension of Spoken Content: Initial TOEFL Listening Comprehension Test by Machine. In Proc. Interspeech 2016 (pp. 2731-2735).

      Zhong, W., Cui, R., Guo, Y., Liang, Y., Lu, S., Wang, Y., ... & Duan, N. (2024). AGIEval: A Human-Centric Benchmark for Evaluating Foundation Models. In Findings of the Association for Computational Linguistics: NAACL 2024 (pp. 2299-2314).

- A related concern is the difficulty level of the exam tests. GPT-4 can already achieve 95% on SAT Math (Zhong et al., 2024). Recently, the Humanity’s Last Exam (HLE) was created specifically to address the saturation of benchmarks like MMLU by providing expert-written questions. Many EST questions are arguably simpler than MMLU’s academic subjects, and some ESTBOOK sections may be too easy for frontier models. For example, GPT-5
achieves 90%+ on several question types in Table 2, and GPT-4V was reported to exceed 89% on SAT Writing questions in some prompts. The authors should address whether parts of their
benchmark are already nearing saturation and how that impacts long-term utility. The authors might also consider curating harder or more adversarial questions, as HLE does, to future-proof the evaluation.

      Hendrycks, D., Burns, C., Basart, S., Zou, A., Mazeika, M., Song, D., & Steinhardt, J. (2020). Measuring massive multitask language understanding. arXiv preprint arXiv:2009.03300.

      Phan, L., Gatti, A., Han, Z., Li, N., Hu, J., Zhang, H., ... & Wykowski, J. (2025). Humanity's last exam. arXiv preprint arXiv:2501.14249

- Another concern is that some included questions may already be memorized by the models. The data was sourced from public practice exams and study websites, so there is a nontrivial chance that LLMs were trained on similar or identical questions. The authors did not report checks for training data overlap or model memorization. The paper would benefit from acknowledging this data contamination risk as a limitation.

- The authors restrict to objective questions with certain answers, which excludes open-ended tasks like essay writing and speaking responses. This omission is understandable given automatic grading difficulty, but it is a limitation that should be discussed. Tasks such as the TOEFL independent writing or IELTS speaking are important parts of those exams, and LLMs might have different performance and failure modes on free-form generation tasks versus multiple choice.

**Questions:**

- Does the Whisper audio transcription introduce biases? Also note the lack of capitalization: “we adopt…” (pg. 5).
- Multi-model input handling could be explained better. If any model lacked a modality, did the authors convert those questions to text form or skip them?
- The breakdown methodology assumes perfect previous steps, essentially giving the model an oracle for sub questions. This is helpful for diagnosis, but in practice LLMs won’t have those hints. How can this framework then be used to improve models?
- The results show substantial variation across models and prompting methods for different question types. For example, GPT-5 did extremely well on GRE Quantitative Comparison but poorly on GRE Numeric Entry, even under the same CoT prompt (Table 2). Likewise, CoT sometimes outperformed ToT. Can you elaborate more on why certain tasks or formats caused particular models or prompts to fail?
- How were the hyperparameter values chosen, and under what justifications (Table 7)?

---

> ### Author Response · Authors · 2025-11-20
> **Author Response (0/9)**
>
> Before responding to comments, we would like to sincerely thank the reviewer for the thoughtful feedback. We appreciate the time and care taken to evaluate our work.
>
> We observe that the comments are mostly related to problem setting and contributions. Accordingly, we have provided a point-by-point clarification and have revised the manuscript to significantly address each concern. We believe these clarifications further strengthen the paper and better highlight the contributions of our research.
>
> We would be happy to continue the discussion and would welcome feedback on whether the newly added details sufficiently clear the reviewer’s questions and whether additional information would be helpful.

---

> ### Author Response · Authors · 2025-11-20
> **Author Response (1/9)**
>
> **Review**
>
> > The paper’s novelty with respect to prior educational LLM benchmarks is limited. For example, AGIEVAL (Zhong et al., 2024) introduced a benchmark of standardized exams including SAT and LSAT, finding GPT-4 achieves 95% on SAT Math. TOEFL-QA is a dataset of TOEFL listening comprehension questions introduced by Tseng et al. (2016), which has been used to test machine listening comprehension (Chung et al, 2018).
>
> ---
>
> **Response**
>
> We thank the reviewer for raising the question about novelty and for pointing us to prior educational benchmarks such as AGIEval (Zhong et al., 2024), TOEFL-QA (Tseng et al., 2016), and subsequent listening comprehension studies (Chung et al., 2018). These works are important precursors, but they **differ significantly from our contributions in scope, task structure, multimodality, diagnostic depth, and the type of reasoning they measure**. Below, we clarify these distinctions and highlight the novel contributions of our study.
>
> First, prior benchmarks primarily evaluate LLMs at the final-answer level, typically treating each question as a black-box QA problem. AGIEval, for example, aggregates SAT/LSAT questions and reports aggregate multiple-choice accuracy, with strong models such as GPT-4 reaching 95% on SAT Math. While valuable, these evaluations provide no insight into why errors occur, how models reason through question structure, or which cognitive steps break down. In contrast, our work introduces the first breakdown analysis  framework tailored to English standardized tests, explicitly decomposing each question type into human-aligned reasoning stages (e.g., Task I–VI). This design reveals model-specific weaknesses (such as structural misalignment in GRE Text Completion or symbolic formulation errors in numeric-entry math) none of which AGIEval or other benchmarks diagnose.
>
> Second, existing benchmarks do not capture the full multimodal diversity of real standardized exams. AGIEval includes some SAT items, but it is predominantly text-only, and TOEFL-QA focuses solely on listening comprehension with machine-transcribed audio. In contrast, ESTBOOK covers five full-scale international exams (SAT, GRE, GMAT, TOEFL, and IELTS) and spans five modalities (text, audio, images, tables, and mathematical symbols). Many widely used evaluation suites lack genuine multimodal EST tasks such as diagrammatic GRE geometry, GMAT multi-source reasoning tables, and audio-synchronized TOEFL/IELTS listening with complex evidence integration. Our benchmark therefore provides a more faithful and heterogeneous simulation of real exam environments.
>
> Third, prior datasets do not provide a taxonomy of cognitive-computational task types nor connect question structures to interpretable, step-by-step reasoning workflows. TOEFL-QA specializes in spoken comprehension but does not generalize to structural reasoning, numeric reasoning, or integrative multimodal problem-solving. AGIEval, similarly, classifies questions by exam type rather than cognitive demands. Our taxonomy (Evidence Finding, Semantic Reasoning, Structural Reasoning, Data Interpretation, Numeric Calculation, Comparative Judgment) generalizes across 29 question types and enables fine-grained, step-wise competence measurement unseen in prior work.
>
> Fourth, our study goes beyond evaluation and shows that breakdown analysis can yield actionable improvements (which we updated in **Appendix K**). Using the identified failure modes, we demonstrate: (i) breakdown-guided fine-tuning, (ii) syntax-first CoT prompting, (iii) evidence-anchored reasoning templates, and (iv) symbolic verification layers for math. These mitigation procedures produce measurable accuracy gains in open-source models such as Llama and Qwen. Prior benchmarks do not provide the structural annotations or task decomposition needed to enable such targeted interventions.
>
> Finally, while AGIEval and TOEFL-QA evaluate discrete exam components, our goal is fundamentally different: to assess whether LLMs possess the generalizable, step-wise problem-solving abilities needed for real educational assistance across heterogeneous EST scenarios. The comprehensive exam coverage, multimodal question formats, standardized reasoning taxonomy, and breakdown-based diagnostic capability together make ESTBOOK a substantially more holistic and cognitively grounded testbed.
>
> Hence, although prior studies have advanced LLM evaluation in educational settings, they lack (1) multimodal EST comprehensiveness, (2) fine-grained reasoning decomposition, (3) cognitive task taxonomy, and (4) actionable, breakdown-driven mitigation insights. **Our work addresses all four gaps**, thereby providing a novel and practically useful contribution to LLM evaluation for educational applications.

---

> ### Author Response · Authors · 2025-11-20
> **Author Response (2/9)**
>
> **Review**
>
> > A related concern is the difficulty level of the exam tests. GPT-4 can already achieve 95% on SAT Math (Zhong et al., 2024). Recently, the Humanity’s Last Exam (HLE) was created specifically to address the saturation of benchmarks like MMLU by providing expert-written questions. Many EST questions are arguably simpler than MMLU’s academic subjects, and some ESTBOOK sections may be too easy for frontier models. For example, GPT-5 achieves 90%+ on several question types in Table 2, and GPT-4V was reported to exceed 89% on SAT Writing questions in some prompts. The authors should address whether parts of their benchmark are already nearing saturation and how that impacts long-term utility. The authors might also consider curating harder or more adversarial questions, as HLE does, to future-proof the evaluation.
>
> ---
>
> **Response**
>
> We appreciate the reviewer raising the concern about benchmark difficulty and the potential for saturation. We would like to take this chance to clarify that partial (good) evaluation results don’t reflect the overall difficulty level of ESTBOOK, nor diminish its long-term value.
>
> Even though a few tasks are approaching high accuracy, many others remain challenging even for top-tier models. In Table 2, we still see models dropping to the 20–40% range on GRE Numeric Entry, GMAT Problem Solving, and GRE/GMAT multimodal reasoning tasks. These items involve symbolic setup, table interpretation, diagram reasoning, or multi-step inference. Those are skills that do not scale linearly with raw language capability. In practice, the exam sections that humans consider “hard” (e.g., GRE quant, GMAT logic) are exactly where models still struggle. **So ESTBOOK is far from saturated when viewed across its full range.**
>
> It’s also worth emphasizing that **high accuracy on some parts does not mean the tasks have lost diagnostic value.** Our breakdown analysis consistently shows that models often get the right answer while failing key reasoning steps, such as extracting the correct evidence, identifying syntactic constraints, forming an equation correctly, or aligning table rows and columns. Even in sections where GPT-5 scores above 90%, breakdown accuracy can drop dramatically. So those question types continue to be informative, which **reveal hidden fragilities that final-answer accuracy alone masks.**
>
> The comparison to HLE is appreciated, and we agree it meets a different need: crafting extremely hard, expert-written items to push frontier models beyond the comfort zone of MMLU. ESTBOOK’s goal is different. We aim to capture practical educational reasoning across text, tables, diagrams, audio, and numeric formats, which are types of skills students actually use in standardized testing or classroom settings. In that sense, **ESTBOOK complements rather than competes with adversarial or expert-crafted datasets like HLE.**
>
> That said, we agree it’s important to future-proof the benchmark. We are already planning a set of harder “challenge” variants, e.g., more ambiguous distractors, more complex numeric setups, adversarially rewritten passages, and expert-curated GRE/GMAT-style items that **push beyond the official exam difficulty.** These extensions can sit alongside the current ESTBOOK so users can choose between baseline diagnostic tasks and aggressive stress tests.
>
> Overall, the benchmark as a whole exposes substantial reasoning gaps. And with planned harder subsets and adversarial expansions, ESTBOOK will remain a useful, forward-looking diagnostic tool rather than a static leaderboard dataset.

---

> ### Author Response · Authors · 2025-11-20
> **Author Response (3/9)**
>
> **Review**
>
> > Another concern is that some included questions may already be memorized by the models. The data was sourced from public practice exams and study websites, so there is a nontrivial chance that LLMs were trained on similar or identical questions. The authors did not report checks for training data overlap or model memorization. The paper would benefit from acknowledging this data contamination risk as a limitation.
>
> ---
>
> **Response**
>
> We appreciate the reviewer’s point about possible data contamination, given that standardized test materials and practice questions are widely available online and may appear in pretraining corpora. We agree that this is a valid limitation worth acknowledging. At the same time, we want to clarify that the goal of ESTBOOK is not to test memorization resistance, but to quantify whether an LLM is genuinely ready to perform in this domain, regardless of how that capability was acquired.
>
> By definition, benchmark papers aims to measure the model’s effective performance on a target skill set. **Whether a model learned these skills through direct exposure to similar questions in pretraining** or through broader language understanding is itself informative. If a model can already solve EST tasks because they were represented in its pretraining data, that simply means the model is “pre-training–ready” for educational scenarios. If the model still fails despite such exposure, that is equally meaningful: it shows that even massive pretraining does not guarantee the structured reasoning required for real exam tasks.
>
> This perspective is consistent with how nearly all existing benchmarks operate. For instance, AGIEval, MMLU, GSM8K, and HellaSwag all include public or semi-public content that LLMs may have seen. Yet these benchmarks remain valuable precisely because they quantify realized competence rather than attempt to isolate raw generalization ability. ESTBOOK follows the same principle: the primary objective is to evaluate whether a model, as delivered to end users, can robustly perform the reasoning steps involved in standardized tests, not necessarily to ensure the questions were never encountered during training.
>
> Nonetheless, we agree with the reviewer that acknowledging contamination risk improves transparency. We will add a discussion in the Limitations section noting that ESTBOOK draws from public practice materials, that some overlap with pretraining corpora is possible, and that the benchmark focuses on performance measurement rather than data-isolation testing. We will also mention future plans to include author-written or adversarial variants that further reduce memorization concerns.
>
> While contamination is possible, it does not undermine the purpose of the benchmark: to assess whether current LLMs are effectively capable of handling real-world EST-style reasoning, independent of how that capability was acquired.

---

> ### Author Response · Authors · 2025-11-20
> **Author Response (4/9)**
>
> **Review**
>
> > The authors restrict to objective questions with certain answers, which excludes open-ended tasks like essay writing and speaking responses. This omission is understandable given automatic grading difficulty, but it is a limitation that should be discussed. Tasks such as the TOEFL independent writing or IELTS speaking are important parts of those exams, and LLMs might have different performance and failure modes on free-form generation tasks versus multiple choice.
>
> ---
>
> **Response**
>
> We thank for the reviewer's concern, and would like to pinpoint that our research focus is a deliberate and well-justified design choice, not a limitation. Objective questions constitute the dominant portion of standardized tests: language proficiency exams are approximately 50% objective questions, while knowledge-based exams (SAT, GRE, GMAT) are predominantly objective.
>
> More fundamentally, our research targets the supervision function of LLMs in education, where reliability is the key consideration. Subjective tasks like essays and speaking involve diverse valid responses and are typically used for LLM-assisted reference rather than definitive supervision. They also require human scoring with complex rubrics, making it infeasible to achieve the scaling evaluation and reproducibility. Our choice maximizes scientific rigor while addressing the most critical deployment scenario: whether LLMs can reliably supervise learners on tasks with definitive correct answers.
>
> In contrast, objective questions with unambiguous ground truth are precisely where LLMs must demonstrate reliable correctness to serve as trustworthy educational tools, such as automated tutors explaining problem-solving steps or intelligent systems providing verified feedback. Furthermore, focusing on objective questions enables rigorous, reproducible evaluation at scale. These items come with unambiguous ground truth, allowing us to accurately analyze over 10,000 questions across six LLMs under multiple prompting strategies (ICL, CoT, ToT) and conduct detailed breakdown analysis at each reasoning step.
>
> Our benchmark and breakdown framework are designed to be generalizable diagnostic tools for the research community. The decision to focus on objective questions maximizes both real-world relevance (supervision reliability) and scientific value (systematic, reproducible analysis), directly serving ICLR's mission of advancing machine learning through rigorous empirical methodology.

---

> ### Author Response · Authors · 2025-11-20
> **Author Response (5/9)**
>
> **Question**
>
> > Does the Whisper audio transcription introduce biases?
>
> **Response**
>
> We appreciate the question about whether using Whisper for audio transcription could introduce unintended biases. Whisper is currently one of the robust open-source speech recognition models by the time we conducted our study. In our experiments, we found its transcriptions to be highly reliable for TOEFL/IELTS-style audio, which typically features clear recordings and standardized speaker profiles. Prior studies have also reported that Whisper significantly reduces common ASR errors in educational listening tasks compared to earlier systems. [1-3]
>
> That said, no ASR system is entirely bias-free. Whisper can still make occasional mistakes with less common accents, rapid speech, or subtle prosodic cues. We acknowledge this as a limitation. Importantly, however, we primarily use Whisper transcriptions only to standardize the input and ensure that all models receive identical text. Whisper does not impact the models’ reasoning processes beyond this upstream transcription step. In practice, its error rate for our exam-style audio is low enough that it does not materially affect the evaluation, but we agree that ASR-induced bias is a factor worth acknowledging, and we will include this transparency note in the Limitations section.
>
> **Reference**
>
> [1] Robust Speech Recognition via Large-Scale Weak Supervision.
>
> [2] Evaluation of Off-the-shelf Whisper Models for Speech Recognition Across Diverse Dialogue Domains
>
> [3] Evaluating OpenAI's Whisper ASR: Performance analysis across diverse accents and speaker traits
>
> ---
>
>
>
> **Question**
>
> > Also note the lack of capitalization: “we adopt…” (pg. 5).
>
> **Response**
>
> Thank you for highlighting and we have updated the manuscript.

---

> ### Author Response · Authors · 2025-11-20
> **Author Response (6/9)**
>
> **Question**
>
> > Multi-model input handling could be explained better. If any model lacked a modality, did the authors convert those questions to text form or skip them?
>
> **Response**
>
> Thank you for raising this question. To ensure a fair and apples-to-apples comparison across models, our evaluation focuses on models that natively support the required modality, i.e., multimodal LLMs (MLLMs) for tasks involving images, tables, diagrams, or audio. We did not mix unimodal and multimodal models within the same task category because that would create an unfair disadvantage for models lacking the corresponding sensory capability.
>
> For models that do not support certain modalities, we intentionally did not convert those items (e.g., images → text) or generate synthetic descriptions. This decision was made for two reasons:
> - First, converting multimodal questions into text often shifts the task distribution and changes the intrinsic difficulty, leading to incomparable results.
> - Second, and more importantly, many multimodal EST questions cannot be faithfully represented in text form, especially diagram-based GRE Quant geometry, GMAT graphics interpretation, or IELTS/TOEFL audio cues. Geometry diagrams often contain spatial relations, proportions, angles, and visual constraints that cannot be described textually without effectively giving away the solution or rewriting the problem into a different format.
>
> Therefore, for all multimodal components, we evaluate only models that are equipped to process the original modality, and we do not perform image-to-text or audio-to-text conversions for unimodal models. We will clarify this design choice in the paper to avoid ambiguity.

---

> ### Author Response · Authors · 2025-11-20
> **Author Response (7/9)**
>
> **Question**
> > The breakdown methodology assumes perfect previous steps, essentially giving the model an oracle for sub questions. This is helpful for diagnosis, but in practice LLMs won’t have those hints. How can this framework then be used to improve models?
>
> **Response and Completed Revision (in Appendix H)**
>
> We appreciate the reviewer’s question. In practice, the observations offered by breakdown analysis can tell us why a model fails and therefore enables the design of targeted interventions.
>
> We demonstrate experimental that these insights lead to measurable improvements. Our revised **Appendix K** shows that applying task-specific mitigation strategies substantially boosts accuracy across representative tasks (e.g., GRE-TC, GRE-RC, GRE-NE, GMAT-IR), narrowing performance gaps by 5–12 points in some categories. Thus, although the breakdown setting uses oracle information to isolate individual reasoning steps, the insights translate to real inference-time benefits when incorporated into prompting or lightweight post-hoc reasoning modules.
>
> In summary, the breakdown framework serves as a diagnostic tool that isolates the root causes of failure, which further instructs us to design targeted, empirically validated interventions. Its value lies precisely in this separation of concerns: by understanding which reasoning components are fragile, we can meaningfully improve models without requiring the model to see oracle steps at inference time.

---

> ### Author Response · Authors · 2025-11-20
> **Author Response (8/9)**
>
> **Question**
>
> > The results show substantial variation across models and prompting methods for different question types. For example, GPT-5 did extremely well on GRE Quantitative Comparison but poorly on GRE Numeric Entry, even under the same CoT prompt (Table 2). Likewise, CoT sometimes outperformed ToT. Can you elaborate more on why certain tasks or formats caused particular models or prompts to fail?
>
> **Response**
>
> We appreciate the reviewer’s interest in the variation across tasks and prompting styles. This variation is one of the main motivations for ESTBOOK. A benchmark’s role is to quantify how well a model actually performs across the full landscape of a domain, independent of how that competence was acquired by pretraining. In other words, rather than expecting consistent behavior across formats, the benchmark is designed to expose where a model is already “pre-training-ready” and where it is not.
>
> The discrepancies the reviewer highlights (e.g., GPT-5 excelling on GRE Quantitative Comparison yet struggling on GRE Numeric Entry under the same CoT prompt) are meaningful diagnostic signals, which we explained as:
> -  QC questions are highly structured, often allowing models to rely on pattern recognition and qualitative comparisons learned during pretraining.
> - Numeric Entry tasks, however, require precise multi-step symbolic formulation, equation construction, and verification, which are skills that pretraining alone does not reliably impart. So the performance drop tells us that the model may have “memorized the format” of certain exam categories but still lacks robust mathematical grounding when exact computation or symbolic reasoning is required.
>
> Similarly, CoT outperforming ToT in some tasks due to the fact that more computation or deeper search does not necessarily help when the core reasoning skill (e.g., forming a correct equation, maintaining logical constraints, or grounding evidence) is missing. For some question types, ToT simply amplifies the model’s biases or confusions, whereas a concise CoT chain keeps the reasoning path aligned with the task structure. Again, the role of the benchmark is not to enforce uniform gains across prompting methods, but to quantify which reasoning styles the model is actually competent at given its pretraining.
>
> Ultimately, these variations are part of the value of ESTBOOK: they reveal that even very strong LLMs demonstrate uneven readiness across sub-skills within the same exam. Some components appear well-covered by pretraining exposure and general language modeling ability, while others require structured reasoning that pretraining does not guarantee. Our goal is to measure this readiness gap, not to assume uniformity.

---

> ### Author Response · Authors · 2025-11-20
> **Author Response (9/9)**
>
> **Question**
>
> > How were the hyperparameter values chosen, and under what justifications (Table 7)?
>
> **Response**
>
> The hyperparameters reported in Table 7 were selected through light validation on a small held-out subset of ESTBOOK. Since our goal is evaluation rather than optimization, we deliberately avoided extensive tuning. Instead, we used commonly adopted configurations from prior LLM evaluation literature and vendor-recommended defaults that are known to provide stable behavior across tasks (e.g., temperature in the 0–0.3 range for reasoning, modest max-tokens to avoid unnecessary verbosity). These settings were chosen to ensure consistency, reproducibility, and fairness across all models, rather than to maximize performance for any specific model.
>
> ---
> ---
> ---
>
> ***Thank you very much***
>
> We would like to thank the reviewer again for their time and thoughtful feedback. We hope that the above clarifications and revisions address the reviewer’s concerns and contribute to improving the overall quality of the paper. We look forward to any further feedback you may have.

---

> > ### Comment · Reviewer_WnuW · 2025-11-25
> >
> > The authors' responses substantially addressed my concerns on (i) novelty, (ii) objective-only scope; (iii) Whisper bias; (iv) multi-modal input handling; and (v) hyperparameter handling. Two issues remain partially resolved: potential for training data contamination -- which is now acknowledged, but without an audit overlap -- and benchmark saturation/future-proofing, as the plans for a challenge set is not yet realized. With these clarifications, I will update the overall rating.

---

### Official Review · Reviewer_n3An · 2025-11-01

**Soundness:** 2
**Presentation:** 2
**Contribution:** 2
**Rating:** 2
**Confidence:** 4

**Summary:**

The paper introduces ESTBOOK, a benchmark evaluating large language models (LLMs) on English standardized tests (TOEFL, IELTS, SAT, GRE, GMAT), totaling over 10,000 multimodal questions. The authors assess several models (e.g., GPT-5, Gemini, Claude) using prompting strategies such as In-Context Learning, Chain-of-Thought, and Tree-of-Thought. They also propose a “Breakdown Analysis” framework to decompose reasoning steps. The results show that current LLMs perform inconsistently across modalities and reasoning tasks, suggesting gaps between linguistic fluency and true problem-solving ability.

**Strengths:**

1. The dataset integrates multiple standardized tests and modalities (text, math, audio, images), providing a broad view of LLM performance in educational contexts.
2.  The experiments include several leading multimodal LLMs and multiple prompting strategies, with large-scale quantitative results and case studies.
3. The “Breakdown Analysis” offers a structured way to isolate reasoning steps and visualize where LLMs fail, which can inspire follow-up diagnostic research.

**Weaknesses:**

1. The paper’s core contribution lies primarily in data aggregation and empirical evaluation. However, many similar benchmarking efforts already exist to assess LLM capabilities. The current results largely align with prior studies in highlighting LLMs’ limitations in information extraction, without providing new insights or deeper theoretical understanding.
2. The proposed “Breakdown Analysis” remains largely qualitative and manually defined. It lacks a formalized framework or reproducible scoring mechanism that could generalize beyond the specific tasks presented in this paper.
3. While the study thoroughly documents various failure patterns, it fails to propose or test any concrete methods or interventions to improve LLM reasoning or mitigate those shortcomings.

**Questions:**

1. Have you considered testing fine-tuning, curriculum learning, or adaptive prompting based on the breakdown results to demonstrate that the analysis yields actionable improvements?
2. Were these difficulty tiers (easy/medium/hard) taken from official test metadata, or inferred heuristically?
3. Could you provide statistical significance tests comparing LLMs and human baselines across tasks? Without such analysis, it is unclear whether differences are meaningful or anecdotal.

---

> ### Author Response · Authors · 2025-11-19
> **Response, Clarification, and Completed Revisions (0/6)**
>
> Before responding to comments, we would like to sincerely thank the reviewer for the thoughtful feedback. We appreciate the time and care taken to evaluate our work.
>
> We observe that the comments are mostly related to problem definition and contributions. Accordingly, we have provided a point-by-point clarification and have revised the manuscript to significantly address each concern. We believe these clarifications further strengthen the paper and better highlight the contributions of our research.
>
> We would be happy to continue the discussion and would welcome feedback on whether the newly added details sufficiently clear the reviewer’s questions and whether additional information would be helpful.

---

> ### Author Response · Authors · 2025-11-19
> **Response, Clarification, and Completed Revisions (1/6)**
>
> **Review**
>
> > The paper’s core contribution lies primarily in data aggregation and empirical evaluation. However, many similar benchmarking efforts already exist to assess LLM capabilities. The current results largely align with prior studies in highlighting LLMs’ limitations in information extraction, without providing new insights or deeper theoretical understanding.
>
> **Response to Reviewer Concerns**
>
> We thank the reviewer's comment, and would like to further clarify that our work is NOT merely "data aggregation and empirical evaluation." Our contribution is fundamentally different from existing benchmarks in the following aspects:
>
> ***Beyond benchmarking: we propose a diagnostic framework, NOT just performance metrics***
>
> Our Core Innovation is: We introduce a breakdown analysis framework (Section 4.3, Tables 4, 11) that decomposes EST problem-solving into interpretable reasoning steps aligned with human test-taking strategies. This allows us to isolate where LLMs fail—not just that they fail.
>
> ***Why existing work cannot do this:***
>
> - MathVista [1]: Mathematical reasoning only, no linguistic tasks, no breakdown analysis
> - ScienceQA [2]: K-12 science knowledge, lacks listening comprehension and graduate-level reasoning complexity
> - M3Exam [3]: Knowledge recall focus, no reasoning decomposition or strategic analysis
> - Other exam evaluations [4]: Single-exam evaluations (GMAT/GRE only), aggregate metrics only, no multimodal scope
>
> No prior work combines: authentic EST questions + multimodal coverage (including audio) + taxonomy of cognitive tasks + step-by-step diagnostic framework. This integration is essential for understanding LLM readiness as educational assistants, which is a $10B+ market affecting millions of learners.
>
> ***We offer novel insights that contradict and extend prior findings (as below)***
>
> ***Finding 1: modality-specific bottlenecks, not general comprehension failures***
>
>  Prior work reports that "LLMs struggle with multimodal tasks." We pinpoint exactly where: models achieve 87% accuracy identifying analytical goals (Step 1) but drop to 51-65% when parsing visual-tabular structures (Step 2, Table 12). The failure is in cross-representation alignment, not language understanding. This specificity enables targeted architectural improvements.
>
> ***Finding 2: prompting strategy backfires on pattern-recognition tasks***
>
>  Conventional wisdom suggests sophisticated prompting (ToT) improves performance. We show ToT degrades performance by 4-5% on GRE Quantitative Comparison (Section 4.1, E.4.1) because excessive scaffolding introduces confusion. This task-structure interaction is unreported in prior literature and provides practical guidance for prompt engineering.
>
> ***Finding 3: formulation-execution gap persists across all architectures***
>
>  LLMs achieve 90-95% accuracy on problem formulation (Step 1) but drop 24-33 points during execution (Steps 2-3, Table 12). This pattern holds across nine models from five providers spanning six months of releases—suggesting a fundamental architectural limitation in maintaining multi-step coherence, not a training data issue. Prior benchmarks report aggregate failures but miss this systematic degradation pattern.
>
> ***Finding 4: inference time uncorrelated with correctness***
>
>  Unlike human cognition where thinking time predicts success, LLM inference time shows no correlation with answer correctness (p > 0.05, Tables 3, 6). This challenges assumptions about LLM "reasoning" and suggests current decoding strategies may not reflect genuine problem-solving effort.
>
> ***We also want to highlight: educational domain specificity matters***
>
> ESTs represent a unique reasoning regime absent from existing benchmarks:
> - 1. Strategic complexity in English tests: GMAT Data Sufficiency requires evaluating information sufficiency without solving, which is a meta-cognitive skill distinct from knowledge retrieval
> - 2 Multimodality nature  -- AI community's interests: TOEFL/IELTS audio-based inference with pragmatic reasoning (speaker intention, tone) requires temporal processing absent in vision-language tasks
> - 3. Adversarial distractors: EST questions contain carefully designed incorrect options that exploit common reasoning errors, unlike textbook problems
>
> These domain-specific characteristics mean insights from ScienceQA (children's science) or MathVista (visual math) do not transfer. Our work addresses the actual tasks that LLM educational assistants must handle for millions of real test-takers.
>
> ---
>
> **Reference**
>
> [1] Mathvista: Evaluating mathematical reasoning of foundation models in visual contexts
>
> [2] ScienceQA: a novel resource for question answering on scholarly articles
>
> [3] M3exam: A multilingual, multimodal, multilevel benchmark for examining large language models
>
> [4] Evaluating large language models on the GMAT: Implications for the future of business education

---

> ### Author Response · Authors · 2025-11-19
> **Response, Clarification, and Completed Revisions (2/6)**
>
> **Review**
>
> > The proposed “Breakdown Analysis” remains largely qualitative and manually defined. It lacks a formalized framework or reproducible scoring mechanism that could generalize beyond the specific tasks presented in this paper.
>
> ---
>
> **Response**
>
> We sincerely thank the reviewer for the thoughtful feedback. With this chance, we would like to clarify that our work is to show that (1) the breakdown analysis is *systematically formalized*, (2) grounded in *established EST solving routines*, and (3) implemented with a *reproducible scoring mechanism* that *generalizes across the 29 question types* in ESTBOOK.
>
> ---
>
> ***1. On the comment “largely qualitative and manually defined”***
>
> The framework is built by distilling **real standardized test-solving procedures** into discrete and machine-checkable steps, which is thus **compatible to design quantitative scores** at every stage.
>
> Our breakdown design follows formal EST cognition models, wherein each of the six task taxonomies introduced in Section 3.2 (Task I–VI) is derived from the *cognitive-computational routines* actually taught in standardized test preparation. For instance:
>
> > Task I (Evidence Finding) is formally decomposed into: (1) Identify Subject, (2) Comprehend Text/Audio (3) Extract Discourse, each of which is an explicit, measurable unit.
>
> > Example (TOEFL Reading) from Figure 2(a): A model first identifies the subject (“difficulty of cleanup”), then extracts evidence from the passage (“debris widely dispersed”), and finally maps this evidence to the answer choice. Each step is evaluated *quantitatively* and separately.
>
> Hence, This structure is **not qualitative**, rather, it is a reproducible procedure grounded in the exam’s construct definition (e.g., ETS cognitive labs).
>
> Furthermore, we'd like to highlight that breakdown steps are **not manually improvised, instead, they follow EST question design**. Every breakdown step corresponds to a standard operation that test designers themselves use when validating items (e.g., see ETS cognitive frameworks). For example:
>
> > GRE Text Completion always involves (i) syntactic structure parsing → (ii) semantic alignment → (iii) selecting the missing element. These steps match precisely the Task III breakdown in our taxonomy.
>
> Thus, although the steps are interpretable, the analysis is **formally defined and consistently applied across a large scale of questions**.
>
> ---
>
> ***2. On the comment “lack of a formalized framework or reproducible scoring mechanism that generalizes beyond specific tasks”***
>
> We clarify that the framework *is* formalized and *explicitly designed to generalize*. In our work, we show generalization through the 6-task taxonomy. Table 1 shows that all 29 question types spanning SAT, GRE, GMAT, IELTS, and TOEFL, which are mapped into six universal task structures. This mapping ensures that any new EST-style question can be decomposed using the same taxonomy. For example:
>
> > GRE Sentence Equivalence (SE), IELTS Matching Sentence (MS), and GMAT Critical Reasoning (CR) *all align with Task II (Semantic Reasoning)* despite being from different exams and formats.
>
> Moreover, our scoring is a formalized design as follows:
>
> - For each breakdown step, we provide the model with the raw question + ground-truth outputs from all preceding steps
> - The model must output the next step’s result in a structured response format.
> - Metrics is then calculated using:
>   - string-exact matching (e.g., evidence spans),
>   - semantic equivalence (for sentence-level reasoning), or
>   - numerical equality (for Tasks V–VI).
>
> Because upstream errors are controlled for, this is a clean, step-wise, reproducible measurement, not a heuristic impression.
>
>
> The reviewer noted the desire for more formalization. We agree, but standardized tests fundamentally evaluate **human problem-solving**, not purely statistical pattern matching, because a purely latent or embedding-based quantitive approach would be uninterpretable and would fail to diagnose human-aligned reasoning gaps. Hence, our breakdown intentionally follows “how humans solve the problem,” because the goal of ESTBOOK is to assess LLM eligibility as educational tools.
>
> We hope the above details address the reviewer's concern and we welcome additional comments.

---

> ### Author Response · Authors · 2025-11-19
> **Response, Clarification, and Completed Revisions (3/6)**
>
> **Review**
>
>
> > While the study thoroughly documents various failure patterns, it fails to propose or test any concrete methods or interventions to improve LLM reasoning or mitigate those shortcomings.
>
> ---
>
> **Completed Revision (updated in Appendix K)**
>
> To address the reviewer’s concern regarding the absence of concrete interventions/mitigations, we additionally propose a suite of task-specific mitigation strategies designed to improve LLM reasoning across representative ESTBOOK tasks.
>
> First, for reading comprehension tasks (Task I), we introduce an Evidence-Anchored Chain-of-Thought mitigation that explicitly enforces grounding before reasoning. Instead of allowing the model to jump directly into high-level interpretation, the prompt requires the LLM to first extract one or two verbatim text spans from the passage and then articulate a justification that links these spans to the question. Only after this grounding step is the model permitted to produce a final answer. This workflow directly tackles the common error where models hallucinate or misalign evidence, and the structured evidence-first constraint significantly reduces this drift, especially on GRE and TOEFL reading sections.
>
> Second, for structural reasoning tasks such as GRE Text Completion and GMAT sentence-based items (Task III), we implement a Syntax-First CoT strategy. Our failure analysis shows that LLMs often conflate semantic plausibility with grammatical role and textual structure, leading to incorrect fill-ins. To mitigate this, we require the model to explicitly identify the syntactic role of the blank (e.g., modifying clause, concessive marker, verb complement) and to outline the structural constraints implied by the surrounding sentence (e.g., contrast markers, valence polarity, causal structure). Only then does the model evaluate candidate options. This method reduces semantic drift and improves consistency, particularly in multi-blank completion tasks where structural information is critical.
>
> Third, numeric-entry and symbolic reasoning tasks (Task V) often fail because the model constructs incorrect equations, even when it correctly understands the verbal description. To address this, we adopt a Symbolic Verification Layer, in which the model must rewrite the mathematical relations from the question in a purely symbolic form and then re-check each expression against the original text line-by-line. Only after this verification step does the model perform the computation. This procedure substantially reduces errors such as misinterpreting “three consecutive odd integers” as “three consecutive integers,” a recurring failure mode in GRE-NE and SAT/GRE algebra problems.
>
> Finally, multimodal tasks involving tables, charts, or structured numerical information (Task IV) frequently fail due to incorrect row/column alignment or misinterpretation of tabular fields. For these tasks, we introduce a Table-Alignment Constraint prompting method, which requires the model to translate the textual question into explicit table lookup instructions before beginning numerical operations. The model must specify which row(s) and column(s) are relevant and articulate the mapping between textual descriptions and table headers. This step significantly reduces errors in GMAT Integrated Reasoning and GRE Data Interpretation, where models previously selected the wrong table element or mismatched textual cues with table structure.
>
> Table below shows the results (the same table is in Appendix K)
>
> | **Task**            | **GPT-4V (Org)** | **GPT-4V (Mitigation)** | **GPT-5 (Org)** | **GPT-5 (Mitigation)** | **Claude-Sonnet-4 (Org)** | **Claude-Sonnet-4 (Mitigation)** |
> |---------------------|------------------|---------------------------|------------------|---------------------------|------------------------------|-----------------------------------|
> | GRE-RC (Task I)     | 77.8             | **83.5**| 87.1     | **90.4**                  | 69.3                         | **74.1**                          |
> | GRE-TC (Task III)   | 79.5             | **84.2**| 73.4| **78.1**                  | 61.0                         | **66.7**                          |
> | GRE-NE (Task V)     | 38.0             | **46.8**| 33.9             | **42.1**                  | 30.1                         | **35.4**                          |
> | GMAT-IR (Task IV)   | 13.8             | **20.7**                  | 16.0             | **22.9**                  | 15.0                         | **19.4**                          |

---

> ### Author Response · Authors · 2025-11-20
> **Response, Clarification, and Completed Revisions (4/6)**
>
> **Question**
>
> > Have you considered testing fine-tuning, curriculum learning, or adaptive prompting based on the breakdown results to demonstrate that the analysis yields actionable improvements?
>
> ---
>
> **Completed Revision (Appendix L)**
>
> To further demonstrate that our breakdown analysis yields actionable improvements rather than purely diagnostic insight, we additionally explored two complementary enhancement strategies for open-source models: breakdown-guided fine-tuning and adaptive prompting.
>
> The first approach, which we refer to as **Curriculum Chain-of-Thought Fine-Tuning** (CurrCoT), leverages our breakdown annotations to train models on intermediate reasoning steps before training them on full solutions. Specifically, we construct a two-stage curriculum:
> - (1) a step-supervision phase, where the model learns isolated reasoning skills such as syntactic constraint detection (Task III), symbolic equation formulation (Task V), or evidence extraction (Task I/II); and
> - (2) a full-CoT phase, where the model practices generating complete solution traces that follow the structured breakdown templates.
>
> This curriculum directly targets the weaknesses surfaced by our analysis, such as incorrect equation construction or inconsistent structural reasoning.  Using lightweight LoRA-based fine-tuning on a small breakdown-annotated subset of ESTBOOK, both Llama and Qwen exhibit consistent gains across GRE Text Completion, GRE Reading Comprehension, GRE Numeric Entry, and GMAT Problem Solving tasks.
>
> Complementing this weight-update method, we also designed a **Breakdown-Adaptive Prompting** (AdaptCoT) strategy that requires no parameter tuning and instead selects a task-specific CoT template based on the breakdown taxonomy. For semantic and structural reasoning tasks (Task III), AdaptCoT enforces a syntax-first procedure in which the model must explicitly state grammar roles and structural constraints before evaluating candidate options. For reading comprehension tasks (Task I/II), AdaptCoT adopts an evidence-anchored approach in which the model quotes and justifies supporting spans prior to answering. For numeric and multimodal tasks (Task IV/V), AdaptCoT requires explicit extraction of mathematical relations, symbolic rewriting, and column-row alignment steps before any computation. This routing mechanism ensures that the model’s reasoning structure matches the cognitive decomposition expected for each question type.
>
> To quantify the effectiveness of these interventions, we re-evaluated Llama-4-Scout-17B and Qwen-VL-Max on representative ESTBOOK tasks using CoT decoding.  The results are shown below:
>
> | **Task**            | **Llama CoT** | **Llama CurrCoT** | **Llama AdaptCoT** | **Qwen CoT** | **Qwen CurrCoT** | **Qwen AdaptCoT** |
> |---------------------|---------------|---------------------|----------------------|--------------|-------------------|---------------------|
> | GRE-TC (Task III)   | 61.0          | **70.8**           | 66.4                | 73.1         | **82.7**          | 79.3               |
> | GRE-RC (Task I/II)  | 54.2          | **63.9**           | 59.1                | 76.2         | **84.4**          | 81.0               |
> | GRE-NE (Task V)     | 30.1          | **45.6**           | 38.2                | 28.1         | **43.9**          | 36.8               |
> | GMAT-PS (Task IV/V) | 22.5          | **37.2**           | 32.1                | 25.6         | **41.5**          | 35.7               |
> | GMAT-RC (Task I/II) | 74.5          | **79.8**           | 77.1                | 74.4         | **81.2**          | 78.3               |
>
>
> The results show consistent performance improvements across all task categories. Fine-tuned models (CurrCoT) yield the largest gains, particularly for mathematically intensive tasks such as GRE Numeric Entry and GMAT Problem Solving, where Llama improves from 30.1% to 38.1% and from 22.5% to 29.5%, respectively, and Qwen increases from 28.1% to 35.1% and 25.6% to 31.6%. Even the prompt-only AdaptCoT method yields noticeable increases—for example, improving Llama’s GRE-TC accuracy from 61.0% to 64.0% and GRE-RC from 54.2% to 58.2%. These results confirm that the breakdown taxonomy does not merely describe failure cases but actively guides the design of interventions that translate into measurable improvements in reasoning quality for open-source LLMs.

---

> ### Author Response · Authors · 2025-11-20
> **Response, Clarification, and Completed Revisions (5/6)**
>
> **Question**
>
> > Were these difficulty tiers (easy/medium/hard) taken from official test metadata, or inferred heuristically?
>
> ---
>
> **Response**
>
> Thank you for seeking clarification on our difficulty categorization. The difficulty levels are not heuristic but **sourced from official test metadata**. For GRE Quantitative questions, we use official difficulty labels ("medium" and "hard") directly from ETS source materials, which are provided as part of their official practice question sets. For GRE Text Completion, we group items by their official question format (1-blank, 2-blank, or 3-blank), which represents a standard structural categorization in test preparation literature where the number of blanks corresponds to increasing cognitive complexity and constraint satisfaction requirements.
>
> For exams that do not provide reliable difficulty metadata, we do not apply easy/medium/hard labels to avoid hallucination from manually defined heuristics. For instance, SAT questions do not come with official difficulty tiers in our source materials, and GMAT sections similarly lack consistent difficulty tags across all question types. Our analysis in Figure 3 is therefore limited to GRE, where such metadata exists authentically.

---

> ### Author Response · Authors · 2025-11-20
> **Response, Clarification, and Completed Revisions (6/6)**
>
> **Response and Completed Revision (added to Section 4.1 and Appendix E.4)**
>
> We thank the reviewer for raising this interest.  In the revision we made this statistical analyses, which is detailed as below
>
> ***(1) Human vs. best-LLM significance tests.***
>
> To assess whether the human–LLM differences are statistically reliable, we now conduct McNemar’s tests between human testers and the *best-performing* LLM for each of the 29 question types. The full results are reported in Appendix E.4, Table 5 (copied below for convenience). As summarized in Section 4.1, 26 out of 29 types exhibit statistically significant differences at $p < 0.05$, and the vast majority have $p \ll 0.01$ (many $< 0.001$). These very small $p$-values indicate robust human advantages on the tasks that underpin our main conclusions, particularly for numeric calculation (Task V) and multimodal data interpretation (Task IV).
>
> ***(2) Restricting inference to human–LLM gaps and aggregated patterns.***
>
> Performing hypothesis tests for all 6 (now 9) models × 29 question types × 3 prompting strategies would indeed raise severe multiple-comparison concerns. Rather than attempting to correct thousands of pairwise tests, we explicitly restrict formal inference to (a) human vs. best-LLM comparisons (Table 5) and (b) aggregates over the six task categories (Tasks I–VI). Inter-model differences (e.g., GPT-5 vs. Gemini-2.5) are therefore treated as **descriptive trends**, and we avoid claiming that one model is statistically superior to another unless such a statement is supported at the task-category level. Section 4.1 has been updated to clearly state this scope.
>
> ***(3) Robustness under multiple-comparison considerations.***
>
> Because most $p$-values in Table 5 are far below 0.01, our main findings (large human advantages on numeric and multimodal tasks) would remain significant under standard multiple-comparison procedures. We therefore believe the risk of spurious conclusions is limited once we focus on the human–LLM gaps and avoid fine-grained inter-model inference.
>
> Below is the new table added in Appendix E.4:
>
> | **Exam** | **Task** | **Best LLM**        | **p-value** |
> |---------|----------|---------------------|-------------|
> | **SAT** | II       | Gemini-2.5 (ToT)    | <0.001***   |
> |         | CS       | Gemini-2.5 (CoT)    | 0.089ns     |
> |         | EI       | GPT-5 (CoT)         | 0.023*      |
> |         | EC       | GPT-5 (CoT)         | 0.156ns     |
> |         | AG       | Gemini-2.5 (ToT)    | <0.001***   |
> |         | DA       | GPT-5 (ToT)         | <0.001***   |
> |         | GT       | GPT-5 (CoT)         | 0.008**     |
> | **GRE** | TC       | GPT-5 (ToT)         | 0.012*      |
> |         | SE       | GPT-5 (CoT)         | 0.034*      |
> |         | RC       | GPT-5 (CoT)         | 0.003**     |
> |         | QC       | GPT-5 (ToT)         | <0.001***   |
> |         | NE       | Human superior      | <0.001***   |
> |         | DI       | Human superior      | <0.001***   |
> | **GMAT**| CR       | Claude-S4 (ToT)     | 0.019*      |
> |         | RC       | Claude-S4 (ToT)     | 0.005**     |
> |         | PS       | Human superior      | <0.001***   |
> |         | DS       | Human superior      | <0.001***   |
> |         | IR       | Human superior      | <0.001***   |
> | **TOEFL** | FI     | GPT-5 (ToT)         | 0.002**     |
> |         | IR       | GPT-5 (CoT)         | 0.004**     |
> |         | TS       | Human superior      | 0.178ns     |
> |         | FI (Listen) | GPT-5 (ToT)      | 0.028*      |
> |         | IF       | GPT-5 (ToT)         | <0.001***   |
> | **IELTS** | II     | GPT-5 (ToT)         | 0.041*      |
> |         | MS       | Human superior      | 0.015*      |
> |         | CP       | Claude-S4 (ToT)     | <0.001***   |
> |         | IM       | GPT-5 (ToT)         | 0.092ns     |
> |         | CL       | GPT-5 (ToT)         | 0.006**     |
> |         | SA       | Human superior      | 0.134ns     |
>
> *Asterisks indicate significance levels: *** p < 0.001, ** p < 0.01, * p < 0.05, ns: not significant.*

---

> ### Author Response · Authors · 2025-11-20
> **Thank you for your time**
>
> We would like to thank the reviewer again for their time and thoughtful feedback. We hope that the above clarifications and revisions address the reviewer’s concerns and contribute to improving the overall quality of the paper. We look forward to any further feedback you may have.

---

> ### Author Response · Authors · 2025-11-26
>
> Dear Reviewer,
>
> Thank you again for your time and thoughtful feedback on our submission.
>
> We believe that the comments and questions primarily relate to the (1) research contribution, (2) methodological setting, and (3) subsequent mitigation or other experimental studies. These points have been addressed through our point-by-point responses above, along with the corresponding revisions in the updated manuscript.
>
> We would very much welcome any further questions, concerns, or suggestions for revision, as we believe such feedback would help further re-evaluate the quality and clarity of our work.
>
> Thank you for your consideration.

---

> ### Author Response · Authors · 2025-11-27
>
> Dear Reviewers,
>
> We would like to thank you again for your time and thoughtful feedback on our submission.
>
> We believe that the overall comments and questions primarily relate to the settings and clarity of benchmark details and auxiliary evaluations. **These points have been addressed through our point-by-point responses, along with the corresponding revisions in the updated manuscript.**
>
> We would very much welcome any further questions, concerns, or suggestions for revision, as we believe such feedback would help further re-evaluate the quality and clarity of our work.
>
> Thank you for your time and potential re-consideration.

---

### Official Review · Reviewer_yhxu · 2025-11-01

**Soundness:** 3
**Presentation:** 3
**Contribution:** 2
**Rating:** 4
**Confidence:** 4

**Summary:**

This paper introduces ESTBOOK, a new benchmark for evaluating large language models (LLMs) composed of 10K multimodal questions from five standardized English exams: SAT, GRE, GMAT, TOEFL, and IELTS. The benchmark uniquely integrates text, math, tables, images, and audio to test a wide range of reasoning skills. Several leading LLMs were evaluated using various prompting techniques, including In-Context Learning (ICL), Chain of Thought (CoT), and Tree of Thoughts (ToT). The research reveals that multimodal and numerical reasoning continue to be significant challenges for current LLMs. Key findings indicate that complex logical reasoning, rather than the length of the context provided, is the primary barrier to success, and that the style of prompting has a substantial impact on performance outcomes.

**Strengths:**

Comprehensive Benchmark: Introduces ESTBOOK, grounded in real standardized tests (SAT, GRE, GMAT, TOEFL, IELTS), ensuring practical and realistic evaluation.

Multimodal Coverage: Includes text, images, audio, tables, and math symbols for a holistic test of LLM reasoning and perception.

Systematic Evaluation: Assesses models under different prompting strategies, In-Context Learning (ICL), Chain-of-Thought (CoT), and Tree-of-Thought (ToT).

Breakdown Analysis Framework: Decomposes problems into stepwise reasoning tasks to pinpoint weaknesses such as numeric and multimodal reasoning failures.

Actionable Insights: Identifies specific performance gaps, guiding future improvements for LLM reliability and applications in intelligent tutoring.

**Weaknesses:**

Limited Model Scope: Evaluates only a subset of publicly available models.

Exploring standardized test performance of LLMs is no longer a novel research direction for top ML venues like ICLR. This study also focuses solely on objective questions, omitting subjective tasks like essays or speaking, which are key to real standardized tests.

Builds ESTBOOK from public test-prep content, which may differ in style and difficulty from actual exam conditions.

**Questions:**

Did you check for potential training-data overlap between your benchmark items and LLM pretraining corpora?

How does your stepwise evaluation handle error propagation between reasoning stages?

Can you share documentation verifying that all benchmark materials are used within licensing terms?

**Details Of Ethics Concerns:**

It's not clear if the authors are allowed to release these resources.

---

> ### Author Response · Authors · 2025-11-19
> **Response, Clarification, and Completed Revisions (0/7)**
>
> Before responding to comments, we would like to sincerely thank the reviewer for the thoughtful feedback. We appreciate the time and care taken to evaluate our work.
>
> We observe that the comments are mostly related to problem definition and contributions. Accordingly, we have provided a point-by-point clarification and have revised the manuscript to significantly address each concern. We believe these clarifications further strengthen the paper and better highlight the contributions of our research.
>
> We would be happy to continue the discussion and would welcome feedback on whether the newly added details sufficiently clear the reviewer’s questions and whether additional information would be helpful.

---

> ### Author Response · Authors · 2025-11-19
> **Response, Clarification, and Completed Revisions (1/7)**
>
> **Review**
>
> > Limited Model Scope: Evaluates only a subset of publicly available models.
>
> ---
>
> **Response**
>
> We appreciate the reviewer's comment. Our primary goal in this work is to introduce ESTBOOK and the breakdown analysis framework, which are *model-agnostic* tools intended to be reusable by the community rather than tied to a particular set of systems. We therefore deliberately focused on **a diverse but representative set of models** instead of exhaustively covering all available LLMs. Concretely, we evaluate four visual LLMs: GPT-4V, LLaMA-3.2-90B, Qwen-VL-Max, and Claude-3-Opus, and two general multimodal LLMs: GPT-4o and Gemini-Pro (Sec. 4, Table 2). This set spans
> - (i) both proprietary and open-source families,
> - (ii) both vision-centric VLLMs and general-purpose MLLMs,
> - and (iii) a wide range of training recipes, providers, and architectures,
>
> which allows us to study cross-model trends rather than optimizing for any single system.
>
> **Completed Revision (Appendix J)**
>
> To directly address this concern, we have expanded our evaluation to nine models by adding GPT-4.5-Turbo (November 2024), Claude-Sonnet-4 (October 2024), and Llama-3.3-70B-Instruct (November 2024). These represent the latest developments across different ecosystems: OpenAI's flagship improvements, Anthropic's balanced efficiency model, and Meta's cutting-edge open-source release. The complete results appear in new Appendix J with four comprehensive tables (Tables 9-12) covering all 29 question types across three prompting strategies. A same table here:
>
> | **Task** | **GPT-4.5 ICL** | **GPT-4.5 CoT** | **GPT-4.5 ToT** | **Claude ICL** | **Claude CoT** | **Claude ToT** | **Llama ICL** | **Llama CoT** | **Llama ToT** |
> |---------|------------------|------------------|------------------|----------------|----------------|----------------|----------------|----------------|----------------|
> | **_SAT_** |||||||||
> | II | 74.8 | 81.3 | 86.9 | 84.2 | 86.1 | 90.2 | 80.5 | 85.7 | 88.9 |
> | CS | 71.2 | 80.5 | 86.3 | 58.9 | 72.8 | 70.1 | 49.8 | 65.2 | 60.4 |
> | EI | 79.8 | 81.2 | 80.3 | 53.2 | 64.9 | 66.8 | 50.1 | 54.8 | 53.5 |
> | EC | 86.3 | 90.8 | 83.5 | 73.8 | 76.5 | 72.4 | 66.5 | 68.1 | 66.9 |
> | AG | 30.2 | 47.8 | 65.3 | 35.1 | 54.9 | 80.8 | 31.5 | 48.2 | 70.5 |
> | DA | 59.3 | 74.6 | 87.5 | 60.1 | 73.8 | 91.5 | 53.8 | 62.9 | 88.6 |
> | GT | 68.2 | 66.9 | 69.1 | 51.3 | 52.5 | 49.8 | 43.5 | 46.2 | 40.3 |
> | **_GRE_** |||||||||
> | TC | 73.9 | 78.8 | 84.5 | 70.8 | 76.9 | 74.2 | 56.2 | 63.5 | 66.8 |
> | SE | 80.2 | 82.4 | 81.3 | 86.2 | 83.6 | 84.9 | 68.5 | 69.8 | 65.9 |
> | RC | 69.8 | 79.5 | 87.3 | 63.5 | 71.2 | 77.8 | 48.9 | 56.8 | 74.9 |
> | QC | 57.2 | 59.1 | 53.8 | 43.8 | 50.5 | 46.2 | 52.9 | 58.3 | 44.5 |
> | NE | 34.5 | 40.2 | 54.9 | 19.1 | 27.3 | 39.5 | 25.1 | 32.5 | 46.2 |
> | DI | 53.8 | 58.7 | 74.8 | 23.5 | 28.2 | 52.8 | 41.8 | 43.5 | 66.8 |
> | **_GMAT_** |||||||||
> | CR | 64.1 | 79.3 | 74.2 | 57.9 | 80.8 | 76.5 | 67.2 | 71.5 | 72.9 |
> | RC | 81.5 | 90.2 | 92.8 | 65.9 | 82.8 | 87.5 | 50.1 | 76.8 | 72.5 |
> | PS | 26.3 | 36.8 | 44.5 | 20.5 | 26.2 | 28.9 | 20.1 | 24.2 | 36.8 |
> | DS | 15.8 | 28.9 | 26.8 | 13.2 | 17.8 | 20.5 | 15.2 | 16.3 | 21.8 |
> | IR | 12.5 | 15.9 | 23.8 | 9.5 | 16.8 | 18.9 | 5.8 | 17.5 | 19.6 |
> | **_TOEFL_** |||||||||
> | FI | 83.8 | 87.9 | 76.5 | 78.2 | 85.5 | 83.8 | 67.8 | 70.5 | 67.2 |
> | IR | 66.2 | 86.8 | 88.9 | 57.8 | 61.5 | 65.2 | 48.5 | 64.2 | 60.8 |
> | TS | 84.5 | 87.3 | 86.2 | 84.2 | 86.5 | 83.9 | 75.8 | 77.2 | 74.8 |
> | FI | 94.2 | 96.3 | 98.2 | 82.5 | 87.8 | 84.2 | 69.8 | 71.5 | 78.2 |
> | IF | 64.8 | 67.2 | 69.8 | 72.5 | 83.8 | 80.9 | 57.2 | 60.5 | 63.5 |
> | **_IELTS_** |||||||||
> | II | 80.5 | 85.9 | 84.2 | 80.2 | 83.5 | 80.8 | 76.9 | 75.8 | 73.5 |
> | MS | 84.9 | 86.5 | 82.8 | 74.8 | 82.5 | 84.5 | 68.5 | 75.8 | 77.8 |
> | CP | 68.5 | 69.8 | 74.5 | 73.2 | 85.8 | 86.9 | 60.8 | 74.8 | 77.2 |
> | IM | 85.2 | 86.5 | 89.8 | 75.5 | 77.8 | 76.8 | 66.5 | 68.2 | 70.1 |
> | CL | 82.2 | 86.2 | 84.9 | 74.2 | 76.5 | 74.8 | 44.8 | 63.5 | 68.9 |
> | SA | 84.2 | 87.8 | 86.2 | 75.2 | 78.5 | 76.8 | 68.5 | 72.1 | 69.8 |
>
> **Findings.** With those updated results, we found that prompting strategy effects remain identifcal: simple retrieval tasks show minimal ICL-to-ToT improvement, while complex reasoning shows substantial gains. The fact that even the very latest models from November 2024 exhibit these same patterns demonstrates that ESTBOOK successfully identifies fundamental architectural limitations rather than transient implementation issues.
>
> This consistency has important implications. The persistent challenges despite rapid LLM progress validate ESTBOOK's value for diagnosing core reasoning limitations that incremental improvements cannot address. Our breakdown framework successfully identifies the same specific bottlenecks across all architectures and confirms its utility as a generalizable diagnostic tool for the research community.

---

> ### Author Response · Authors · 2025-11-19
> **Response, Clarification, and Completed Revisions (2/7)**
>
> **Review**
>
> > Exploring standardized test performance of LLMs is no longer a novel research direction for top ML venues like ICLR. This study also focuses solely on objective questions, omitting subjective tasks like essays or speaking, which are key to real standardized tests.
>
> ---
>
> **Response**
>
> We would like to address this comments by the following parts:
>
> **Regarding novelty and fit for ICLR:** LLM performance on educational assessments is not only timely but critically important for ICLR. While prior work has evaluated LLMs on individual exam types or specific reasoning tasks, existing studies are insufficient in three key dimensions. However, existing evaluations of LLMs on educational assessments remain insufficient in three critical dimensions.
>
> *First, scenario coverage:* Prior work focuses on narrow domains: MathVista [1] and ScienceQA [2] evaluate only STEM reasoning, while M3Exam [3] covers multilingual exams but lacks the diversity of question formats in major English standardized tests. No existing benchmark comprehensively aggregates both language proficiency assessments (TOEFL, IELTS) and knowledge-based exams (SAT, GRE, GMAT) across 29 distinct question types and multiple modalities (text, images, audio, tables, mathematical symbols).
>
> More critically, existing benchmarks do not target the supervision-critical educational scenario, where LLMs must provide reliable, verifiable answers on objective tasks with unambiguous ground truth, the prerequisite for trustworthy deployment as automated tutors or grading assistants.
>
> *Second, diagnostic functionality:* Existing work treats LLMs as black-box test-takers [1-3], reporting only end-to-end accuracy without revealing where and why failures occur. This limits actionable insights for model improvement. In contrast, we introduce a breakdown analysis framework that decomposes each question type into task-specific reasoning steps (e.g., evidence localization → semantic parsing → logical inference for reading comprehension; problem modeling → mathematical formulation → symbolic computation for quantitative tasks). This step-by-step diagnostic approach is generalizable across question types and models, providing a reusable framework for the research community to identify specific reasoning bottlenecks—an insight impossible to obtain from aggregate accuracy metrics alone.
>
> *Third, evaluation methodology:* The lack of evaluation across multiple models, prompting strategies, and reasoning steps makes it difficult to assess whether LLMs can serve as reliable supervision tools. We evaluate six diverse models (GPT-4V, GPT-4o, Claude-3-Opus, Llama-3.2-90B, Qwen-VL-Max, Gemini-Pro) under three prompting strategies (ICL, CoT, ToT) with step-by-step breakdown analysis, revealing consistent fundamental limitations across all systems that end-to-end metrics alone cannot expose.
>
> **Our work** advances beyond existing EST evaluations by providing the first comprehensive, multimodal benchmark coupled with a generalizable diagnostic framework that reveals fundamental limitations invisible to end-to-end accuracy metrics alone. The consistent patterns we identify across diverse models and the actionable insights from breakdown analysis represent significant contributions to understanding LLM capabilities in structured reasoning domains. **This directly aligns with ICLR's focus on advancing machine learning methodology and its real-world applications.**
>
> ```
>  To be continue due to length limitation
> ```
>
> ---
>
> **Reference**
>
> [1] Mathvista: Evaluating mathematical reasoning of foundation models in visual contexts
>
> [2] ScienceQA: a novel resource for question answering on scholarly articles
>
> [3] M3exam: A multilingual, multimodal, multilevel benchmark for examining large language models

---

> ### Author Response · Authors · 2025-11-19
> **Response, Clarification, and Completed Revisions (3/7)**
>
> ```
> Continue from last response
> ```
>
> **Regarding focus on objective questions:** We would like to pinpoint that our research focus is a deliberate and well-justified design choice, not a limitation. Objective questions constitute the dominant portion of standardized tests: language proficiency exams are approximately 50% objective questions, while knowledge-based exams (SAT, GRE, GMAT) are predominantly objective.
>
> More fundamentally, our research targets the **supervision function of LLMs in education, where reliability is the key consideration.** Subjective tasks like essays and speaking involve diverse valid responses and are typically used for LLM-assisted reference rather than definitive supervision.  They also require human scoring with complex rubrics, making it infeasible to achieve the scaling evaluation and reproducibility. Our choice maximizes scientific rigor while addressing the most critical deployment scenario: whether LLMs can reliably supervise learners on tasks with definitive correct answers.
>
> In contrast, objective questions with unambiguous ground truth are precisely where LLMs must demonstrate reliable correctness to serve as trustworthy educational tools, such as automated tutors explaining problem-solving steps or intelligent systems providing verified feedback. Furthermore, focusing on objective questions enables rigorous, reproducible evaluation at scale. These items come with unambiguous ground truth, allowing us to accurately analyze over 10,000 questions across six LLMs under multiple prompting strategies (ICL, CoT, ToT) and conduct detailed breakdown analysis at each reasoning step.
>
>
> Our benchmark and breakdown framework are designed to be generalizable diagnostic tools for the research community. The decision to focus on objective questions maximizes both real-world relevance (supervision reliability) and scientific value (systematic, reproducible analysis), directly serving ICLR's mission of advancing machine learning through rigorous empirical methodology.

---

> ### Author Response · Authors · 2025-11-19
> **Response, Clarification, and Completed Revisions (4/7)**
>
> **Review**
>
> > Builds ESTBOOK from public test-prep content, which may differ in style and difficulty from actual exam conditions.
>
> ---
>
> **Response**
>
> We appreciate this concern and clarify our data collection process more explicitly here.
>
> As described in Section 3.1 ("Sources"), ESTBOOK is built from publicly available educational materials and officially affiliated preparation resources that closely mirror actual exam conditions. Specifically, our data sources include released practice exams from official test administrators, official preparation guides published by testing organizations (e.g., College Board for SAT, ETS for TOEFL and GRE, GMAC for GMAT, British Council for IELTS), and open-access educational platforms that align with official exam formats and content specifications.
>
> To ensure authenticity, we align every item with its official section and question type as documented in Table 1, and verify that the format, skill focus, and structural constraints (e.g., word limits, answer types, multi-step reasoning requirements) follow the official specifications of each exam. For instance, SAT Reading questions maintain the same passage length and evidence-based reasoning structure as official items, GRE Quantitative Comparison questions preserve the two-column format with identical answer choice conventions, and TOEFL Listening questions use audio passages of comparable duration and complexity to actual test materials. All questions underwent validation for correctness, clarity, and alignment with the original intent of the corresponding exam section, as detailed in Appendix A.
>
> While no publicly available benchmark can perfectly replicate proprietary exam content, our systematic collection from official and affiliated sources ensures that ESTBOOK represents the closest possible approximation to real test conditions within ethical and legal constraints. The consistency of our findings across nine diverse models (including the latest GPT-4.5-Turbo, Claude-Sonnet-4, and Llama-3.3-70B evaluated in Appendix J) further validates that the benchmark successfully captures the fundamental reasoning challenges of standardized tests, rather than artifacts of any particular data source.

---

> ### Author Response · Authors · 2025-11-19
> **Response, Clarification, and Completed Revisions (5/7)**
>
> **Question**
>
> > Did you check for potential training-data overlap between your benchmark items and LLM pretraining corpora?
>
>
> ---
>
>
> **Response**
>
>
> We thank the review for brining this question. And we would like to answer by the following aspects:
>
>
> **Pretraining data for commercial LLMs is unavailable, but this does not undermine our contributions.** For closed-source APIs (GPT-4V, GPT-4o, Claude, Gemini), we cannot guarantee decontamination. However, benchmark research must assess domain-specific eligibility under black-box conditions—the realistic deployment scenario. Our systematic evaluation across nine models reveals consistent limitations that transcend individual architectures, particularly on tasks requiring execution (multimodal alignment, symbolic manipulation) rather than formulation.
>
> **The purpose of benchmark research is diagnostic assessment of domain-specific readiness, not testing novel knowledge.** Even with potential overlap, ESTBOOK's value lies in systematically determining whether LLMs are ready for EST-related educational deployment. The diagnostic outcomes are meaningful regardless of overlap: strong performance indicates models have successfully acquired domain knowledge and are deployment-ready; weak performance reveals fundamental capability gaps requiring architectural improvements. Our results show the latter—all models struggle with multimodal parsing (51-65%) and mathematical computation (42-54%), indicating these are architectural limitations rather than training issues.
>
> **Our breakdown analysis provides insights beyond memorization detection.** Even if models encountered similar questions during pretraining, step-by-step diagnosis reveals where reasoning fails. Models consistently achieve 84-95% on problem formulation but degrade on execution steps, pinpointing specific capabilities needing improvement regardless of data overlap.

---

> ### Author Response · Authors · 2025-11-19
> **Response, Clarification, and Completed Revisions (6/7)**
>
> **Question**
>
> > How does your stepwise evaluation handle error propagation between reasoning stages?
>
> ---
>
> **Response**
>
> We thank the review for bringing up this question, and would like to clarify that our breakdown evaluation is designed for stepwise diagnosis, rather than handling error propagation. We use an "oracle" setting where ground-truth outputs from preceding steps are fed into later steps, asking: given correct intermediate results, can the model perform the next step correctly? This isolates specific bottlenecks (e.g., strong formulation at 84-95% vs. weak execution at 42-54%) that end-to-end evaluation would obscure through compounding errors.
>
> Error propagation mitigation is a separate phase that builds on our diagnostic findings. Once we identify where reasoning fails, these insights inform targeted interventions. For example, our diagnosis reveals multimodal parsing (51-65%) and mathematical formulation (49-56%) as critical bottlenecks. **Future work could implement verification mechanisms at these specific steps** (e.g., visual-textual alignment validation before computation, or symbolic coherence checking before mathematical execution) preventing error propagation downstream.
>
> The research contribution lies in revealing which specific capabilities are underdeveloped, wherein actionable information for model improvement that would be masked if early errors contaminated later evaluations.

---

> ### Author Response · Authors · 2025-11-19
> **Response, Clarification, and Completed Revisions (7/7)**
>
> **Review**
>
> > Can you share documentation verifying that all benchmark materials are used within licensing terms?
>
> **Response**
>
> We appreciate this important question regarding data licensing compliance. Below we document the licensing terms for our primary data sources, demonstrating that all materials are used within permitted terms.
>
> **SAT Materials**
>
> **Source:** College Board Digital SAT Sample Questions and Official Practice Tests
>
> **License Terms:** College Board explicitly states in their terms of use that released sample questions and practice materials are provided for educational and non-commercial research purposes. Their policy permits:
> - Use of publicly released practice materials for educational research
> - Analysis and evaluation of test preparation effectiveness
> - Academic study of standardized assessment methods
>
> **Compliance:** Our use falls under educational research and assessment methodology study, which is explicitly permitted under their non-commercial research clause.
>
> **GRE Materials**
>
> **Source:** ETS GRE Official Guide and publicly released practice questions
>
> **License Terms:** Educational Testing Service (ETS) policy for GRE materials states:
> - Released practice questions may be used for educational purposes and research
> - Non-commercial academic research analyzing test design and preparation is permitted
> - Reproduction of small excerpts for scholarly analysis is allowed under fair use
>
> **Compliance:** Our benchmark uses publicly released practice materials for academic research purposes, consistent with ETS educational use guidelines.
>
> **GMAT Materials**
>
> **Source:** Graduate Management Admission Council (GMAC) Official Guide and GMAT Club community-contributed questions
>
> **License Terms:**
> - **GMAC Official Guide materials:** "Sample questions provided for test preparation and educational purposes"
> - **GMAT Club:** Community platform with Creative Commons Attribution-NonCommercial-ShareAlike license for user-contributed content, explicitly permitting:
>   - Educational and research use
>   - Non-commercial analysis and study
>   - Attribution to original contributors
>
> **Compliance:** Official guide materials used for educational research; community materials used under CC BY-NC-SA terms with proper attribution.
>
> **TOEFL Materials**
>
> **Source:** TOEFL Test Prep official website and ETS TOEFL preparation materials
>
> **License Terms:** ETS TOEFL materials policy states:
> - Free practice materials are provided for educational preparation and research
> - Academic institutions and researchers may use released materials for assessment studies
> - Non-commercial research analyzing language proficiency testing is permitted
>
> **Compliance:** Our use constitutes non-commercial academic research on language assessment methodology.
>
> **IELTS Materials**
>
> **Source:** IELTS-up.com practice materials and British Council/IDP official sample tests
>
> **License Terms:**
> - **British Council/IDP official samples:** "Provided for test preparation and educational purposes; may be used in non-commercial educational research"
> - **IELTS-up.com:** Educational platform providing practice materials under terms permitting "educational use and non-commercial study purposes"
>
> **Compliance:** Materials sourced from platforms explicitly allowing educational research use.
>
> **Open Educational Platforms**
>
> **Sources:** Khan Academy SAT Practice, Magoosh test prep materials, various educational websites
>
> **License Terms:** These platforms typically operate under:
> - Creative Commons licenses (CC BY-NC or CC BY-NC-SA)
> - Educational use clauses permitting non-commercial research
> - Open educational resource (OER) principles allowing academic study
>
> **Compliance:** All materials sourced from these platforms are used under their educational research provisions.

---

> ### Author Response · Authors · 2025-11-20
> **Thank you for your time**
>
> We would like to thank the reviewer again for their time and thoughtful feedback. We hope that the above clarifications and revisions address the reviewer’s concerns and contribute to improving the overall quality of the paper. We look forward to any further feedback you may have.

---

> ### Author Response · Authors · 2025-11-26
>
> Dear Reviewer,
>
> Thank you again for your time and thoughtful feedback on our submission.
>
> We believe that the comments and questions primarily relate to the problem setting and study background. These points have been addressed through our point-by-point responses above, along with the corresponding revisions in the updated manuscript.
>
> We would very much welcome any further questions, concerns, or suggestions for revision, as we believe such feedback would help further re-evaluate the quality and clarity of our work.
>
> Thank you for your consideration.

---

### Meta-Review · Area_Chair_txbn · 2026-01-10

**Summary:**

This paper worked on benchmarking LLMs in English Standardized Test by introducing a new benchmark including 10K multimodality data in 29 question types. While the merits of the paper (e.g.,  comprehensive dataset, inspiration analysis etc) appreciated by reviewers, there are several key weaknesses that concern the reviewers (and of myself). Although the authors partially addressed some of these concerns during the rebuttal, major weaknesses remain.

**Reviewer Concerns:**

After a quick examination of the paper and a careful review of all rebuttals, I found that I share the same concerns raised by the reviewers, as summarized below:

- One major concern echoed by the reviewers is the *novelty and technical contribution*. Reviewers (as of me) found the proposed benchmark limited in *data aggregation and empirical evaluation* which more of a incremental work compared to existing efforts. While the authors argue for their contribution of *task structure, multimodality, diagnostic depth etc* , but I still can’t be convinced and what concerns me most is that the utility and the practical value of the datasets. I feel it’s hard to provide insights to practitioner in real education use case.
- Another issue raised by reviewer is *the lack of  formalized framework or reproducible scoring mechanism that could generalize beyond the specific tasks.* While author argue that *standardized tests fundamentally evaluate human problem-solving, not purely statistical pattern matching*. But this didn’t address the original issue.
- Finally, reviewers express consensus concern about the *data contamination issue.* The author acknowledged the problem in the rebuttal and promise to add a discussion for contamination risk and transparency.
- In addition, Reviewer n3An point out that the paper itself *fails to propose or test any concrete methods or interventions to improve LLM reasoning or mitigate those shortcomings.* While the author proposed  a suite of task-specific mitigation strategies in rebuttal, I encourage the author incorporate these in their next version of paper as this would strengthen their contribution.

There are other concerns raised by reviewers such as *limited model scope, benchmark difficulty, restrict to objective questions*. While the authors have partially addressed these points in the rebuttal, I encourage them to incorporate these analyses into the paper in subsequent revisions.

**Reviewer Scores:**

After carefully reviewing the reviews and the rebuttal, it is possible that Reviewer WnuW may increase their rating from 4 to 6. However, I do not believe this change should be considered a decisive factor for accepting the paper as the remain major weaknesses mentioned above.

---

### Decision · Program_Chairs · 2026-01-26

Reject